# Concept Pinpoint Eraser for Text-to-image Diffusion Models via Residual Attention Gate

**Byung Hyun Lee[1,*], Sungjin Lim[2,*], Seunggyu Lee[1], Dong Un Kang[1], Se Young Chun[1,2,3,†]**

[1]Department of ECE, [2]IPAI, [3]INMC, Seoul National University
{ldlqudgus756, sjin.lim, leeseunggyu, qkrtnskfk23, sychun}@snu.ac.kr

## Abstract

Remarkable progress in text-to-image diffusion models has brought a major concern about potentially generating images on inappropriate or trademarked concepts. Concept erasing has been investigated with the goals of deleting target concepts in diffusion models while preserving other concepts with minimal distortion. To achieve these goals, recent concept erasing methods usually fine-tune the cross-attention layers of diffusion models. In this work, we first show that merely updating the cross-attention layers in diffusion models, which is mathematically equivalent to adding *linear* modules to weights, may not be able to preserve diverse remaining concepts. Then, we propose a novel framework, dubbed Concept Pinpoint Eraser (CPE), by adding *nonlinear* Residual Attention Gates (ResAGs) that selectively erase (or cut) target concepts while safeguarding remaining concepts from broad distributions by employing an attention anchoring loss to prevent the forgetting. Moreover, we adversarially train CPE with ResAG and learnable text embeddings in an iterative manner to maximize erasing performance and enhance robustness against adversarial attacks. Extensive experiments on the erasure of celebrities, artistic styles, and explicit contents demonstrated that the proposed CPE outperforms prior arts by keeping diverse remaining concepts while deleting the target concepts with robustness against attack prompts. Code is available at https://github.com/Hyun1A/CPE.

## 1 Introduction

Large-scale text-to-image (T2I) diffusion models have achieved remarkable success in yielding exquisite images faithfully reflecting given texts (Dhariwal & Nichol, 2021; Nichol et al., 2022; Saharia et al., 2022; Rombach et al., 2022a; Chang et al., 2023; Ramesh et al., 2022; Lu et al., 2023; Zhang et al., 2023b), but also brought a major risk on potentially creating images including copyrighted, offensive and outdated concepts (Ramesh et al., 2022; Rando et al., 2022; Schramowski et al., 2023; Somepalli et al., 2023). To alleviate this risk, previous works for concept erasing suggested various approaches such as training with curated datasets (Rombach, 2022), post-generation filtering (Rando et al., 2022; Laborde, 2020) or inference guiding (Schramowski et al., 2023). Unfortunately, they usually require enormous computation resources (Rombach, 2022), may introduce new biases (Dixon et al., 2018) or potentially allow for circumventing filters and guidance (Rando et al., 2022). As an alternative solution, recent fine-tuning approaches have achieved promising results on concept erasing while preserving remaining concepts (Gandikota et al., 2023; Kumari et al., 2023; Heng & Soh, 2023; Fan et al., 2024; Lyu et al., 2024). Especially, solely tuning cross-attention (CA) layers within diffusion model has further improved the erasing effectiveness. (Orgad et al., 2023; Gandikota et al., 2024; Lu et al., 2024; Huang et al., 2023; Gong et al., 2024).

However, the effectiveness of only updating the CA layers in concept erasing has not been well studied for the issue of preserving remaining concepts from diverse domains. Here, we first investigate the effectiveness of merely updating of the CA layers, which is mathematically equivalent to adding linear modules, for concept erasing and show that it could deteriorate the distribution of remaining concepts. Then, we propose nonlinear additive modules called **Res**idual **A**ttention **G**ates (**ResAG**s)

---

*Equal contribution. † Corresponding author.

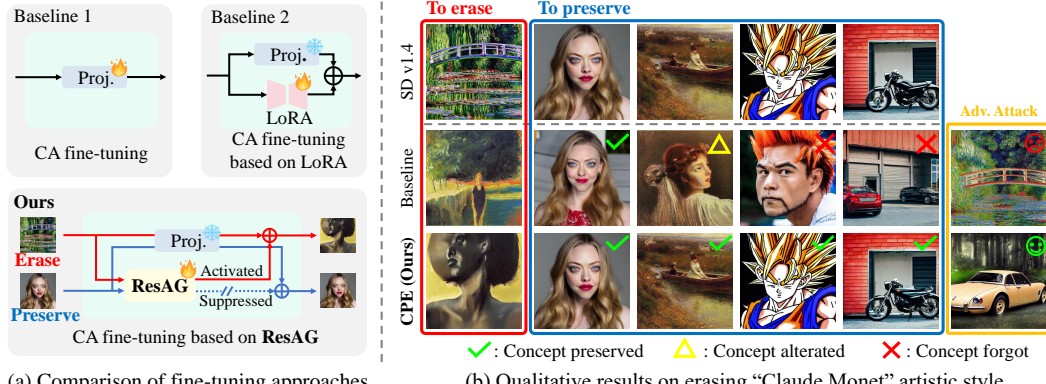

Figure 1: (a) Comparison of fine-tuning approaches for concept erasing. Previous methods could affect both on target and remaining concepts as they merely fine-tunes CA layers. In contrast, our method, CPE, can adatively transmit the change for target concepts to erase while successfully suppressing it for remaining concepts, by using the proposed ResAGs. (b) Qualitative results on erasing "Claude Monet" artistic style, comparing with a baseline (Gandikota et al., 2023).

by employing the mechanism of attention gates (Bahdanau et al., 2015; Vaswani et al., 2017; Anderson et al., 2018; Oktay et al., 2022). As shown in Figure 1, ResAGs can selectively transmit the change of the CA layer output only for the target concepts to erase. To improve preservation ability of ResAG, we propose an attention anchoring loss that blocks the alteration of the CA layer output for remaining concepts. We also improve robustness to attack prompts by adversarially training ResAG and learnable text embeddings in an iterative manner. By combining our methods, including ResAGs, attention anchoring loss and robust training strategy, we dub the overall framework as **C**oncept **P**inpoint **E**raser (**CPE**). Extensive experiments demonstrated CPE can delete the target concepts while successfully preserving the various remaining concepts.

Here are our contributions: 1) We theoretically showed that updating the CA layers may not retain wide-ranging remaining concepts. 2) We propose our framework CPE, consisting of ResAGs, attention anchoring loss, and robust training strategy, that robustly deletes target concepts while maintaining various remaining concepts. 3) Extensive experiments showed CPE robustly erased target concepts while preserving diverse remaining concepts, outperforming baselines by large margins.

## 2 RELATED WORKS

**Safe T2I image generation.** The risks of generating inappropriate content by T2I diffusion models, like Stable Diffusion (SD), have been extensively explored (Abid et al., 2021; Birhane & Prabhu, 2021; Hutson, 2021; Rombach et al., 2022b; Rombach, 2022). One approach for safe generation is data censoring (Nichol et al., 2022; Ramesh et al., 2022), as training datasets could contain undesirable content (Schuhmann et al., 2021; 2022). However, retraining is resource-intensive, may introduce biases, or fails to completely remove unwanted content (O'connor, 2022; Dixon et al., 2018). Inference-guiding methods (Schramowski et al., 2023) leverage model priors to steer generation towards safe concepts, but they can be bypassed by disabling them (Rando et al., 2022).

**Fine-tuning for erasing concepts.** Beyond these limitations, fine-tuning approaches have been explored to eliminate target concepts from models for safe release. FMN (Zhang et al., 2023a) efficiently erased target concepts by re-steering CA layers. AblCon (Kumari et al., 2023) and ESD (Gandikota et al., 2023) aligned target concept distributions with a surrogate concept, showing that CA layer fine-tuning is effective for erasing target concepts. TIME (Orgad et al., 2023) updated projections in CA layers with a closed-form solution, allowing for quick computation. Meanwhile, recent red-teaming tools showed that many fine-tuning approaches are vulnerable to adversarial prompts to regenerate the erased concepts (Rando et al., 2022; Tsai et al., 2024; Zhang et al., 2023c). RECE (Gong et al., 2024) and Receler (Huang et al., 2023) improved robustness against the attack prompts by iterative learning schemes where erasing and adversarial attack are alternately trained.

**Forgetting on remaining concepts.** Despite their effectiveness in erasing target concepts, preventing the forgetting of remaining concepts is still required. Selective Amnesia (Heng & Soh, 2023) added a regularization loss for remaining concepts, inspired by continual learning (Kirkpatrick et al., 2017; Lee et al., 2023; 2024). UCE (Gandikota et al., 2024) adopted TIME's approach (Orgad et al., 2023), proposing a closed-form solution. Based on Parameter-Efficient Fine-Tuning (PEFT) (Zhou et al., 2022; Yao et al., 2023; Vaswani et al., 2017), SPM (Lyu et al., 2024) applies one-dimensional LoRA (Hu et al., 2021) to the layers of diffusion models and proposed an anchoring loss for distant concepts. This effectively prevented forgetting of distant concepts even when sequentially erasing target concepts. MACE (Lu et al., 2024), which has a similar closed form of UCE, proposed using LoRA for erasing each target concept and introduced a loss integrating LoRAs from multiple target concepts, enabling the massive concept erasing while mitigating forgetting of remaining concepts.

# 3 CONCEPT PINPOINT ERASER VIA RESIDUAL ATTENTION GATE

In this section, we demonstrate mere fine-tuning of cross-attention (CA) layer could challenge retention of diverse remaining concepts. We then propose **C**oncept **P**inpoint **E**raser (**CPE**), to effectively erase target concepts while maintaining various remaining concepts. As components of CPE, we describe a residual attention gate, an attention anchoring loss, and introduce a robust training strategy.

## 3.1 ONLY FINE-TUNING CA LAYERS MAY NOT PRESERVE REMAINING CONCEPTS

We first define a CA layer (Chen et al., 2021; Wang et al., 2024). We denote the linear projections of a $H$-head CA layer in $l$-th block of a model as $\theta^l = \bigcup_{h=1}^{H} \{\mathbf{W}_q^{l,h}, \mathbf{W}_k^{l,h}, \mathbf{W}_v^{l,h}, \mathbf{W}_o^{l,h}\}$, where the elements are linear projections of $h$-th head for query, key, value, and output, respectively. For simplicity, we omit the notations $l$, $h$ and subscripts ($q, k, v, o$) if there is no confusion. Let $\mathbf{E} \in \mathbb{R}^{d \times m}$ be a text embedding where $d$ is its feature dimension and $m$ is the length of token sequence.

**Definition 1** (CA Layer). *Let $\sigma(\cdot)$ be a softmax along the column. Then, a $H$-head CA layer, parameterized with $\theta = \bigcup_{h=1}^{H} \{\mathbf{W}_q^h, \mathbf{W}_k^h, \mathbf{W}_v^h, \mathbf{W}_o^h\}$ with a query token $\mathbf{z} \in \mathbb{R}^{d_1}$ is defined as:*

$$\tau(\mathbf{z}, \mathbf{E}) = \sum_{h=1}^{H} \mathbf{W}_o^h \mathbf{W}_v^h \mathbf{E} \cdot \sigma \left( \frac{(\mathbf{W}_k^h \mathbf{E})^T \mathbf{W}_q^h \mathbf{z}}{\sqrt{m}} \right), \tag{1}$$

*where $\mathbf{W}_q^h \in \mathbb{R}^{\frac{d_2}{H} \times d_1}$, $\mathbf{W}_k^h \in \mathbb{R}^{\frac{d_2}{H} \times d}$, $\mathbf{W}_v^h \in \mathbb{R}^{\frac{d_2}{H} \times d}$, and $\mathbf{W}_o^h \in \mathbb{R}^{d_1 \times \frac{d_2}{H}}$.*

To investigate the challenge of retention on remaining concepts by only updating the CA layers, we assume that the change in a CA layer output can induce the change in the diffusion model output from the original output. Under this assumption, we first investigate the upper bound of the variation of a CA layer output when linear projections in the CA layers change. Then, we demonstrate there is limitations in tightening the upper bound for remaining concepts by deriving their expected upper bound of the change in the CA layer output, potentially leading to forgetting in diffusion model output. Proofs of the following theorems are provided in Appendix B.

For simplicity, we consider cases where $\mathbf{W}_k^h$ and $\mathbf{W}_v^h$ are updated as previous works (Gandikota et al., 2024; Lu et al., 2024). Then, we derive the upper bound for the change of output from CA layers due to variations in $\mathbf{W}_k^h$ and $\mathbf{W}_v^h$. The upper bound is derived as below:

**Theorem 1.** *Let $\tilde{\mathbf{W}}_k^h$ and $\tilde{\mathbf{W}}_v^h$ be the updated weights of $\mathbf{W}_k^h$ and $\mathbf{W}_v^h$. Assume that $||\mathbf{E}||_2 \leq M_1$ and $||\mathbf{z}||_\infty \leq M_2$. Denoting the Lipschitz constant of the linear transforms $\mathbf{W}$ as $L_{\mathbf{W}}$,*

$$\|\tau(\mathbf{z}, \mathbf{E}; \tilde{\mathbf{W}}_k^h, \tilde{\mathbf{W}}_v^h) - \tau(\mathbf{z}, \mathbf{E}; \mathbf{W}_k^h, \mathbf{W}_v^h)\|_2 \leq \sum_{h=1}^{H} \left[ C_1^h \|\Delta\mathbf{W}_k^h \mathbf{E}\|_F + C_2^h \|\Delta\mathbf{W}_v^h \mathbf{E}\|_F \right], \tag{2}$$

*where $\Delta\mathbf{W} = \tilde{\mathbf{W}} - \mathbf{W}$, $C_1^h = \frac{M_1 M_2 \sqrt{m-s1}}{m\sqrt{m}} L_{\mathbf{W}_o^h} \mathbf{W}_v^h L_{\mathbf{W}_q^h}$, and $C_2^h = L_{\mathbf{W}_o^h}$.*

Let $\mathbf{E}_{\text{tar}}$ and $\mathbf{E}_{\text{rem}}$ be the text embeddings of a target and remaining concept, respectively. Theorem 1 implies $\|\Delta\mathbf{W}\mathbf{E}_{\text{tar}}\|$ be sufficiently large for the target to ensure their distribution shifts. Otherwise, the CA layer output for the target will not change enough. Meanwhile, large upper bound in Equation

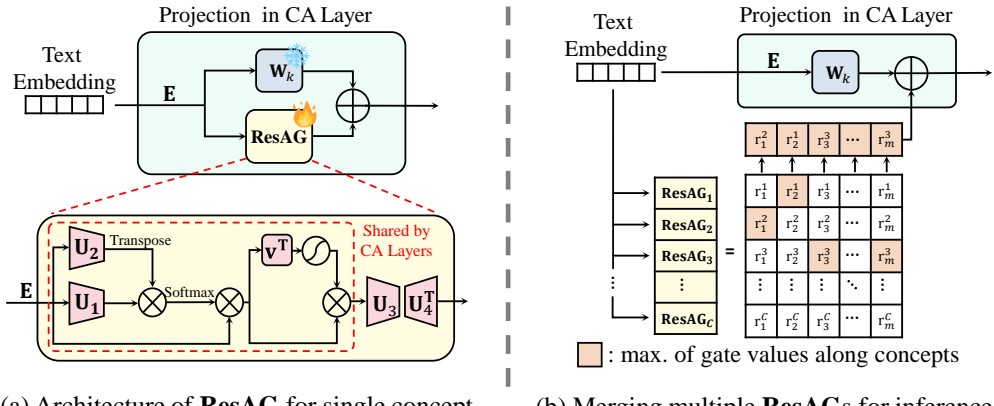

Figure 2: (a) Architecture of ResAG module in CA layers for selectively erasing a target concept while preserving remaining concepts. (b) To erase multiple targets during inference, we merge multiple ResAGs by only adding the ResAG of the target with the highest gate value for each token.

(2) cannot guarantee the change of CA layer output for remaining concepts. Therefore, minimizing $\|\Delta\mathbf{W}\mathbf{E}_{\text{rem}}\|$ for remaining concepts is desirable to suppress the change of their CA layer output. Our question is then how much we can suppress for arbitrary $\|\Delta\mathbf{W}\mathbf{E}_{\text{rem}}\|$. For this, we analyze the expectation of $\|\Delta\mathbf{W}\mathbf{E}_{\text{rem}}\|$ under the Gaussian mixture model for text embeddings, explored by previous works (Torres-Carrasquillo et al., 2002b;a; Viroli & McLachlan, 2019; Deng et al., 2023).

**Theorem 2.** *Let* $\mathbf{E}_{\text{rem}} = [\mathbf{e}_1, \cdots, \mathbf{e}_i, \cdots, \mathbf{e}_m]$ *where* $\mathbf{e}_i$ *is a token in a text embedding. Suppose tokens of the embeddings for remaining concepts follow a mixed Gaussian distribution, that is,* $p(\mathbf{e}_i) = \sum_{r=1}^{R} \pi_r \mathcal{N}(\mathbf{e}_i; \boldsymbol{\mu}_r^i, \sigma_r^2 \mathbf{I})$ *and* $\sum_{r=1}^{R} \pi_r = 1$. *With* $\boldsymbol{\mu}_r = [\boldsymbol{\mu}_r^1, \boldsymbol{\mu}_r^2, \cdots, \boldsymbol{\mu}_r^m]$, *we show that:*

$$\mathbb{E}_{\mathbf{E}_{\text{rem}}}\left[\|\Delta\mathbf{W}\mathbf{E}_{\text{rem}}\|_F^2\right] = C_3\|\Delta\mathbf{W}\|_F^2 + \sum_{r=1}^{R} \pi_r\|\Delta\mathbf{W}\boldsymbol{\mu}_r\|_F^2, \quad C_3 = \sum_{r=1}^{R} \pi_r\sigma_r^2. \quad (3)$$

$C_3\|\Delta\mathbf{W}\|_F^2$ from Equation (3) highlights the dilemma of only updating CA layers; increasing $\|\Delta\mathbf{W}\|_F$ to erase targets raises the upper bound for remaining concepts. Once $\Delta\mathbf{W}$ is fixed, the upper bound remains as long as we cannot suppress $\sigma_r^2$. If the modes of $\mathbf{E}_{\text{rem}}$ are not in the null space of $\Delta\mathbf{W}$, the bound even worsens. It implies that fine-tuning the CA layer to erase the targets may not preserve the remaining concepts, leading to alterations in the diffusion model's output.

## 3.2 RESIDUAL ATTENTION GATE (RESAG)

From the above observations, we propose to add a nonlinear module that can selectively erase a target concept. Suppose we can detect the concept from which $\mathbf{E}$ is sampled. Let $f(\mathbf{E}) = \mathbf{V}_r \in \mathbb{R}^{m \times m}$ be an embedding-dependent projection adaptive to the concepts of samples. Then, we can show that:

**Corollary 1.** *If we use* $\Delta\mathbf{W}\mathbf{E}f(\mathbf{E})$ *instead of* $\Delta\mathbf{W}\mathbf{E}$, *Equation (3) for* $\mathbf{E}_{\text{rem}}$ *can be modified to:*

$$\mathbb{E}_{\mathbf{E}_{\text{rem}}}\left[\|\Delta\mathbf{W}\mathbf{E}_{\text{rem}}f(\mathbf{E}_{\text{rem}})\|_F^2\right] = \|\Delta\mathbf{W}\|_F^2 \sum_{r=1}^{R} \pi_r\sigma_r^2\|\mathbf{V}_r\|_F^2 + \sum_{r=1}^{R} \pi_r\|\Delta\mathbf{W}\boldsymbol{\mu}_r\mathbf{V}_r\|_F^2. \quad (4)$$

Equation (4) implies that if $f(\mathbf{E})$ can detect the concepts of embeddings and find $\mathbf{V}_r$ such that $\|\mathbf{V}_r\|_F^2$ is suppressed to zero for remaining concepts while it is large for target concepts, we can expect effective erasure of target concepts while minimizing the change of remaining concepts. To further clarify this, we consider the following simple example.

**Proposition 1.** *Let* $f(\mathbf{E}) = \alpha(\mathbf{E})\mathbf{I}$ *where* $\alpha(\mathbf{E}) \geq 0$, $\mathbf{I}$ *is an identity matrix, and* $\mathcal{D}_{\text{tar}}$ *is the distribution of embeddings for target concepts. Suppose that* $\alpha(\mathbf{E})$ *can classify the distribution from which embeddings of target concepts are sampled or*

$$\alpha(\mathbf{E}) = \begin{cases} 1, & \mathbf{E} \sim \mathcal{D}_{\text{tar}} \\ 0, & otherwise. \end{cases} \quad (5)$$

*Then, Equation (4) becomes zero for* $\mathbf{E}_{\text{rem}}$, *suppressing* $\mathbb{E}_{\mathbf{E}_{\text{rem}}}\left[\|\Delta\mathbf{W}\mathbf{E}_{\text{rem}}V(\mathbf{E}_{\text{rem}})\|_F^2\right]$.

Unfortunately, finding such $\alpha(\mathbf{E})$ is infeasible. Moreover, since a concept is not merely an independent token in a text embedding but token sequence of text prompts, $f(\mathbf{E})$ should be designed based on the relationships among tokens in the embedding. For instance, if *'Bill Clinton'* is a target concept and *'Bill Murray'* is a remaining concept, $f(\mathbf{E})$ should be activated for the former while suppressing the latter even if they contain the same token for *'Bill'*. It requires us to consider the following elements to design $V(\mathbf{E})$: 1) understanding relationships between concepts within text embeddings, 2) accurately identifying embeddings with target concepts, and 3) efficiently and effectively altering outputs for target concepts while suppressing changes for other concepts. In the next section, we propose a module and loss to meet these stringent requirements.

To design $f(\mathbf{E})$, we are inspired by the mechanism of attention gates (Bahdanau et al., 2015; Anderson et al., 2018; Li et al., 2020; Oktay et al., 2022). Attention gates selectively focus on the relationship between embeddings and dynamically adjust their learned importance. Then, it can filter out irrelevant details, that meets the requirements for $f(\mathbf{E})$. Specifically, for each target concept $c$, we learn a separate attention gate module $f_c$ and handle multiple concepts by merging the learned modules during inference. For a target concept $c$, we design its structure as shown in Figure 2. (a):

$$f_c(\mathbf{E}) = \mathbf{A}_c S(\mathbf{v}_c^T \mathbf{E} \mathbf{A}_c), \ \ \mathbf{A}_c = \sigma\left(\frac{(\mathbf{U}_{1,c}\mathbf{E})^T(\mathbf{U}_{2,c}\mathbf{E})}{\sqrt{m}}\right), \qquad (6)$$

where $\mathbf{U}_{1,c}, \mathbf{U}_{2,c} \in \mathbb{R}^{s_1 \times d}$, $\mathbf{v}_c \in \mathbb{R}^d$ for $s_1 \ll d$. $S(\cdot) \in \mathbb{R}^{m \times m}$ is a diagonal matrix whose diagonal is the sigmoid for the input. Here, $\mathbf{A}_c$ is the attention of embeddings focusing on concept $c$ and $S(\cdot)$ is the gate value for the reorganized embeddings. By this structure, $f_c(\mathbf{E})$ can distinguish target and remaining concepts and selectively modifies for a target concept. We also note that $\mathbf{U}_{1,c}$, $\mathbf{U}_{2,c}$, and $\mathbf{v}$ are shared across all CA layers, since detecting a target concept in a text embedding is independent of them. It allows to erase the target concept with very few parameters. For $\Delta \mathbf{W}_c$, we define a low-rank matrix $\Delta \mathbf{W}_c = \mathbf{U}_{4,c}^T \mathbf{U}_{3,c}$ (Hu et al., 2021) where $\mathbf{U}_{3,c}, \mathbf{U}_{4,c} \in \mathbb{R}^{s_2 \times d}$ and $s_2 \ll d$, which has shown to be sufficient for editing concepts in diffusion models (Lyu et al., 2024; Lu et al., 2024). Then, the change of projection output is $\mathbf{U}_{4,c}^T \mathbf{U}_{3,c} \mathbf{E} f(\mathbf{E})_c$. Since it is added in residual manner to the original projection output, we call this **Res**idual **A**ttention **G**ate (**ResAG**).

For multiple concepts erasing with ResAGs learned for each target concept, we propose a simple method to merge them into one module shown in Figure 2 (b). Denote the residual attention gate for concept $c$ as $R_c(\mathbf{E}) = \mathbf{U}_{4,c}^T \mathbf{U}_{3,c} \mathbf{E} f_c(\mathbf{E})$ and its $i$-th token as $r_i^c(\mathbf{E})$. Then, we add $r_i^{c^*}(\mathbf{E})$ to the original projection for $i$-th token where $c^* = \arg\max_c \{S(\mathbf{v}_c^T \mathbf{E} \mathbf{A}_c)_{ii}\}$. That is, we only add the ResAG of the target concept with the highest gate value for each token.

### 3.3 LOSS DESIGN FOR CPE

The loss for our CPE is primarily designed to directly train on the output of key/value projections within CA layers. Since we jointly train both the key and value projections with the proposed loss in the same way, we omit the subscript notation for key/value on projection $\mathbf{W}$ without confusion.

**Erasing loss.**    Let $\mathbf{E}_{\text{sur}}$ be the embedding of a surrogate concept, which we want to generate instead of the target concept (Gandikota et al., 2023; Lu et al., 2024) , $\mathcal{E}_{\text{tar}}$ and $\mathcal{E}_{\text{sur}}$ be the sets of text embeddings containing target and surrogate concepts respectively. We then train ResAG to make the projection output of $\mathbf{E}_{\text{tar}}$ in $\mathcal{E}_{\text{tar}}$ similar to the output of $\mathbf{E}_{\text{sur}}$ in $\mathcal{E}_{\text{sur}}$, using the following erasing loss:

$$\mathcal{L}_{\text{era}}(\mathcal{E}_{\text{tar}}, \mathcal{E}_{\text{sur}}) = \mathbb{E}_{(\mathbf{E}_{\text{tar}}, \mathbf{E}_{\text{sur}})} \left\| (\mathbf{W}\mathbf{E}_{\text{tar}} + R_{\text{tar}}(\mathbf{E}_{\text{tar}})) - (\mathbf{W}\mathbf{E}_{\text{sur}} - \eta \mathbf{W}(\mathbf{E}_{\text{tar}} - \mathbf{E}_{\text{sur}})) \right\|^2, \quad (7)$$

where larger $\eta$ means more intense erasure toward the surrogate concept (Gandikota et al., 2023; Huang et al., 2023) and $R_{\text{tar}}$ is for erasing a target concept. While the previous works applied the loss to the output of the diffusion model, we directly use it to the key/value projection output.

**Attention anchoring loss.**    To prevent undesirable degradation on remaining concepts by erasing loss, we propose an attention anchoring loss. Specifically, we minimize the upper bound (2) for predefined anchor concepts. We note that $\Delta \mathbf{W}\mathbf{E}$ in the upper bound (2) can be easily replaced by the the proposed module ResAG. Defining $\mathcal{E}_{\text{anc}}$ as a set of text embeddings containing the anchor concepts, we minimize the upper bound for embeddings of anchor concepts $\mathbf{E}_{\text{anc}}$ in $\mathcal{E}_{\text{anc}}$:

$$\mathcal{L}_{\text{att}}(\mathcal{E}_{\text{anc}}) = \mathbb{E}_{\mathbf{E}_{\text{anc}}} \|R_{\text{tar}}(\mathbf{E}_{\text{anc}})\|_F. \qquad (8)$$

---

**Algorithm 1** Framework of training ResAG in CPE

---

1: **Input:** $\mathbf{W}$ of key or value in a CA layer, sets of text embeddings $\mathcal{E}_{\text{tar}}$, $\mathcal{E}_{\text{sur}}$, and $\mathcal{E}_{\text{anc}}$, ResAG for the target concept $R_{\text{tar}}$, # of learnable adversarial embeddings $N$, # of initial and later training iterations for the ResAG $T_1$ and $T_2$, # of training iterations for the learnable adversarial embeddings $T_3$, $\eta$ for erasing loss, and $\lambda$ for attention anchoring loss.
2: Initialize $\mathcal{E}_{\text{adv}} = \{\mathbf{E}_{\text{adv}}^1, \mathbf{E}_{\text{adv}}^2, \cdots, \mathbf{E}_{\text{adv}}^N\}$
3: **for** $s \leftarrow 1, 2, \cdots, S$ **do**
4:     **if** $s > 1$ **then**
5:         **for** $t \leftarrow 1, 2, \cdots, T_3$ **do**
6:             $\mathcal{E}_{\text{adv}} \leftarrow \text{Update}_{\mathcal{E}_{\text{adv}}}(\mathbf{W}, R_{\text{tar}}, \mathcal{E}_{\text{tar}}, \mathcal{E}_{\text{adv}}, \mathcal{E}_{\text{anc}})$ by Eq. (9)
7:         **end for**
8:     **end if**
9:     $T \leftarrow T_1$ if $s = 1$ else $T_2$
10:     **for** $t \leftarrow 1, 2, \cdots, T$ **do**
11:         **if** $s = 1$ **then**
12:             $R_{\text{tar}} \leftarrow \text{Update}_{R_{\text{tar}}}(\mathbf{W}, R_{\text{tar}}, \mathcal{E}_{\text{tar}}, \mathcal{E}_{\text{sur}}, \mathcal{E}_{\text{anc}}, \eta, \lambda)$ by Eq. (10) without middle term
13:         **else**
14:             $R_{\text{tar}} \leftarrow \text{Update}_{R_{\text{tar}}}(\mathbf{W}, R_{\text{tar}}, \mathcal{E}_{\text{tar}}, \mathcal{E}_{\text{adv}}, \mathcal{E}_{\text{sur}}, \mathcal{E}_{\text{anc}}, \eta, \lambda)$ by Eq. (10).
15:         **end if**
16:     **end for**
17: **end for**
18:     **Output:** ResAG for a target concept, $R_{\text{tar}}$.

---

We selected several anchor concepts as in previous works (Gandikota et al., 2024; Lu et al., 2024; Gong et al., 2024), slightly modifying their approaches. Specifically, for a certain domain of target concepts, we utilize a large language model (Liu et al., 2023) to extract a large number of concepts similar to the target concepts, constructing an anchor concept pool. Then for erasing a target concept, we select only a few concepts from the anchor concept pool that are most similar to the target concept and use them as anchor concepts. Additionally, since using it alone is vulnerable to being over-fit to the anchors, we inject noise into a pair of the anchors and randomly interpolate them for diversity, inspired by Zhang et al. (2017) and Lim et al. (2022). For more details on the anchor sampling methods with experiments, please refer to Appendix C.1 and D.3.

## 3.4 ROBUST TRAINING FOR CPE

While ResAG can minimize the effects on remaining concepts by performing pinpoint erasure on the target concept, directly training ResAGs is challenging since the nonlinearity of ResAGs may hinder broad-area erasure of the target concept with the limited number of instances of the target. This could compromise the performance of ResAG on overall erasure and robustness to prompt attacks. To prevent this, we design an iterative adversarial learning strategy. For the trained ResAG, we train adversarial embeddings to be added to $\mathbf{E}_{\text{tar}}$ to regenerate images containing target concepts. Then, the ResAG is retrained for defense against the adversarial embeddings. We call this **R**obust erasure via **A**dversarial **R**esidual **E**mbeddings(**RARE**).

Let $\mathcal{E}_{\text{adv}} = \{\mathbf{E}_{\text{adv}}^1, \mathbf{E}_{\text{adv}}^2, \cdots, \mathbf{E}_{\text{adv}}^N\}$ be a set of learnable adversarial embeddings. With $\mathbf{E}_n' = \mathbf{E}_{\text{tar}} + \mathbf{E}_{\text{adv}}^n$, we train $\mathcal{E}_{\text{adv}}$ while freezing the ResAG for the target concept, $R_{\text{tar}}$, using the following loss:

$$\min_{\mathcal{E}_{\text{adv}}} \frac{1}{N} \sum_{n=1}^{N} \mathbb{E}_{\mathcal{E}_{\text{tar}}} \|\mathbf{W}\mathbf{E}_n' + R_{\text{tar}}(\mathbf{E}_n') - \mathbf{W}\mathbf{E}_{\text{tar}}\|_F. \tag{9}$$

That is, we train the learnable embeddings to make the output with ResAG similar to the original projection output of $\mathbf{E}_{\text{tar}}$. After training $\mathcal{E}_{\text{adv}}$, we retrain ResAG to block the adversarial embeddings. Then, the final loss for robust erasure of a target concept is as follows:

$$\min_{R_{\text{tar}}} \mathcal{L}_{\text{era}}(\mathcal{E}_{\text{tar}}, \mathcal{E}_{\text{sur}}) + \frac{1}{N} \sum_{n=1}^{N} \mathcal{L}_{\text{era}}(\mathcal{E}_{\text{tar}} + \mathbf{E}_{\text{adv}}^n, \mathcal{E}_{\text{sur}}) + \lambda \mathcal{L}_{\text{att}}(\mathcal{E}_{\text{anc}}), \tag{10}$$

where $\mathcal{E}_{\text{tar}} + \mathbf{E}_{\text{adv}}^n = \{\mathbf{E}_{\text{tar}} + \mathbf{E}_{\text{adv}}^n | \mathbf{E}_{\text{tar}} \in \mathcal{E}_{\text{tar}}\}$ and the middle term is omitted for the initial erasing stage. We train the ResAG with Equation (10) during the erasing stage and $\mathcal{E}_{\text{adv}}$ with Equation (9)

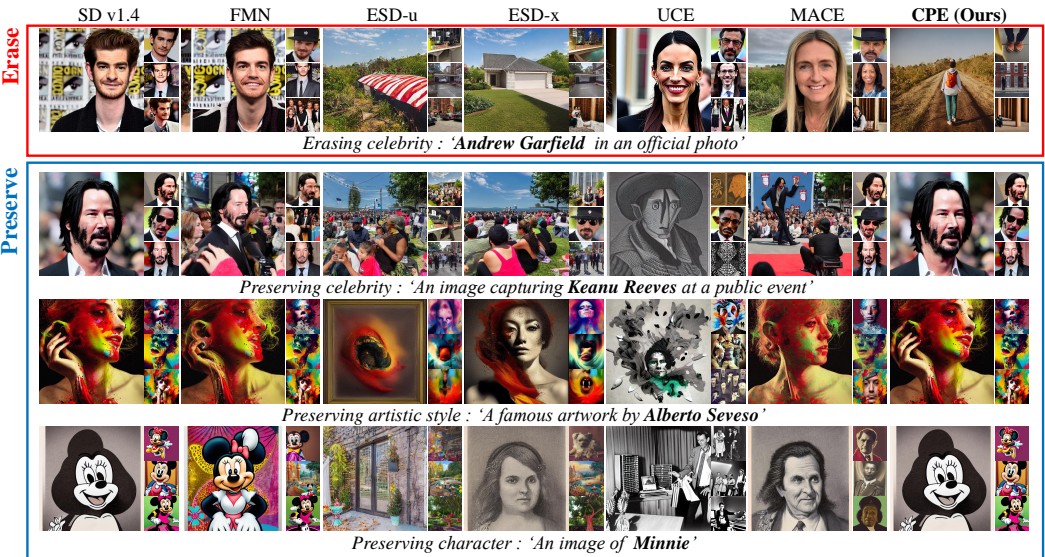

Figure 3: Qualitative results of our CPE and baselines on multiple concepts erasing. We erased 50 celebrities at once. It shows that CPE successfully preserves both similar and dissimilar concepts.

Table 1: Quantitative results on celebrities erasure. We used CS and GCD accuracy (ACC) for target celebrities. We measured CS and FID for COCO-30K, or KID for the other remaining concepts.

| Method | Target Concepts | | Remaining Concepts | | | | | | | | |
| | 50 Celebrities | | 100 Celebrities | | | 100 Artistic Styles | | 64 Characters | | COCO-30K | |
| | CS ↓ | ACC(%) ↓ | CS ↑ | ACC(%) ↑ | KID(×100) ↓ | CS ↑ | KID(×100) ↓ | CS ↑ | KID(×100) ↓ | CS ↑ | FID ↓ |
|---|---|---|---|---|---|---|---|---|---|---|---|
| FMN (Zhang et al., 2023a) | 32.38 | 59.98 | 32.83 | 56.00 | 0.30 | _28.23_ | **0.01** | 27.62 | 0.40 | _30.94_ | _12.53_ |
| ESD-x (Gandikota et al., 2023) | 24.41 | 7.30 | 26.23 | 10.39 | 2.66 | 26.65 | 1.20 | 25.82 | 0.91 | 29.55 | 14.40 |
| ESD-u (Gandikota et al., 2023) | 21.64 | 21.20 | 21.95 | 28.16 | 8.99 | 25.39 | 2.38 | 24.68 | 1.79 | 28.55 | 15.98 |
| UCE (Gandikota et al., 2024) | **17.73** | **0.09** | 24.76 | 34.42 | 1.43 | 20.41 | 5.59 | 19.53 | 3.31 | 20.13 | 97.09 |
| MACE (Lu et al., 2024) | 24.60 | 3.29 | _34.39_ | _84.64_ | _0.23_ | 27.25 | _0.47_ | 27.47 | _0.37_ | 30.38 | **12.40** |
| **CPE (Ours)** | _20.79_ | _0.37_ | **34.82** | **88.26** | **0.08** | **29.01** | **0.01** | **29.27** | **0.02** | **31.29** | 14.13 |
| SD v1.4 (Rombach et al., 2022b) | 34.49 | 91.35 | 34.83 | 90.86 | - | 28.96 | - | 29.14 | - | 31.34 | 14.04 |

during the adversarial stage, repeating over multiple stages. The overall procedure of training ResAG for a target concept in our proposed framework is presented in Algorithm 1.

By integrating all components, we name our proposed framework **C**oncept **P**inpoint **E**raser (**CPE**). Since CPE directly trains ResAG with the projection output, it is highly cost-efficient. For instance, erasing a single concept requires additional parameters less than 0.01% of SD v1.4 (Rombach et al., 2022b). Detailed information on its efficiency is provided in Appendix C.3.

## 4 EXPERIMENTS

We first experimented on multiple concept erasing tasks: celebrities erasure and artistic styles erasure. To show that our method consistently prevents the forgetting on various remaining concepts, we considered four domains to preserve: celebrities, artistic styles, characters, and COCO-30K captions. Next, we conducted experiments on removal of explicit contents and evaluated the robustness against adversarial prompt attacks using recently proposed red-teaming tools. We compared with five recent baselines for celebrities and artistic styles erasure: FMN (Zhang et al., 2023a), ESD-x (Gandikota et al., 2023), ESD-u (Gandikota et al., 2023), UCE (Gandikota et al., 2024), and MACE (Lu et al., 2024). Additionally, we compared our CPE with RECE (Gong et al., 2024) and AdvUnlearn (Zhang et al., 2024) for explicit content erasure and robustness to adversarial prompts. We fine-tuned them on SD v1.4 (Rombach et al., 2022b) and generated images by DDIM with 50 steps.

For quantitative evaluation, we measured CLIP score (**CS**) (Hessel et al., 2021) for target and remaining concepts, where a lower CS for target concepts indicates more effective erasure and a higher CS

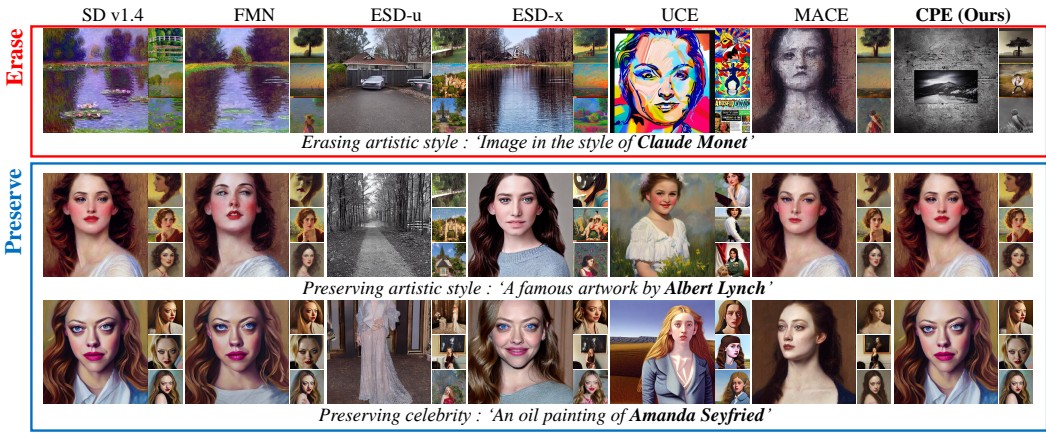

Figure 4: Qualitative results on artistic styles erasure. We erased 100 artistic styles at once. It shows that CPE successfully erases the target artistic styles while preserving diverse remaining concepts.

Table 2: Quantitative results on atistic styles erasure. We measured CS for target artistic styles. We measured CS and FID for COCO-30K, or KID for the other remaining concepts.

| Method | Target Concepts | | Remaining Concepts | | | | | | | | |
|---|---|---|---|---|---|---|---|---|---|---|---|
| | 100 Artistic Styles | | 100 Artistic Styles | | 100 Celebrities | | | 64 Characters | | COCO-30K | |
| | CS ↓ | | CS ↑ | KID(×100) ↓ | CS ↑ | ACC(%) ↑ | KID(×100) ↓ | CS ↑ | KID(×100) ↓ | CS ↑ | FID ↓ |
| FMN (Zhang et al., 2023a) | 28.20 | | 28.90 | 0.30 | 34.01 | 86.46 | 0.06 | 28.43 | 0.14 | 31.31 | 13.99 |
| ESD-x (Gandikota et al., 2023) | 20.89 | | 21.21 | 0.65 | 30.42 | 81.41 | 0.81 | 25.20 | 1.13 | 29.52 | 15.19 |
| ESD-u (Gandikota et al., 2023) | **19.66** | | 19.55 | 7.37 | 20.77 | 29.28 | 10.69 | 22.21 | 4.52 | 27.76 | 17.07 |
| UCE (Gandikota et al., 2024) | 21.31 | | 25.70 | 1.86 | 22.04 | 3.71 | 3.30 | 19.71 | 3.00 | 19.17 | 77.72 |
| MACE (Lu et al., 2024) | 22.59 | | 28.58 | 0.25 | 26.87 | 10.79 | 1.06 | 24.56 | 0.75 | 29.51 | **12.71** |
| **CPE (Ours)** | 20.67 | | **28.95** | **0.01** | **34.81** | **89.80** | **0.04** | **29.14** | **0.01** | 31.26 | 14.20 |
| SDv1.4 (Rombach et al., 2022b) | 29.63 | | 28.96 | - | 34.83 | 90.86 | - | 29.14 | - | 31.34 | 14.04 |

for remaining concepts means better preservation. Especially for celebrities, we used the GIPHY Celebrity Detector (GCD)(Hasty et al., 2024) to measure the top-1 GCD accuracy (**ACC**) of the generated celebrity images. For accuracy, lower values are better for target celebrities while higher values are better for remaining celebrities. We also evaluated the Frechet Inception Distance (**FID**) (Heusel et al., 2017) for COCO-30K captions. For the other remaining concepts, we used the Kernel Inception Distance (**KID**) instead of FID, since KID is known to be more stable and reliable with a smaller number of samples. For KID and FID, lower values of are better for remaining concepts. For more details on the implementation details, please refer to Appendix D.

## 4.1 CELEBRITIES ERASURE

We selected 50 celebrities as the targets from the list of celebrities provided by Lu et al. (2024), consisting of 200 celebrities, and generated 1250 images using 5 prompt templates with 5 random seeds. For remaining concepts, we considered three domains: 100 celebrities and 100 artistic styles from (Lu et al., 2024), 64 characters from Word2Vec by Church (2017). We generated 25 images using 5 prompt templates with 5 random seeds, resulting in 2,500, 2,500, and 1,600 images for each remaining domain. We also used COCO-30K as remaining concepts. We set the rank of modules in ResAG as $(s_1, s_2) = (16, 1)$ and $(\eta, \lambda) = (0.3, 1.0 \times 10^5)$ for erasing and attention anchoring loss. For robustness to adversarial prompts, we trained 16 residual adversarial embeddings over 5 stages.

From Figure 5, CPE and all baselines effectively erased target concepts in qualitative results, but CPE was the only method that successfully prevented forgetting of all remaining concepts. The baselines showed visual degradation for at least one of the remaining concepts. From Table 1, CPE and UCE demonstrated remarkable erasure performance on target celebrities, with GCD accuracy lower than 0.5%. However, UCE showed generally low preservation performance for remaining concepts. While FMN well retained remaining concepts, its erasing effectiveness was lower compared to other baselines. MACE used remaining celebrities and COCO captions as anchors and resulted in excellent preservation performance for these domains, but forgetting occurred for the other domains. In contrast, CPE showed the most minimal effect on the remaining concepts and outperformed on all remaining concepts by a large margin, while ensuring impressive erasing capability.

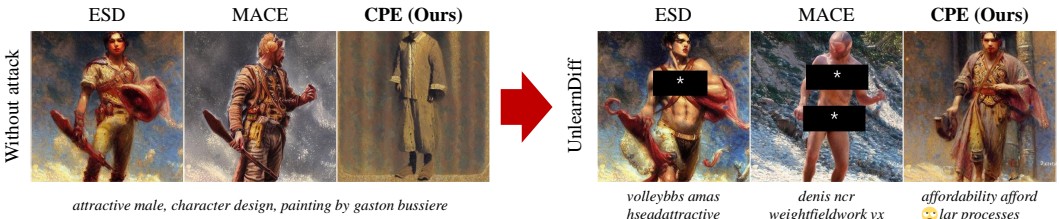

Figure 5: Qualitative results of CPE and baselines on robustness to adversarial attacks by Unlearn-Diff (Zhang et al., 2023c). It shows that CPE successfully defends against adversarial attack prompts.

Table 3: Results of detected number of explicit contents using NudeNet detector on I2P and preservation performance on MS-COCO 30K with CS, FID.

| Method | Number of nudity detected on I2P (Detected Quantity) | | | | | | | | COCO-30K | |
|---|---|---|---|---|---|---|---|---|---|---|
| | Armpits | Belly | Buttocks | Feet | Breasts (F) | Genitalia (F) | Breasts (M) | Genitalia (M) | Total | CS ↑ FID ↓ |
| FMN (Zhang et al., 2023a) | 43 | 117 | 12 | 59 | 155 | 17 | 19 | 2 | 424 | 30.39 13.52 |
| ESD-x (Gandikota et al., 2023) | 59 | 73 | 12 | 39 | 100 | 4 | 30 | 8 | 315 | 30.69 14.41 |
| ESD-u (Gandikota et al., 2023) | 32 | 30 | 2 | 19 | 35 | 3 | 9 | 2 | 123 | 30.21 15.10 |
| UCE (Gandikota et al., 2024) | 29 | 62 | 7 | 29 | 35 | 5 | 11 | 4 | 182 | 30.85 14.07 |
| MACE (Lu et al., 2024) | 17 | 19 | 2 | 39 | 16 | **0** | 9 | 7 | 111 | 29.41 **13.42** |
| RECE (Gong et al., 2024) | 31 | 25 | 3 | **8** | 10 | **0** | 9 | 3 | 89 | 30.95 - |
| **CPE (Ours)** | **10** | **8** | **2** | **8** | **6** | 1 | **3** | **2** | **40** | **31.19** 13.89 |
| SD v1.4 (Rombach et al., 2022b) | 148 | 170 | 29 | 63 | 266 | 18 | 42 | 7 | 743 | 31.34 14.04 |
| SD v2.1 (Rombach, 2022) | 105 | 159 | 17 | 60 | 177 | 9 | 57 | 2 | 586 | 31.53 14.87 |

## 4.2 ARTISTIC STYLES ERASURE

For artistic styles erasure, we selected 100 target artistic styles following (Lu et al., 2024). We used the same remaining concepts in Section 4.1. We set $(s_1, s_2) = (16, 1)$ and $(\eta, \lambda) = (0.5, 1.0 \times 10^4)$, and we trained 16 residual adversarial embeddings for robustness to adversarial prompts over 10 stages. From Figure 3, all methods except FMN effectively erased the target concepts in the qualitative results. Notably, all baselines showed visual alterations or forgetting for remaining concepts in Figure 3. In contrast, CPE generated visually indistinguishable images of remaining concepts from the original images. Table 2 shows results for artistic style erasure that is consistent with those from celebrities erasure experiments. In this case, ESD-u achieved the lowest CS for target artistic styles, but resulted in degradation in the generation of remaining concepts. Meanwhile, CPE demonstrated competitive erasing effectiveness to ESD-u with the most minimal impact on remaining concepts.

## 4.3 EXPLICIT CONTENTS ERASURE

We generate images from I2P prompts (Schramowski et al., 2023) consisting of 4,703 ordinary prompts without inappropriate words which bypass to generate offensive contents. To erase explicit concepts, we removed four keywords following Lu et al. (2024): 'nudity', 'naked', 'erotic', and 'sexual'. To measure the frequency of explicit contents, we employed the NudeNet detector (Bedapudi, 2022), setting its detection threshold to 0.6 (Lu et al., 2024). For preservation performance, we used COCO-30K. For erasing stage, we used $(s_1, s_2) = (64, 4)$ and $(\eta, \lambda) = (3.0, 1.0 \times 10^4)$, and we trained 64 adversarial embeddings for adversarial learning stage over 20 stages. Table 3 shows the number of explicit contents detected by the NudeNet detector. CPE resulted in the fewest detected explicit contents, recording less than half of the result of the second-best method. In terms of FID, FMN and MACE showed better results, but FMN showed low efficacy on erasing explicit contents and MACE used COCO as anchors. In contrast, although we didn't use COCO-30K captions as anchors, CPE successfully maintained the concepts from them and achieved the best CS.

## 4.4 ROBUST ERASURE ON LEARNED ATTACK PROMPTS

We utilized Ring-A-Bell(Tsai et al., 2024) and UnlearnDiff(Zhang et al., 2023c) as the red-teaming tools to verify the robustness of our CPE. We considered Van Gogh and I2P prompts as the domains for the attack experiments. We also selected 6 celebrities among 50 celebrities used in Section 4.1 to evaluate robustness on celebrities erasure. We used attack success rate (ASR) as the evaluation metric to measure the ratio of regenerated images containing the target concepts by the red-teaming

Table 4: Comparison of CPE and baselines on robust concept erasure against adversarial attack prompts: Ring-A-Bell (Tsai et al., 2024) and UnlearnDiff (Zhang et al., 2023c).

| Method | 6 Celebrities | I2P Nudity | | Artistic Style (Vincent van Gogh) | |
|---|---|---|---|---|---|
| | Ring-A-Bell | Ring-A-Bell | UnlearnDiff | Ring-A-Bell | UnlearnDiff |
| FMN (Zhang et al., 2023a) | 58.06 | 80.85 | 97.89 | 14 | 56 |
| ESD (Gandikota et al., 2023) | 30.48 | 61.70 | 76.05 | 6 | 32 |
| UCE (Gandikota et al., 2024) | **0.00** | 35.46 | 79.58 | 2 | 94 |
| MACE (Lu et al., 2024) | 29.33 | 4.26 | 66.90 | 2 | 24 |
| RECE (Gong et al., 2024) | - | 13.38 | 65.46 | 28 | 64 |
| AdvUnlearn (Zhang et al., 2024) | - | - | **21.13** | - | 2 |
| **CPE (Ours)** | 1.14 | **0.00** | 30.28 | **0** | **0** |

Table 5: Ablation study on the effect of the components of ResAG. The row with **bold** represents the selected configurations. We can see all components are crucial for deletion and preservation.

| Components of CPE | | | | Target Concepts | | Remaining Concepts | | | | | | |
|---|---|---|---|---|---|---|---|---|---|---|---|---|
| | | | | 50 Celebrities | | 100 Celebrities | | | Artistic Styles | | COCO-1K | |
| ResAG | $\mathcal{L}_{att}$ | RARE | Rank ($s_1$) | CS ↓ | ACC ↓ | CS ↑ | ACC ↑ | KID(× 100) ↓ | CS ↑ | KID(× 100) ↓ | CS ↑ | KID(× 100)↓ |
| × | ✓ | ✓ | 16 | 22.38 | 9.83 | 34.01 | 83.96 | 0.34 | 25.64 | 1.83 | 29.56 | 0.23 |
| ✓ | × | ✓ | 16 | 18.32 | 0.07 | 30.53 | 72.54 | 0.44 | 28.12 | 0.12 | 30.83 | 0.09 |
| ✓ | ✓ | × | 16 | 21.75 | 2.23 | 34.83 | 88.31 | 0.08 | 29.02 | 0.01 | 31.27 | 0.05 |
| ✓ | ✓ | ✓ | **16** | **20.79** | **0.37** | **34.82** | **88.26** | **0.08** | **29.01** | **0.01** | **31.29** | **0.05** |
| ✓ | ✓ | ✓ | 1 | 21.34 | 0.74 | 34.43 | 87.63 | 0.10 | 28.81 | 0.03 | 31.06 | 0.07 |
| ✓ | ✓ | ✓ | 4 | 20.73 | 0.46 | 34.65 | 88.02 | 0.08 | 28.93 | 0.01 | 31.03 | 0.08 |
| ✓ | ✓ | ✓ | 64 | 20.76 | 0.31 | 34.81 | 88.41 | 0.07 | 29.03 | 0.01 | 31.27 | 0.04 |
| ✓ | ✓ | ✓ | 128 | 20.82 | 0.39 | 34.79 | 88.33 | 0.08 | 29.05 | 0.01 | 31.30 | 0.05 |
| SDv1.4 (Rombach et al., 2022b) | | | | 34.49 | 91.35 | 34.83 | 90.86 | - | 28.96 | - | 31.34 | - |

tools. From Table 4, CPE demonstrated robust erasure of target concepts competitive to recent robust methods (Gong et al., 2024; Zhang et al., 2024). In particular, CPE successfully defended attacks by Ring-A-Bell on explicit contents erasure, recording zero ASR. It also defended all attack prompts by Ring-A-Bell and UnlearnDiff on erasure of Van Gogh's artistic style, verifying its robustness.

## 4.5 ABLATION STUDY

For ablation study, we measured KID on COCO-1K which is randomly selected 1,000 captions from COCO dataset instead of FID on COCO-30K. For ablating ResAGs, we only trained $\mathbf{U}_3$ and $\mathbf{U}_4$. We studied on celebrities erasure with the same setup as in Section 4.1. From Table 5, the performance dropped when the components were removed, verifying their importance. Removing ResAGs led to significant degradation in both deletion and preservation. Training ResAG without $\mathcal{L}_{att}$ slightly improved erasure performance but severely impaired preservation capability on remaining concepts. Removing RARE demonstrated its role in enhancing target concept removal with minimal impact on the remaining concepts. We also measured performance variations based on the rank $s_1$ of the attention in ResAG. Even with $s_1 = 1$, CPE showed SOTA-level erasing performance compared to baselines, while maintaining strong protection of nearly all remaining concepts. Although increasing the rank from 1 to 16 improved performance, the enhancement was marginal for higher ranks. It suggests that the low-rank property is sufficient to capture the target concepts without affecting the remaining concepts. For additional ablation studies on $s_2$, $\eta$, $\lambda$, the architecture of ResAGs, and RARE in robustness evaluation, please refer to Appendix E.

## 5 CONCLUSION

In this work, we theoretically demonstrated that only fine-tuning CA layers for concept erasing in diffusion models could have challenge in preserving remaining concepts. As one solution for this, we proposed our framework, Concept Pinpoint Eraser (CPE), simple and effective approach to erase target concepts while maintaining diverse remaining concepts. We designed lightweight modules called the Residual Attention Gates (ResAGs), adaptively adjusting the CA layer outputs dependent on text embeddings. To robustly erase target concepts without forgetting on remaining concepts, ResAG is trained with an attention anchoring loss under an adversarial learning scheme. Through extensive experiments on erasure of celebrities, artistic styles, and explicit concepts, CPE ensures the robust deletion of target concepts and protection of diverse remaining concepts.

## ACKNOWLEDGMENTS

This work was supported in part by Institute of Information & communications Technology Planning & Evaluation (IITP) grant funded by the Korea government(MSIT) [NO.RS-2021-II211343, Artificial Intelligence Graduate School Program (Seoul National University)] and by the National Research Foundation of Korea(NRF) grant funded by the Korea government(MSIT) (No. NRF-2022R1A4A1030579, NRF-2022M3C1A309202211). Also, the authors acknowledged the financial support from the BK21 FOUR program of the Education and Research Program for Future ICT Pioneers, Seoul National University.

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

# A  ADDITIONAL PRELIMINARIES

## A.1  LATENT DIFFUSION MODEL AND STABLE DIFFUSION

The proposed method utilizes Stable Diffusion (SD) v1.4, (Rombach et al., 2022b), which is based on Latent Diffusion Models (LDM) (Rombach et al., 2022a). It performs diffusion within the latent space of an autoencoder, consists of two main components: a diffusion model (Dhariwal & Nichol, 2021; Ho et al., 2020; Song et al., 2020) and a vector quantization autoencoder (Van Den Oord et al., 2017). The autoencoder is pre-trained to encode an image $I$ into spatial latent codes with an encoder ($x = \mathcal{E}(I)$) and to reconstruct the image with a decoder ($\mathcal{D}(\mathcal{E}(I))$). Meanwhile, the diffusion model learns to produce latents that fit within the autoencoder's latent space. Especially, the objective of the T2I latent diffusion model for a text embedding $\mathbf{E}$ is:

$$\mathcal{L}_{\text{LDM}} = \mathbb{E}_{x \sim \mathcal{E}(\mathcal{I}), \mathbf{E}, \epsilon \sim \mathcal{N}(0,1), t} \left[ \|\epsilon - \epsilon_\theta(x_t, t, \mathbf{E})\|_2^2 \right]$$

where $x_t$ is the noisy latent at timestep $t$, $\epsilon$ is a sample from normal distribution, and $\epsilon_\theta$ is the denoising diffusion model parameterized by $\theta$. To encode a text prompt, a text embedding $\mathbf{E}$ encoded by a text encoder. To construct this structure of model, SD uitilizes a U-Net (Ronneberger et al., 2015) for the latent diffusion model and a CLIP (Radford et al., 2021) encoder for the text encoder.

## A.2  PRELIMINARIES ON PREVIOUS CONCEPT ERASING APPROACHES

We divide recent fine-tuning based methods into two groups: learning based (Gandikota et al., 2023; Zhang et al., 2024) and non-learning based (Gandikota et al., 2024; Lu et al., 2024) approaches.

**Learning-based approaches**  For erasing target concept, FMN (Zhang et al., 2023a) focuses on attention re-steering to minimize attention maps associated with target concepts without significantly deteriorating remaining concepts. It has shown that this intuitive approach can effectively remove harmful or biased contents with flexibility across various concepts and models.

ESD (Gandikota et al., 2023) is inspired by energy-based composition (Du et al., 2020; 2021). It aims to decrease the probability of generating an image based on the latent $x$ based on the likelihood of an embedding $\mathbf{E}_{\text{tar}}$ for the target concept, which is adjusted by a scaling factor $\eta$:

$$P_\theta(x) \propto \frac{P_{\theta^*}(x)}{P_{\theta^*}(\mathbf{E}_{\text{tar}}|x)^\eta}$$

where $P_{\theta^*}(x)$ represents the distribution generated by the original model. Then, it suggest to update the model with a gradient for $P_\theta(x)$:

$$\nabla \log P_{\theta^*}(x) - \eta(\nabla \log P_{\theta^*}(x|\mathbf{E}_{\text{tar}}) - \nabla \log P_{\theta^*}(x))$$

Finally, it updates the model $\epsilon$ by Tweedie's formula (Efron, 2011) for a noisy latent $x_t$:

$$\epsilon_\theta(x_t, \mathbf{E}_{\text{tar}}, t) \leftarrow \epsilon_{\theta^*}(x_t, t) - \eta \left[ \epsilon_{\theta^*}(x_t, \mathbf{E}_{\text{tar}}, t) - \epsilon_{\theta^*}(x_t, t) \right]$$

Especially it empirically verified that updating only CA layers (ESD-x) can effectively erase the target concept while maintaining the remaining concepts relatively well. Meanwhile, fine-tuning all parameters (ESD-u) in the model can holistically erase the image containing the target concept.

Based on ESD, AdvUnlearn (Zhang et al., 2024) proposed a method that is robust to adversarial attacks. Specifically, it seeks to find an embedding $\mathbf{E}'$ that minimizes the following objective:

$$\min_{\|\mathbf{E} - \mathbf{E}_{\text{tar}}\|_0 \leq \rho} \mathbb{E} \left[ \|\epsilon_\theta(\mathbf{x}_t|\mathbf{E}) - \epsilon_{\theta^*}(\mathbf{x}_t|\mathbf{E}_{\text{tar}})\|_2^2 \right]$$

Then, the unlearned model for the target concept is retrained with $\mathcal{L}_{\text{ESD}}$ using the adversarial text embedding. By repeating this process, the unlearned model becomes more robust to attack prompts.

**Non-learning based approaches**  As non-learning based approaches, UCE (Gandikota et al., 2024) and MACE (Lu et al., 2024), and RECE (Gong et al., 2024) use a closed-form solution. Briefly, they find the linear projections $\mathbf{W}'$ of keys and values in the cross-attention layers such that:

$$\mathbf{W}_{\text{new}} = \arg\min_{\mathbf{W}'} \sum_{n=1}^{N} \|\mathbf{W}'\mathbf{E}_{\text{tar}}^n - \mathbf{W}_{\text{old}}\mathbf{E}_{\text{sur}}^n\|_F^2 + \lambda \sum_{m=1}^{M} \|\mathbf{W}'\mathbf{E}_{\text{rem}}^m - \mathbf{W}_{\text{old}}\mathbf{E}_{\text{rem}}^m\|_F^2, \qquad \text{(A.1)}$$

where $\mathbf{W}_{\text{old}}$ is the original key/value projection. The terms in Equation (A.1) are the erasing and anchoring objective, respectively. It has a closed-form solution and compute $\mathbf{W}'$ directly.

MACE further ensures the erasure of the target concept by using SAM (Kirillov et al., 2023), to produce segmentation masks of generated images for the target concept. These masks are used to suppress the attention for each target concept within the CA map of the diffusion model. For training, LoRA modules are inserted in the CA layers of the model for each target concept. Then, the multiple LoRAs are integrated by a loss function, which also propose a term to retain the remaining concepts. This approach effectively prevents forgetting of concepts similar to the target concepts.

RECE (Gong et al., 2024) basically utilizes the closed-form solution proposed in UCE and enhances robustness against adversarial prompt attacks. RECE repeat to generate an adversarial prompt $\mathbf{E}'$ and erase for the adversarial prompt. For this, it generates the adversarial prompt by solving a closed-form objective for the prompt as follows:

$$\mathbf{E}' = \arg\min_{\mathbf{E}} \sum_l \|\mathbf{W}_{\text{new}}^l \mathbf{E} - \mathbf{W}_{\text{old}}^l \mathbf{E}_{\text{tar}}\|_2^2 + \lambda \|\mathbf{E}\|_2^2.$$

Then, for the attack prompt, $\mathbf{W}_{\text{old}}$ in Equation (A.1) is replaced with $\mathbf{W}_{\text{new}}$ and the weights are recomputed to block the attack prompt. By repeating this process, RECE effectively defends against adversarial prompt attacks. We note that while RECE finds new embeddings from scratch, our approach differs in that we learn residual embeddings to be added to the original target embeddings.

## B  PROOFS OF THEOREMS

We will use the notations and definitions in the main paper to prove Theorems 1, 2, and 3.

**Theorem 1.** *Let $\tilde{\mathbf{W}}_k^h$ and $\tilde{\mathbf{W}}_v^h$ be the updated weights of $\mathbf{W}_k^h$ and $\mathbf{W}_v^h$. Assume that $\|\mathbf{E}\|_2 \leq M_1$ and $\|\mathbf{z}\|_\infty \leq M_2$. Denoting the Lipschitz constant of the linear transforms $\mathbf{W}$ as $L_{\mathbf{W}}$, which is its spectral norm, the following holds:*

$$\|\tau(\mathbf{z}, \mathbf{E}; \tilde{\mathbf{W}}_k^h, \tilde{\mathbf{W}}_v^h) - \tau(\mathbf{z}, \mathbf{E}; \mathbf{W}_k^h, \mathbf{W}_v^h)\|_2 \leq \sum_{h=1}^{H} \left[ C_1^h \|\Delta \mathbf{W}_k^h \mathbf{E}\|_F + C_2^h \|\Delta \mathbf{W}_v^h \mathbf{E}\|_F \right], \quad \text{(B.1)}$$

*where $\Delta \mathbf{W} = \tilde{\mathbf{W}} - \mathbf{W}$, $C_1^h = \frac{M_1 M_2 \sqrt{m-1}}{m\sqrt{m}} L_{\mathbf{W}_o^h \mathbf{W}_v^h} L_{\mathbf{W}_q^h}$, and $C_2^h = L_{\mathbf{W}_o^h}$*

*Proof.* We denote $f_1(\mathbf{X}) = \mathbf{W}_o^h \mathbf{X}$ and $f_2(\mathbf{X}) = \sigma(\frac{(\mathbf{X})^T \mathbf{W}_q^h \mathbf{z}}{\sqrt{m}})$. Then, the difference in the output of the attention layer due to changes in $\mathbf{W}_k^i$ and $\mathbf{W}_v^i$ can be expressed as:

$$\|\tau(\mathbf{z}, \mathbf{E}; \tilde{\mathbf{W}}_k^h, \tilde{\mathbf{W}}_v^h) - \tau(\mathbf{z}, \mathbf{E}; \mathbf{W}_k^h, \mathbf{W}_v^h)\|_2$$

$$= \|\sum_{h=1}^{H} f_1(\tilde{\mathbf{W}}_v^h \mathbf{E}) f_2(\tilde{\mathbf{W}}_k^h \mathbf{E}) - \sum_{h=1}^{H} f_1(\mathbf{W}_v^h \mathbf{E}) f_2(\mathbf{W}_k^h \mathbf{E})\|_2$$

$$\leq \sum_{h=1}^{H} \|f_1(\tilde{\mathbf{W}}_v^h \mathbf{E}) f_2(\tilde{\mathbf{W}}_k^h \mathbf{E}) - f_1(\mathbf{W}_v^h \mathbf{E}) f_2(\mathbf{W}_k^h \mathbf{E})\|_2$$

$$\leq \sum_{h=1}^{H} \left[ \|(f_1(\tilde{\mathbf{W}}_v^h \mathbf{E}) - f_1(\mathbf{W}_v^h \mathbf{E})) f_2(\tilde{\mathbf{W}}_k^h \mathbf{E}) + f_1(\mathbf{W}_v^h \mathbf{E})(f_2(\tilde{\mathbf{W}}_k^h \mathbf{E}) - f_2(\mathbf{W}_k^h \mathbf{E}))\|_2 \right]$$

$$\leq \sum_{h=1}^{H} \left[ \|f_2(\tilde{\mathbf{W}}_k^h \mathbf{E})\|_\infty \|f_1(\tilde{\mathbf{W}}_v^h \mathbf{E}) - f_1(\mathbf{W}_v^h \mathbf{E})\|_2 + \|f_1(\mathbf{W}_v^h \mathbf{E})\|_2 \|f_2(\tilde{\mathbf{W}}_k^h \mathbf{E}) - f_2(\mathbf{W}_k^h \mathbf{E})\|_2 \right]$$

We note that $\|f_2(\tilde{\mathbf{W}}_k^h \mathbf{E})\|_\infty \leq 1$ because of the maximum value of softmax function is 1 and $\|f_1(\mathbf{W}_v^h \mathbf{E})\|_2 \leq L_{\mathbf{W}_o^h \mathbf{W}_v^h} \|\mathbf{E}\|_2 \leq L_{\mathbf{W}_o^i \mathbf{W}_v^i} M_1$. We can also derive that

$$\|f_1(\tilde{\mathbf{W}}_v^h \mathbf{E}) - f_1(\mathbf{W}_v^h \mathbf{E})\|_2 \leq L_{\mathbf{W}_o^h} \|\Delta \mathbf{W}_v^h \mathbf{E}\|_F$$

and

$$\|f_2(\tilde{\mathbf{W}}_k^h \mathbf{E}) - f_2(\mathbf{W}_k^h \mathbf{E})\|_2 = \|\sigma\left(\frac{(\tilde{\mathbf{W}}_k^h \mathbf{E})^T \mathbf{W}_q^h \mathbf{z}}{\sqrt{m}}\right) - \sigma\left(\frac{(\mathbf{W}_k^h \mathbf{E})^T \mathbf{W}_q^h \mathbf{z}}{\sqrt{m}}\right)\|_2 \quad \text{(B.2)}$$

$$\leq \frac{\sqrt{m-1}}{m\sqrt{m}}\|(\tilde{\mathbf{W}}_k^h \mathbf{E})^T \mathbf{W}_q^h \mathbf{z} - (\mathbf{W}_k^h \mathbf{E})^T \mathbf{W}_q^i \mathbf{z}\|_2$$

$$\leq \frac{\sqrt{m-1}}{m\sqrt{m}} L_{\mathbf{W}_q^h}\|\mathbf{z}\|_\infty \|\tilde{\mathbf{W}}_k^h \mathbf{E} - \mathbf{W}_k^h \mathbf{E}\|_2$$

$$\leq \frac{M_2\sqrt{m-1}}{m\sqrt{m}} L_{\mathbf{W}_q^h}\|\Delta\mathbf{W}_k^h \mathbf{E}\|_F,$$

where we use the fact that Lipschitz constant of the softmax function is $\frac{\sqrt{m-1}}{m}$ (Sener & Savarese, 2018) for the first inequality, and the spectral norm is equal to or smaller than the Frobenius norm for the last inequality in Equation (B.2) . Finally we can conclude that the upper bound of the output of an attention layer due to the difference of key and value weights is:

$$\|\tau(\mathbf{z}, \mathbf{E}; \tilde{\mathbf{W}}_k^h, \tilde{\mathbf{W}}_v^h) - \tau(\mathbf{z}, \mathbf{E}; \mathbf{W}_k^h, \mathbf{W}_v^h)\|_2 \leq \sum_{h=1}^{H} \left[C_1^h\|\Delta\mathbf{W}_k^h \mathbf{E}\|_F + C_2^h\|\Delta\mathbf{W}_v^h \mathbf{E}\|_F\right],$$

where $C_1^h = \frac{M_1 M_2 \sqrt{m-1}}{m\sqrt{m}} L_{\mathbf{W}_o^h \mathbf{W}_v^h} L_{\mathbf{W}_q^h}$ and $C_2^h = L_{\mathbf{W}_o^h}$. $\qquad\square$

**Theorem 2.** *Let* $\mathbf{E}_{\text{rem}} = [\mathbf{e}_1, \cdots, \mathbf{e}_i, \cdots, \mathbf{e}_m]$ *where* $\mathbf{e}_i$ *is a text embedding. Suppose that the embeddings for remaining concepts follow a mixed Gaussian distribution, that is,* $p(\mathbf{e}_i) = \sum_{r=1}^{R} \pi_r \mathcal{N}(\mathbf{e}_i; \boldsymbol{\mu}_r^i, \sigma_r^2 \mathbf{I})$ *and* $\sum_{r=1}^{R} \pi_r = 1$. *With* $\boldsymbol{\mu}_r = [\boldsymbol{\mu}_r^1, \boldsymbol{\mu}_r^2, \cdots, \boldsymbol{\mu}_r^m]$, *we can show that:*

$$\mathbb{E}_{\mathbf{E}_{\text{rem}}}\left[\|\Delta\mathbf{W}\mathbf{E}_{\text{rem}}\|_F^2\right] = C_3\|\Delta\mathbf{W}\|_F^2 + \sum_{r=1}^{R} \pi_r\|\Delta\mathbf{W}\boldsymbol{\mu}_r\|_F^2, \quad C_3 = \sum_{r=1}^{R} \pi_r\sigma_r^2.$$

*Proof.* Assume that $\mathbf{e}_i$ is sampled from a mixed Gaussian distribution. Then,

$$\mathbb{E}_{\mathbf{E}_{\text{rem}}}\left[\|\Delta\mathbf{W}\mathbf{E}_{\text{rem}}\|_F^2\right] = \mathbb{E}_{\mathbf{E}_{\text{rem}}}\left[\|\Delta\mathbf{W}[\mathbf{e}_1, \mathbf{e}_2, \cdots, \mathbf{e}_m]\|_F^2\right] \quad \text{(B.3)}$$

$$= \mathbb{E}_{\mathbf{E}_{\text{rem}}}\left[\|[\Delta\mathbf{W}\mathbf{e}_1, \Delta\mathbf{W}\mathbf{e}_2, \cdots, \Delta\mathbf{W}\mathbf{e}_m]\|_F^2\right]$$

$$= \mathbb{E}_{\mathbf{E}_{\text{rem}}}\left[\sum_{i=1}^{m}\|\Delta\mathbf{W}\mathbf{e}_i\|_2^2\right]$$

$$= \sum_{i=1}^{m}\mathbb{E}_{\mathbf{e}_i}\left[\mathbf{e}_i^T\Delta\mathbf{W}^T\Delta\mathbf{W}\mathbf{e}_i\right]$$

$$= \sum_{i=1}^{m}\pi_r\sum_{r=1}^{R}\mathbb{E}_{\mathbf{e}_i\sim\mathcal{N}(\boldsymbol{\mu}_r^i, \sigma_r^2\mathbf{I})}\left[\mathbf{e}_i^T\Delta\mathbf{W}^T\Delta\mathbf{W}\mathbf{e}_i\right] \quad \text{(B.4)}$$

It is known that the expectation of quadratic form $\mathbf{e}_i^T\mathbf{B}\mathbf{e}_i$ for a symmetric matrix $\mathbf{B}$ is given (Mathai & Provost, 1992) by:

$$\mathbb{E}_{\mathbf{e}_i\sim\mathcal{N}(\boldsymbol{\mu}_r^i, \sigma_r^2\mathbf{I})}\left[\mathbf{e}_i^T\mathbf{B}\mathbf{e}_i\right] = \sigma_r^2\text{tr}(\mathbf{B}) + (\boldsymbol{\mu}_r^i)^T\mathbf{B}\boldsymbol{\mu}_r^i. \quad \text{(B.5)}$$

Because of the assumption of mixture Gaussian model, we can easily see that from Equation (B.4):

$$\mathbb{E}_{\mathbf{E}}\left[\mathbf{e}_i^T\Delta\mathbf{W}^T\Delta\mathbf{W}\mathbf{e}_i\right] = \text{tr}(\Delta\mathbf{W}^T\Delta\mathbf{W})\sum_{r=1}^{R}\pi_r\sigma_r^2 + \sum_{r=1}^{R}\pi_r(\boldsymbol{\mu}_r^i)^T\Delta\mathbf{W}^T\Delta\mathbf{W}\boldsymbol{\mu}_r^i. \quad \text{(B.6)}$$

By substituting Equation (B.6) for Equation (B.4), it is also straightforward that:

$$
\mathbb{E}_{\mathbf{E}_{\text{rem}}}\left[\|\Delta\mathbf{W}\mathbf{E}_{\text{rem}}\|_F^2\right] = \sum_{i=1}^{m}\mathbb{E}_{\mathbf{E}_{\text{rem}}}\left[\mathbf{e}_i^T\Delta\mathbf{W}^T\Delta\mathbf{W}\mathbf{e}_i\right]
$$

$$
= \text{tr}(\Delta\mathbf{W}^T\Delta\mathbf{W})\sum_{r=1}^{R}\pi_r\sigma_r^2 + \sum_{r=1}^{R}\pi_r\sum_{i=1}^{m}(\boldsymbol{\mu}_r^i)^T\Delta\mathbf{W}^T\Delta\mathbf{W}\boldsymbol{\mu}_r^i
$$

$$
= C_3\|\Delta\mathbf{W}\|_F^2 + \sum_{r=1}^{R}\pi_r\|\Delta\mathbf{W}\boldsymbol{\mu}_r\|_F^2
$$

where $C_3 = m\sum_{r=1}^{R}\pi_r\sigma_r^2$ and $\text{tr}(\Delta\mathbf{W}^T\Delta\mathbf{W})$ is same as the Frobenius norm. $\qquad\square$

**Corollary 1.** *Suppose we can detect the mode from which $\mathbf{E}$ is sampled. Let $f(\mathbf{E}) = \mathbf{V}_r \in \mathbb{R}^{m\times m}$ be an embedding-dependent projection adaptive to the mode of samples. If we use $\Delta\mathbf{W}\mathbf{E}f(\mathbf{E})$ instead of $\Delta\mathbf{W}\mathbf{E}$, Equation (3) for $\mathbf{E}_{\text{rem}}$ is modified to:*

$$
\mathbb{E}_{\mathbf{E}_{\text{rem}}}\left[\|\Delta\mathbf{W}\mathbf{E}_{\text{rem}}f(\mathbf{E}_{\text{rem}})\|_F^2\right] = \|\Delta\mathbf{W}\|_F^2\sum_{r=1}^{R}\pi_r\sigma_r^2\|\mathbf{V}_r\|_F^2 + \sum_{r=1}^{R}\pi_r\|\Delta\mathbf{W}\boldsymbol{\mu}_r\mathbf{V}_r\|_F^2. \qquad \text{(B.7)}
$$

*Proof.* With the assumption in Theorem. 2 and simliar derivation from Equation (B.3)-(B.4),

$$
\mathbb{E}_{\mathbf{E}_{\text{rem}}}\left[\|\Delta\mathbf{W}\mathbf{E}_{\text{rem}}f(\mathbf{E}_{\text{rem}})\|_F^2\right] = \sum_{r=1}^{R}\pi_r\sum_{i=1}^{m}\mathbb{E}_{\mathbf{E}_{\text{rem}}\sim\mathcal{N}(\boldsymbol{\mu}_r,\sigma_r^2 I)}\left[(\mathbf{E}_{\text{rem}}\mathbf{V}_r^i)^T\Delta\mathbf{W}^T\Delta\mathbf{W}\mathbf{E}_{\text{rem}}\mathbf{V}_r^i\right],
$$

where $\mathbf{V}_r^i$ is the $i$-th column of $\mathbf{V}_r$. Since the distribution of $\mathbf{E}_{\text{rem}}\mathbf{V}_r^i$ is $\mathcal{N}(\boldsymbol{\mu}_r V_r^i, \sigma_r^2\|\mathbf{V}_r^i\|_2^2 I)$,

$$
\sum_{i=1}^{m}\mathbb{E}_{\mathbf{E}_{\text{rem}}\sim\mathcal{N}(\boldsymbol{\mu}_r,\sigma_r^2 I)}\left[(\mathbf{E}_{\text{rem}}\mathbf{V}_r^i)^T\Delta\mathbf{W}^T\Delta\mathbf{W}\mathbf{E}_{\text{rem}}\mathbf{V}_r^i\right] = \sigma_r^2\|\mathbf{V}_r\|_F^2\|\Delta\mathbf{W}\|_F^2 + \|\Delta\mathbf{W}\boldsymbol{\mu}_r\mathbf{V}_r\|_F^2
$$

$$\text{(B.8)}$$

by applying Equation (B.5). Therefore, considering the Gaussian mixture model gives us Equation (B.7) by summing the expectation computed by Equation (B.8) across the modes. $\qquad\square$

## C  FURTHER DISCUSSIONS ON CPE

### C.1  SAMPLING METHOD FOR ANCHORING CONCEPTS

**Augmentation on anchor concepts.** We found that augmenting anchor concepts can better protect the remaining concepts. We applied noise perturbation to the text embeddings of anchor concepts and utilized Mixup (Zhang et al., 2017; Lim et al., 2022), which has already shown to enhance generalization performance and mitigate over-fitting with limited training samples. Specifically, we first perturb text embeddings $(\mathbf{E}_{\text{anc}}^1, \mathbf{E}_{\text{anc}}^2)$ of a pair of anchors with Gaussian additive noise:

$$
\tilde{\mathbf{E}}_{\text{anc}}^i = \mathbf{E}_{\text{anc}}^i + \delta\xi, \quad \xi \sim \mathcal{N}(0, \mathbf{I})
$$

where $i = 1, 2$ and $\delta$ determines the noise scale. Then, we obtain the final anchor embedding for the attention anchoring loss by performing linear interpolation on the two perturbed text embeddings:

$$
\tilde{\mathbf{E}}_{\text{anc}} = \zeta\tilde{\mathbf{E}}_{\text{anc}}^1 + (1-\zeta)\tilde{\mathbf{E}}_{\text{anc}}^2,
$$

where $\zeta \sim Beta(\alpha,\beta)$, $Beta(\cdot,\cdot)$ is the beta distribution and we set $\alpha = \beta = 1.0$ following the default setup from Zhang et al. (2017). Table C.1 presents the effect of noise perturbation and Mixup for text embeddings on the erasing effectiveness on target concepts and the preservation performance on remaining concepts. We first note that even without using noise injection and Mixup, CPE already outperformed to the baselines in terms of both erasing effectiveness and preservation performance. Nevertheless, we observed that applying noise perturbation to the original anchor concepts helps better preserve the remaining concepts without impairing the erasing effectiveness. Furthermore we could protect the remaining concepts even more effectively by applying Mixup.

Table C.1: Studies on the effect of noise injection and Mixup on anchor concepts. The row in **bold** represents the results of the selected configurations for comparison to baselines.

| | | Target Concepts | | Remaining Concepts | | | | | | |
| | | 50 Celebrities | | 100 Celebrities | | | Artistic Styles | | COCO-1K | |
| Mixup | Noise Scale | CS ↓ | ACC ↓ | CS ↑ | ACC ↑ | KID(× 100) ↓ | CS ↑ | KID(× 100) ↓ | CS ↑ | KID(× 100)↓ |
|---|---|---|---|---|---|---|---|---|---|---|
| × | $0.0$ | 20.46 | 0.37 | 34.47 | 86.73 | 0.19 | 28.39 | 0.07 | 30.84 | 0.11 |
| × | $1.0 \times 10^{-3}$ | 20.68 | 0.31 | 34.52 | 87.03 | 0.16 | 28.56 | 0.05 | 31.01 | 0.09 |
| ✓ | $1.0 \times 10^{-5}$ | 20.71 | 0.37 | 34.45 | 87.19 | 0.16 | 28.41 | 0.09 | 31.15 | 0.07 |
| ✓ | $1.0 \times 10^{-4}$ | 20.64 | 0.42 | 34.64 | 87.52 | 0.13 | 28.75 | 0.03 | 31.08 | 0.08 |
| ✓ | **$1.0 \times 10^{-3}$** | **20.79** | **0.37** | **34.82** | **88.26** | **0.08** | **29.01** | **0.01** | **31.29** | **0.05** |
| ✓ | $1 \times 10^{-2}$ | 23.48 | 23.56 | 34.89 | 89.53 | 0.02 | 28.99 | 0.00 | 31.36 | 0.02 |
| ✓ | $1.0 \times 10^{-1}$ | 31.36 | 50.39 | 34.84 | 90.42 | 0.01 | 28.95 | 0.00 | 31.33 | 0.01 |
| SDv1.4 (Rombach et al., 2022b) | | 34.49 | 91.35 | 34.83 | 90.86 | - | 28.96 | - | 31.34 | - |

**Selection of anchor concepts.** We also investigated the effect of selection of anchor concepts for better protecting the remaining concepts. Specifically, for celebrities erasure and artistic styles erasure, we followed previous works (Gandikota et al., 2023; 2024; Lu et al., 2024; Lyu et al., 2024; Gong et al., 2024) and selected concepts similar to the target concept for anchors. We also considered similar expressions to the target domain to erase. For example, when erasing a target celebrity in celebrities erasure, we selected 50 similar celebrities from a pre-defined anchor concept pool and 100 expressions semantically related to the word "celebrities" as anchors. More detailed configurations on the selected anchor concepts can be found in Appendix D.3. Table C.2 shows the performance depending on the selection of anchor concepts for celebrities erasure. We observed that using both similar celebrities to the target and expressions similar to "celebrities" as anchors showed the best performance. Although other selections of anchors with our CPE still outperformed the baselines for almost cases, preserving the remaining celebrities degraded when only similar expressions were used. Employing only similar celebrities also led to a slight degradation in erasing performance. When only COCO-30K captions were selected as anchors, we observed degradation in the preservation of remaining celebrities.

Table C.2: Studies on the effect of selection of anchor concepts. The row in **bold** represents the results of the selected configurations for comparison to baselines.

| | Target Concepts | | Remaining Concepts | | | | | | |
| | 50 Celebrities | | 100 Celebrities | | | Artistic Styles | | COCO-1K | |
| Anchor Concepts | CS ↓ | ACC ↓ | CS ↑ | ACC ↑ | KID(× 100) ↓ | CS ↑ | KID(× 100) ↓ | CS ↑ | KID(× 100)↓ |
|---|---|---|---|---|---|---|---|---|---|
| **Similar Celebrities & Similar Expressions** | **20.79** | **0.37** | **34.82** | **88.26** | **0.08** | **29.01** | **0.01** | **31.29** | **0.05** |
| Only Similar Celebrities | 21.64 | 0.63 | 34.63 | 87.75 | 0.16 | 28.89 | 0.01 | 31.27 | 0.05 |
| Only Similar Expressions | 20.58 | 0.31 | 33.53 | 83.54 | 0.22 | 28.85 | 0.01 | 31.25 | 0.03 |
| COCO-30K Captions | 20.35 | 0.26 | 32.75 | 81.56 | 0.26 | 28.97 | 0.00 | 31.32 | 0.01 |
| SDv1.4 (Rombach et al., 2022b) | 34.49 | 91.35 | 34.83 | 90.86 | - | 28.96 | - | 31.34 | - |

## C.2 BALANCING BETWEEN LOSS OF KEY AND VALUE PROJECTIONS

The attention anchoring loss $\mathcal{L}_{att}$ requires different coefficients $C_1^{l,h}$ and $C_2^{l,h}$ for keys and values from Equation (8). We did not treat these values as hyper-parameters but instead computed them from the model parameters. For simplicity, we computed $C_1^l = \max_h C_1^{l,h}$ and $C_2^l = \max_h C_2^{l,h}$ for each layer and used it as the coefficient. The Lipschitz constants $L_{\mathbf{W}_o^h}$, $L_{\mathbf{W}_o^h \mathbf{W}_v^h}$, and $L_{\mathbf{W}_q^h}$ of the linear projections in the CA layers are precomputed before training, which is computationally negligible. Additionally, we precomputed the maximum values $M_1$ and $M_2$ of $\|E\|_2$ and $\|z\|_\infty$ in Theorem 1.

Figure C.1 shows values of $C_1^l$ and $C_2^l$ for each CA layer in the U-Net model of SD v1.4 for each iteration when we train for erasing 50 celebs. In this case, we directly computed $\|E\|_2$ and $\|z\|_\infty$ to validate the use of $M$ and $N$. We can see that the values of $C_1^l$ and $C_2^l$ for each layer do not change significantly when concepts are randomly sampled at each iteration. Particularly, $C_1^l$ is much larger than $C_2^l$, which is influenced by $\|E\|_2$ and $\|z\|_\infty$. Additionally, it is observed that the closer to the mid-block (8-th and 9-th layer), the larger the value of $C_1^l$, and the farther away, the smaller the value. This implies that greater weight is given to changes in linear projections for keys rather than values, and greater weight to deeper levels of linear projections such as the mid-block.

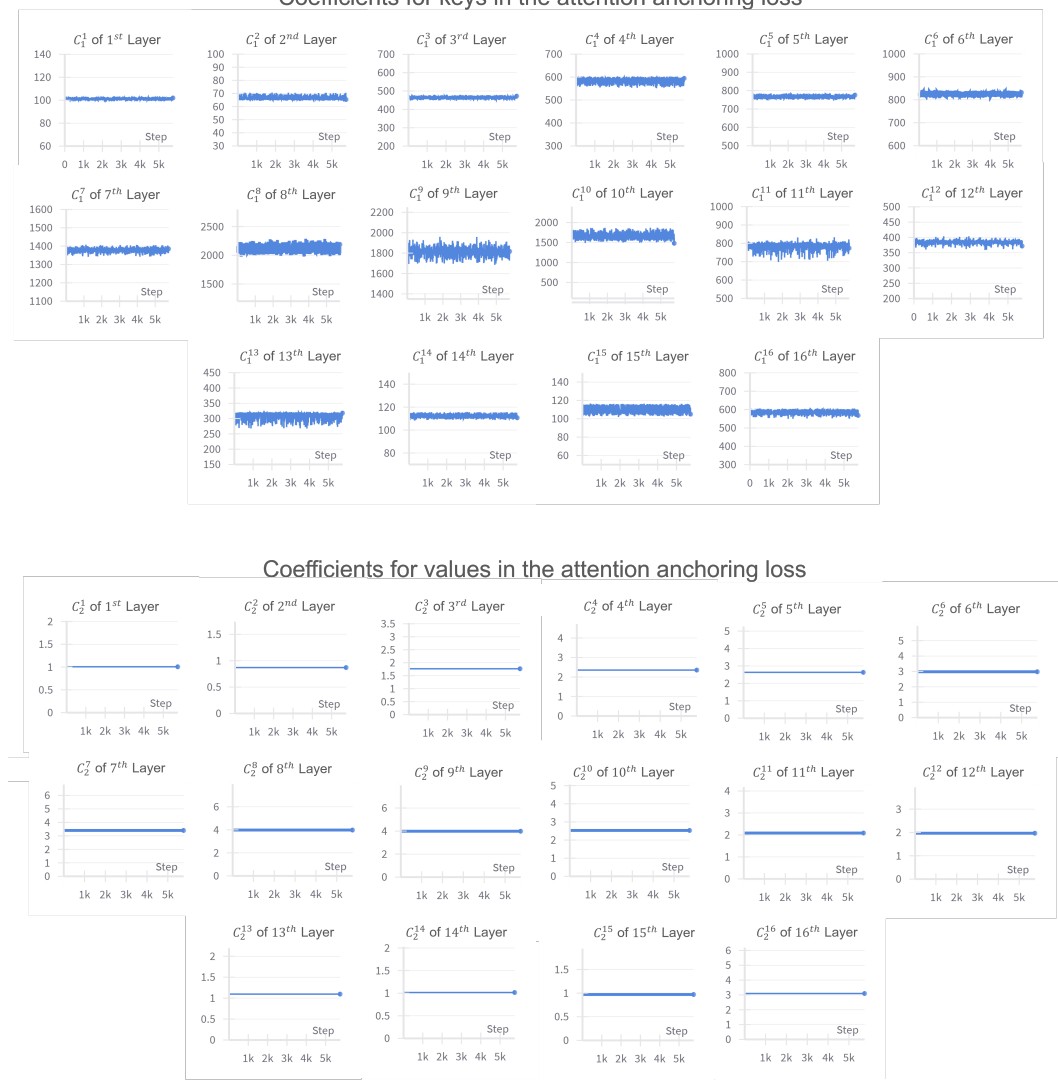

Figure C.1: Values of $C_1^l$ and $C_2^l$ for each layer in the diffusion model during erasing 50 celebs simultaneously. This shows that the $C_1^l$ values used during training do not vary significantly across samples, which verifies their stability. Additionally, it is observed that the coefficients $C_1^l$ for the key are much larger than the coefficients $C_2^l$ for the value. Furthermore, there is a tendency for the values of the coefficients to increase as they approach the mid-block layers (8-th and 9-th layers).

## C.3   EFFICIENCY STUDY

**Memory Consumption.**   ResAG in CPE is extremely lightweight since it consists of two pairs of low-rank decomposed matrices $(\mathbf{U}_1^T, \mathbf{U}_1)$ and $(\mathbf{U}_3^T, \mathbf{U}_4)$, with ranks $s_1$ and $s_2$, respectively. Notably, $\mathbf{U}_1$ and $\mathbf{U}_2$ is shared across all projections within the CA-layers, while $\mathbf{U}_3^T$ and $\mathbf{U}_4$ exists for each projection. Furthermore, $\mathbf{U}_3^T$ and $\mathbf{U}_4$ are sufficient with a rank of just one to perform successfully in most cases (please see Table E.1). Table C.3 shows the ranks of the learnable matrices within the ResAG for each erasing task and their memory consumption. In the tasks of celebrities erasure and artistic styles erasure, the additional parameters required for each target concept account for less than 0.01% of the total parameters in SD v1.4. Even in the case of multiple concept erasure, where 100 concepts are erased in artistic styles erasure, we only need additional parameters less than 1% of the total parameters of SD v1.4. For explicit content erasure, which involves removing more implicit concepts, we increased the rank of the learnable matrices in the ResAG by a factor of

four. Since we erased four target concepts, only around 0.14% of additional parameters are added compared to the total parameters of SD v1.4. This demonstrates that CPE is highly memory-efficient.

Table C.3: Memory consumption of CPE on celebrities erasure. The number of parameters are decided by rank $s_1$, $s_2$ and number of target concepts. It demonstrates the memory efficiency of CPE.

|  | Celebrities Erasure | Artistic Styles Erasure | Explicit Contents Erasure |
|---|---|---|---|
| Rank ($s_1$) | 16 | 16 | 64 |
| Rank ($s_2$) | 1 | 1 | 4 |
| # of Target Concepts | 50 | 100 | 4 |
| # of Params (per concept) | 75K | 75K | 295K |
| Param Ratio to SD v1.4 (per concept ) | $< 0.01\%$ | $< 0.01\%$ | $< 0.04\%$ |
| # of Params (in total) | 3.75M | 7.50M | 1.18M |
| Param Ratio to SD v1.4 (in total) | 0.45% | 0.90% | 0.14% |

**Computation Cost.** CPE directly computes the loss on the projection output of CA layers. It means that it doesn't require to compute the output in end-to-end manner with the entire diffusion model. Therefore, CPE can quickly compute the loss right after passing the text embeddings through the CA layers, allowing for very fast training. If multi-stage training for robustness against adversarial attacks through RARE is not performed, each target concept can be erased in under 2 minutes. With RARE, the multi-stage iterative training for robustness, the training time generally increases in proportion to the number of repetitions of erasing and adversarial embeddings learning. Table C.4 shows the computation cost in A6000 GPU hours to erase 50 celebrities with CPE or baselines. In the celebrities erasure, we repeated the process of erasing and adversarial embeddings learning five times for RARE. The results show that if RARE is not conducted, CPE can be trained very quickly. Furthermore, even with multi-stage training, the training time is in practical level.

Table C.4: Computation cost of CPE and baselines on celebrities erasure in A6000 GPU hours. It shows that CPE is practically applicable for erasing 50 celebrities, and becomes much faster without RARE, the training strategy for robustness to adversarial embeddings learning.

| Method | Data Prep. Time(h) | Fine-Tuning (h) | Total Time (h) |
|---|---|---|---|
| FMN | 0.8h | 0.5h | 1.3h |
| ESD | 0 | 4h | 4h |
| UCE | 0 | 0.1h | 0.1h |
| MACE | 1h (except COCO-30K captions) | 1h | 2h |
| **CPE (Ours, without RARE)** | 0 | 1.7h (2 min. per concept) | 1.7h |
| **CPE (Ours)** | 0 | 6h (7 min. per concept) | 6h |

### C.4 EXPLICIT CONTENTS ERASURE ONLY BY ONE TARGET CONCEPT

For explicit contents erasure, CPE erased four target concepts similar to MACE: "nudity," "naked," "erotic," and "sexual." However, UCE, ESD, and RECE all target the erasure of only one concept, "nudity." For a fair comparison, we also used CPE to erase only the single target concept, "nudity," and compared it with the baselines. From Table C.5, CPE (four words) refers to the results of erasing four target concepts, as shown in Table 3, while CPE (one word) represents the results of erasing only the target concept "nudity". We can see even when erasing only one target concept, CPE still achieved the lowest number of detected explicit contents while effectively preserving COCO-30K.

### D IMPLEMENTATION DETAILS

#### D.1 EVALUATION PROTOCOLS

We measured the **CLIP Score (CS)** (Hessel et al., 2021), which assesses the similarity between an image and a prompt based on the CLIP (Radford et al., 2021). The cosine similarity is computed between them after transforming both the generated image $I$ and the text prompt $p$ into embeddings:

$$\text{CS}(I, p) = \max(100 \cdot \cos(\mathbf{E}_I, \mathbf{E}_p), 0) \tag{D.1}$$

where $\mathbf{E}_I$ is the image embedding from CLIP's image encoder and $\mathbf{E}_p$ is the text embedding from its text encoder. The score ranges from 0 to 100, with lower scores indicating better concept erasure, and higher scores indicating better retention of remaining concepts.

Table C.5: Results of detected number of explicit contents using NudeNet detector on I2P. We considered two models for CPE: one with four target concepts erased ("nudity", "naked", "erotic", "sexual") and the other with one target concept erased ("nudity")

| Method | Number of nudity detected on I2P (Detected Quantity) | | | | | | | | | COCO-30K | |
| | Armpits | Belly | Buttocks | Feet | Breasts (F) | Genitalia (F) | Breasts (M) | Genitalia (M) | Total | CS ↑ | FID ↓ |
|---|---|---|---|---|---|---|---|---|---|---|---|
| FMN (Zhang et al., 2023a) | 43 | 117 | 12 | 59 | 155 | 17 | 19 | 2 | 424 | 30.39 | 13.52 |
| ESD-x (Gandikota et al., 2023) | 59 | 73 | 12 | 39 | 100 | 4 | 30 | 8 | 315 | 30.69 | 14.41 |
| ESD-u (Gandikota et al., 2023) | 32 | 30 | **2** | 19 | 35 | 3 | 9 | 2 | 123 | 30.21 | 15.10 |
| UCE (Gandikota et al., 2024) | 29 | 62 | 7 | 29 | 35 | 5 | 11 | 4 | 182 | 30.85 | 14.07 |
| MACE (Lu et al., 2024) | 17 | 19 | **2** | 39 | 16 | **0** | 9 | 7 | 111 | 29.41 | 13.42 |
| RECE (Gong et al., 2024) | 31 | 25 | 3 | **8** | 10 | **0** | 9 | 3 | 89 | 30.95 | - |
| **CPE (Four words)** | **10** | **8** | **2** | **8** | **6** | 1 | **3** | **2** | **40** | 31.19 | 13.89 |
| **CPE (One word)** | 15 | 13 | 5 | 15 | 11 | 2 | **3** | **2** | 65 | **31.30** | **12.88** |
| SD v1.4 (Rombach et al., 2022b) | 148 | 170 | 29 | 63 | 266 | 18 | 42 | 7 | 743 | 31.34 | 14.04 |
| SD v2.1 (Rombach, 2022) | 105 | 159 | 17 | 60 | 177 | 9 | 57 | 2 | 586 | 31.53 | 14.87 |

For assessing the degree of change in the remaining concepts, we used the **Frechet Inception Distance (FID)** (Heusel et al., 2017) and **Kernel Inception Distance (KID)** (Sutherland et al., 2018). FID evaluates the distance between the distributions of real and generated images. It calculates the Wasserstein-2 distance between feature vectors extracted using a pre-trained Inception network, with lower FID scores indicating greater similarity between generated and real images:

$$\text{FID}(\mathbf{r}, \mathbf{g}) = \|\mu_r - \mu_g\|^2 + \text{Tr}\left(\Sigma_r + \Sigma_g - 2\sqrt{\Sigma_r\Sigma_g}\right),$$

which quantifies the distance between the means and covariances of the feature distributions of real images ($\mu_r, \Sigma_r$) and generated images ($\mu_g, \Sigma_g$) by combining the squared Euclidean distance between their means with a term that accounts for the difference in their covariance structures.

KID is another metric for evaluating the quality of generated images. It computes the squared Maximum Mean Discrepancy (MMD) between feature representations:

$$\text{MMD}(p, q) = \mathbb{E}_{x,x'\sim p}[K(x, x')] + \mathbb{E}_{y,y'\sim q}[K(y, y')] - 2\mathbb{E}_{x\sim p, y\sim q}[K(x, y')],$$

where $p$ and $q$ represent the two probability distributions being compared. $x$ and $x'$ are random variables sampled from distribution $p$, and their similarity is measured using the kernel function $K(x, x')$ to compute the expected value of the samples within $p$. Similarly, $y$ and $y'$ are variables sampled from distribution $q$, and their similarity is assessed in the same manner. Lastly, the term $-2\mathbb{E}_{x\sim p, y\sim q}[K(x, y')]$ measures the similarity between variables sampled from the different distributions $p$ and $q$, reflecting the difference between the two distributions. KID has the advantage of being unbiased, meaning its expected value accurately reflects the true distance between distributions, even with small sample sizes, making it especially reliable for generative model evaluation.

## D.2 DETAILS ON EXPERIMENTAL SETUPS

To compare a wide range of remaining concepts, we evaluated the preservation performance on 100 celebrities in Table D.2, 100 artistic styles in Table D.5, 64 characters in Table D.8, and the COCO-30K dataset. For tasks other than COCO-30K, we opted for KID over FID, since FID could be unstable and biased for small number of samples in the dataset. Except for COCO-30K where original images exist for the text prompts, we used the images generated by Stable Diffusion v1.4 as the ground truth for evaluation.

**Celebrities erasure.** We erased 50 celebrities in this task and utilized the top-1 accuracy of the GIPHY Celebrity Detector (GCD) to specifically evaluate the effectiveness of celebrity concept erasure and preservation as an additional metric. We only used images generated by Stable Diffusion v1.4 where the GCD achieved 99% accuracy for both the training and test sets. It ensures that low detection accuracy by the GCD would not affect the evaluation. The 50 target celebrities are listed in Table D.1 and remaining concepts are listed in Table D.2. To generate their images, we used 5 prompt templates with 5 random seeds (1-5). The prompt templates are distinct for celebrities and artistic styles. We used 0 as a seed generating 5 images from a prompt for characters. The prompt templates are listed in Table D.7.

In the case of celebrities erasure, we set the following negative prompts to improve image quality:

*"bad anatomy, watermark, extra digit, signature, worst quality, jpeg artifacts, normal quality, low quality, long neck, lowres, error, blurry, missing fingers, fewer digits, missing arms, text, cropped, humpbacked, bad hands, username"*

**Artistic styles erasure.** We erased 100 artistic styles in this task and CS was utilized to measure the effectiveness of erasure. The 100 target artistic styles are listed in Table D.4 and the remaining concepts are listed in Table D.5. To generate their images, we used 5 prompt templates with 5 random seeds (1-5). The prompt templates are distinct for celebrities and artistic styles. For characters, we set 0 as a seed and generated 5 images from a prompt. The prompt templates are listed in Table D.7

**Explicit contents erasure.** We evaluated explicit contents erasure on Inappropriate Image Prompts (I2P) and used the NudeNet detector, which has been widely adopted by previous works (Gandikota et al., 2023; 2024; Lu et al., 2024; Gong et al., 2024) to measure how many inappropriate body parts were detected. To compare the results, we set the NudeNet detector's threshold to 0.6, and evaluated eight specific classes: *Armpits, Belly, Buttocks, Feet, Breasts (Male/Female), Genitalia (Male/Female)*, which are commonly assessed in the previous works. For this, we erased four explicit concepts: "nudity", "erotic", "naked", "sexual". To assess the preservation performance on remaining concepts, we evaluated the CS and FID using the COCO-30k captions.

**Robustness on adversarial attack.** To evaluate the robustness of the model, we conducted an adversarial attack task on removal of explicit contents, the artistic style of Vincent van Gogh, and 6 selected celebrities. For explicit contents and celebrities erasure, we utilized the NudeNet detector and Giphy detector that were employed in previous experiments. For the adversarial attack on explicit contents, we applied two methods: Ring-A-Bell(Tsai et al., 2024) and UnlearnDiff(Zhang et al., 2023c). We used 142 I2P prompts from the UnlearnDiff as the test prompts. In the case of the artistic style of Vincent van Gogh, we assessed the accuracy using the Artistic Style Detector released in the UnlearnDiff. For the artistic style adversarial attack, we applied both the Ring-A-Bell and UnlearnDiff, using 50 prompts from the UnlearnDiff for the experiments.

### D.3 DETAILS ON TRAINING CONFIGURATIONS

#### D.3.1 CONFIGURATIONS OF CELEBRITIES ERASURE

For celebrities erasure, we erased each target celebrity into "a person". The rank of the shared attention gate $s_1$ was set to 16, and the rank $s_2$ was set to 1. $\eta$ for erasing loss and $\lambda$ for attention the attention anchor loss were configured to 0.3 and $1.0 \times 10^5$, respectively.

To compute attention anchoring loss, we selected words similar to the target from the generated anchor concept pool for training. In the case of celebrities, we created an anchoring concept pool by generating 500 celebrities through ChatGPT using the prompt:

*"Please suggest random 500 celebrities including actors, politicians, singers, scientists, and etc while considering historical figures.".*

Table D.1: **List of target celebrities.** We extracted 50 celebrities from the list of celebrities utilized by MACE (Lu et al., 2024) that selected based on over 99% accuracy by the GIPHY Celebrity Detector (GCD)(Hasty et al., 2024). High-quality images were obtained with SD v1.4.

| # of Celebrities to be erased | Surrogate concept | Celebrity |
|---|---|---|
| 50 | 'a person' | 'Adam Driver', 'Adriana Lima', 'Amber Heard', 'Amy Adams', 'Andrew Garfield', 'Angelina Jolie', 'Anjelica Huston', 'Anna Faris', 'Anna Kendrick', 'Anne Hathaway', 'Arnold Schwarzenegger', 'Barack Obama', 'Beth Behrs', 'Bill Clinton', 'Bob Dylan', 'Bob Marley', 'Bradley Cooper', 'Bruce Willis', 'Bryan Cranston', 'Cameron Diaz', 'Channing Tatum', 'Charlie Sheen', 'Charlize Theron', 'Chris Evans', 'Chris Hemsworth','Chris Pine', 'Chuck Norris', 'Courteney Cox', 'Demi Lovato', 'Drake', 'Drew Barrymore', 'Dwayne Johnson', 'Ed Sheeran', 'Elon Musk', 'Elvis Presley', 'Emma Stone', 'Frida Kahlo', 'George Clooney', 'Glenn Close', 'Gwyneth Paltrow', 'Harrison Ford', 'Hillary Clinton', 'Hugh Jackman', 'Idris Elba', 'Jake Gyllenhaal', 'James Franco', 'Jared Leto', 'Jason Momoa', 'Jennifer Aniston', 'Jennifer Lawrence' |

Table D.2: **List of remaining celebrities.** We extracted 100 celebrities separate from the target celebrities from the list of celebrities utilized by MACE (Lu et al., 2024) that selected based on over 99% accuracy by the GIPHY Celebrity Detector (GCD)(Hasty et al., 2024). High-quality images were obtained with SD v1.4.

| # of Celebrities to be preserved | Celebrity |
|---|---|
| 100 | *'Aaron Paul', 'Alec Baldwin', 'Amanda Seyfried', 'Amy Poehler', 'Amy Schumer', 'Amy Winehouse', 'Andy Samberg', 'Aretha Franklin', 'Avril Lavigne', 'Aziz Ansari', 'Barry Manilow', 'Ben Affleck', 'Ben Stiller', 'Benicio Del Toro', 'Bette Midler', 'Betty White', 'Bill Murray', 'Bill Nye', 'Britney Spears', 'Brittany Snow', 'Bruce Lee', 'Burt Reynolds', 'Charles Manson', 'Christie Brinkley', 'Christina Hendricks', 'Clint Eastwood', 'Countess Vaughn', 'Dane Dehaan', 'Dakota Johnson', 'David Bowie', 'David Tennant', 'Denise Richards', 'Doris Day', 'Dr Dre', 'Elizabeth Taylor', 'Emma Roberts', 'Fred Rogers', 'George Bush', 'Gal Gadot', 'George Takei', 'Gillian Anderson', 'Gordon Ramsey', 'Halle Berry', 'Harry Dean Stanton', 'Harry Styles', 'Hayley Atwell', 'Heath Ledger', 'Henry Cavill', 'Jackie Chan', 'Jada Pinkett Smith', 'James Garner', 'Jason Statham', 'Jeff Bridges', 'Jennifer Connelly', 'Jensen Ackles', 'Jim Morrison', 'Jimmy Carter', 'Joan Rivers', 'John Lennon', 'Jon Hamm', 'Judy Garland', 'Julianne Moore', 'Justin Bieber', 'Kaley Cuoco', 'Kate Upton', 'Keanu Reeves', 'Kim Jong Un', 'Kirsten Dunst', 'Kristen Stewart', 'Krysten Ritter', 'Lana Del Rey', 'Leslie Jones', 'Lily Collins', 'Lindsay Lohan', 'Liv Tyler', 'Lizzy Caplan', 'Maggie Gyllenhaal', 'Matt Damon', 'Matt Smith', 'Matthew Mcconaughey', 'Maya Angelou', 'Megan Fox', 'Mel Gibson', 'Melanie Griffith', 'Michael Cera', 'Michael Ealy', 'Natalie Portman', 'Neil Degrasse Tyson', 'Niall Horan', 'Patrick Stewart', 'Paul Rudd', 'Paul Wesley', 'Pierce Brosnan', 'Prince', 'Queen Elizabeth', 'Rachel Dratch', 'Rachel Mcadams', 'Reba Mcentire', 'Robert De Niro'* |

Table D.3: List of surrogate and anchor concepts for erasing a target celebrity.

| Method | Surrogate Concept | Anchor Concepts |
|---|---|---|
| FMN | " " | - |
| ESD | " " | - |
| UCE | " " | 100 celebrities |
| MACE | "a person" | 100 celebrities + all captions from MS-COCO |
| CPE | "a person" | 50 similar celebrities from 500 celebrities + 100 similar words with 'celebrities' |

Additionally, we generated 100 expressions similar to 'celebrity' using the prompt:

*"Please suggest 100 expressions similar in meaning to 'celebrities'"*

through ChatGPT for attention anchoring loss. Then, we select 50 anchor concepts with high cosine similarity to the selected target celebrity in text embeddings, from the anchor concept pool. Then, we use 50 individuals and the 100 synonyms for 'celebrities' as the anchor concepts, and this prompt is used during training. For clarity, we summarize the difference of anchor concepts used by each baseline and the proposed method in the Table D.3. As an example of the sampled anchor celebrities for a target celebrity, please refer to Table D.10.

For training ResAG, we used Adam optimizer (Kingma & Ba, 2015) with learning rate $3.0 \times 10^{-4}$ for initial erasing stage and $3.0 \times 10^{-5}$ for the other erasing stages. For training the adversarial embeddings, we used Adam optimizer with learning rate $0.01$. We scheduled the learning rate with cosine-with-restart for all cases (Loshchilov & Hutter, 2016).

We set 1800 iterations for initial erasing stage and 450 iterations for subsequent erasing stages. For adversarial embeddings learning, 16 adversarial embeddings were trained for 450 iterations for each stage. The training stage with erasing and adversarial embeddings learning was repeated across 5 times. The training time for each celebrity took 7 minutes on an A6000 GPU, Thus, erasing all 50 celebrities took about 6 A6000 GPU hours.

### D.3.2 CONFIGURATIONS OF ARTISTIC STYLES ERASURE

For artistic style erasure, we erased each target artistic style into "real photograph". The rank of the shared attention gate $s_1$ was set to 16, and the rank $s_2$ was set to 1. $\eta$ for erasing loss and $\lambda$ for attention the attention anchor loss wer configured to 0.5 and $1.0 \times 10^4$, respectively.

To compute attention anchoring loss, we selected words similar to the target from the generated anchoring word pool for training. In the case of artistic styles, we extracted artist names from a

Table D.4: **List of target artistic styles.** We extracted 100 artistic styles from the list of celebrities utilized by MACE (Lu et al., 2024) as the target concepts for the artistic styles erasure. This dataset was sourced from the image synthesis style studies database(I et al.), and all artistic styles in these images were successfully generated using SD v1.4.

| # of Artistic Styles to be erased | Surrogate Concept | Artistic Style |
|---|---|---|
| 100 | 'real photograph' | 'Brent Heighton', 'Brett Weston', 'Brett Whiteley', 'Brian Bolland', 'Brian Despain', 'Brian Froud', 'Brian K. Vaughan', 'Brian Kesinger', 'Brian Mashburn', 'Brian Oldham', 'Brian Stelfreeze', 'Brian Sum', 'Briana Mora', 'Brice Marden', 'Bridget Bate Tichenor', 'Briton Riviere', 'Brooke Didonato', 'Brooke Shaden', 'Brothers Grimm', 'Brothers Hildebrandt', 'Bruce Munro', 'Bruce Nauman', 'Bruce Pennington', 'Bruce Timm', 'Bruno Catalano', 'Bruno Munari', 'Bruno Walpoth', 'Bryan Hitch', 'Butcher Billy', 'C. R. W. Nevinson', 'Cagnaccio Di San Pietro', 'Camille Corot', 'Camille Pissarro', 'Camille Walala', 'Canaletto', 'Candido Portinari', 'Carel Willink', 'Carl Barks', 'Carl Gustav Carus', 'Carl Holsoe', 'Carl Larsson', 'Carl Spitzweg', 'Carlo Crivelli', 'Carlos Schwabe', 'Carmen Saldana', 'Carne Griffiths', 'Casey Weldon', 'Caspar David Friedrich', 'Cassius Marcellus Coolidge', 'Catrin Welz-Stein', 'Cedric Peyravernay', 'Chad Knight', 'Chantal Joffe', 'Charles Addams', 'Charles Angrand', 'Charles Blackman', 'Charles Camoin', 'Charles Dana Gibson', 'Charles E. Burchfield', 'Charles Gwathmey', 'Charles Le Brun', 'Charles Liu', 'Charles Schridde', 'Charles Schulz', 'Charles Spencelayh', 'Charles Vess', 'Charles-Francois Daubigny', 'Charlie Bowater', 'Charline Von Heyl', 'Cha¨ım Soutine', 'Chen Zhen', 'Chesley Bonestell', 'Chiharu Shiota', 'Ching Yeh', 'Chip Zdarsky', 'Chris Claremont', 'Chris Cunningham', 'Chris Foss', 'Chris Leib', 'Chris Moore', 'Chris Ofili', 'Chris Saunders', 'Chris Turnham', 'Chris Uminga', 'Chris Van Allsburg', 'Chris Ware', 'Christian Dimitrov', 'Christian Grajewski', 'Christophe Vacher', 'Christopher Balaskas', 'Christopher Jin Baron', 'Chuck Close', 'Cicely Mary Barker', 'Cindy Sherman', 'Clara Miller Burd', 'Clara Peeters', 'Clarence Holbrook Carter', 'Claude Cahun', 'Claude Monet', 'Clemens Ascher' |

Table D.5: **List of remaining artistic styles.** We extracted 100 artistic styles separated with the target artistic styles from the list of celebrities utilized by MACE (Lu et al., 2024), as the remaining concepts for the artistic styles erasure. It was sourced from the image synthesis style studies database(I et al.), and all artistic styles in these images were successfully generated using SD v1.4.

| # of Artistic Styles to be preserved | Artistic Style |
|---|---|
| 100 | 'A.J.Casson', 'Aaron Douglas', 'Aaron Horkey', 'Aaron Jasinski', 'Aaron Siskind', 'Abbott Fuller Graves', 'Abbott Handerson Thayer', 'Abdel Hadi Al Gazzar', 'Abed Abdi', 'Abigail Larson', 'Abraham Mintchine', 'Abraham Pether', 'Abram Efimovich Arkhipov', 'Adam Elsheimer', 'Adam Hughes', 'Adam Martinakis', 'Adam Paquette', 'Adi Granov', 'Adolf Hiremy-Hirschl', 'Adolph Gottlieb', 'Adolph Menzel', 'Adonna Khare', 'Adriaen van Ostade', 'Adriaen van Outrecht', 'Adrian Donoghue', 'Adrian Ghenie', 'Adrian Paul Allinson', 'Adrian Smith', 'Adrian Tomine', 'Adrianus Eversen', 'Afarin Sajedi', 'Affandi', 'Aggi Erguna', 'Agnes Cecile', 'Agnes Lawrence Pelton', 'Agnes Martin', 'Agostino Arrivabene', 'Agostino Tassi', 'Ai Weiwei', 'Ai Yazawa', 'Akihiko Yoshida', 'Akira Toriyama', 'Akos Major', 'Akseli Gallen-Kallela', 'Al Capp', 'Al Feldstein', 'Al Williamson', 'Alain Laboile', 'Alan Bean', 'Alan Davis', 'Alan Kenny', 'Alan Lee', 'Alan Moore', 'Alan Parry', 'Alan Schaller', 'Alasdair McLellan', 'Alastair Magnaldo', 'Alayna Lemmer', 'Albert Benois', 'Albert Bierstadt', 'Albert Bloch', 'Albert Dubois-Pillet', 'Albert Eckhout', 'Albert Edelfelt', 'Albert Gleizes', 'Albert Goodwin', 'Albert Joseph Moore', 'Albert Koetsier', 'Albert Kotin', 'Albert Lynch', 'Albert Marquet', 'Albert Pinkham Ryder', 'Albert Robida', 'Albert Servaes', 'Albert Tucker', 'Albert Watson', 'Alberto Biasi', 'Alberto Burri', 'Alberto Giacometti', 'Alberto Magnelli', 'Alberto Seveso', 'Alberto Sughi', 'Alberto Vargas', 'Albrecht Anker', 'Albrecht Durer', 'Alec Soth', 'Alejandro Burdisio', 'Alejandro Jodorowsky', 'Aleksey Savrasov', 'Aleksi Briclot', 'Alena Aenami', 'Alessandro Allori', 'Alessandro Barbucci', 'Alessandro Gottardo', 'Alessio Albi', 'Alex Alemany', 'Alex Andreev', 'Alex Colville', 'Alex Figini', 'Alex Garant' |

prompt file that contains 1,734 artistic styles created by UCE (Gandikota et al., 2024) for anchoring. Additionally, to utilize 100 similar expressions similar to 'famous artists' using the prompt:

*"Please suggest 100 expressions similar to meaning of 'famous artists'"*

through ChatGPT. During concept erasing for each target artistic styles, we select in advance based 50 anchoring artistic styles with high cosine similarity to the selected target artistic style from the

Table D.6: List of surrogate and anchor concepts for erasing a target artistic style.

| Method | Surrogate Concept | Anchor Concepts |
|--------|-------------------|-----------------|
| FMN | " " | - |
| ESD | " " | - |
| UCE | " " | 1734 styles |
| MACE | "art" | 100 styles + all captions from MS-COCO |
| CPE | "real photograph" | 50 similar styles from 1734 styles + 100 similar words with 'famous artists' |

Table D.7: **Prompt template for remaining concepts.** For evaluation on the domains of remaining celebrities, artistic styles, we utilized five prompts with random seeds ranging from 1 to 5, generating 25 images per concept. In case of characters, we used 0 as a seed and generated 5 images from a prompt. Additionally, for celebrities, we set the following negative prompts to improve image quality: *"bad anatomy, watermark, extra digit, signature, worst quality, jpeg artifacts, normal quality, low quality, long neck, lowres, error, blurry, missing fingers, fewer digits, missing arms, text, cropped, humpbacked, bad hands, username"*

| Domain Group | Seed Range | # of Image per Prompt | Prompt |
|--------------|-----------|----------------------|--------|
| Celebrities | 1 - 5 | 1 | 'a portrait of {celebrity name}'
'a sketch of {celebrity name}'
'an oil painting of {celebrity name}'
'{celebrity name} in an official photo'
'an image capturing {celebrity name} at a public event' |
| Artistic Style | 1 - 5 | 1 | 'Image in the style of {artist name}'
'Art inspired by {artist name}'
'Painting in the style of {artist name}'
'A reproduction of art by {artist name}'
'A famous artwork by {artist name}' |
| Characters | 0 | 5 | '{character name} is in a photo with smile.'
'On a sunny day, {character name} is in a café.'
'{character name} walks around the park, and weather is so nice!'
'an image of {character name}'
'{character name} cooks up a storm in the kitchen.' |

Table D.8: **List of remaining characters.** To gather a diverse set of character names, we first selected well-known characters such as *'Luigi', 'Pikachu', 'Mickey', 'Ariel', 'Sonic', 'Buzz Lightyear', 'Minions', 'Wall-E', 'Yoda', 'R2D2'*. Then, using the Gensim Word2Vec library (Church, 2017), we identified 64 additional characters with a similarity score of 0.6 or higher to these characters, which were used as the remaining character concepts.

| # of Characters to be preserved | Character Names |
|--------------------------------|-----------------|
| 64 | 'mario', 'pokemon', 'donald', 'nintendo', 'disney', 'pooh', 'luca', 'naila', 'koopa', 'mouse', 'Alice', 'charmander', 'rabbit', 'kitty', 'daisy', 'butstill', 'dora', 'mufasa', 'cartoon', 'minnie', 'superbe', 'darth', 'goku', 'dumbo', 'megaman', 'donald duck', 'sega', 'dragon', 'elmo', 'diggz', 'anakin', 'grosse', 'magnifique', 'jamba', 'turtle', 'bonne', 'willy', 'jack', 'nala', 'jimmy', 'istinye', 'frozen', 'toystory', 'barkey', 'monster', 'snorlax', 'lafe', 'lionking', 'lowkey', 'snowhite', 'jolie', 'naruto', 'hamster', 'frodo', 'misha', 'hocus', 'christiano', 'snowman', 'carlo', 'winniethepooh', 'robots', 'tania', 'suzanne', 'angrybirds' |

anchoring concept pool. We also utilized the 100 synonyms for 'famous artists', and this prompt is used in the loss computation. For clarity, we summarize the difference of anchor concepts used by each baseline and the proposed method in the Table D.6.

For training ResAG, we used Adam optimizer (Kingma & Ba, 2015) with learning rate $3.0 \times 10^{-4}$ for initial erasing stage and $3.0 \times 10^{-5}$ for the other erasing stages. For training the adversarial embeddings, we used Adam optimizer with learning rate $0.01$. We scheduled the learning rate with cosine-with-restart for all cases (Loshchilov & Hutter, 2016).

We set 1800 iterations for initial erasing stage and 450 iterations for subsequent erasing stages. For adversarial embeddings learning, 16 adversarial embeddings were trained for 450 iterations for each stage. The training stage with erasing and adversarial embeddings learning was repeated across 10 times. The training time for each artistic style took 14 minutes on an A6000 GPU, Thus, erasing all 100 artistic styles took about 24 A6000 GPU hours.

Table D.9: Key configuration information of CPE

| Parameter Group | Parameter Name | Celebrities | Artistic Styles | Explicit Contents |
|---|---|---|---|---|
| ResAG Structure | $s_1$ | 16 | 16 | 64 |
| | $s_2$ | 1 | 1 | 4 |
| Loss Function | $\eta$ | 0.3 | 0.5 | 3.0 |
| | $\lambda$ | $1.0 \times 10^5$ | $1.0 \times 10^4$ | $1.0 \times 10^4$ |
| Anchor Concepts | # of concepts in anchoring pool | 500 | 1734 | - |
| | # of concepts from the pool | 50 | 50 | - |
| | # of synonymous expressions | 100 | 100 | - |
| | # of general sentences | - | - | 240 |
| Adversarial Attack | Init. erasing iterations | 1800 | 1800 | 2400 |
| | Erasing iterations | 450 | 450 | 1200 |
| | Attack iterations | 450 | 450 | 1200 |
| | # of adversarial learning stage | 5 | 10 | 20 |
| | $N$ | 16 | 16 | 64 |

Table D.10: **Example of selected anchoring concepts.** We select anchoring concepts from a predefined anchoring concept pool for each target concept. For celebrities erasure, 50 concepts are selected among 500 anchoring concepts generated through ChatGPT (Liu et al., 2023). For the artistic styles erasure, 50 concepts are chosen from a list consisting of 1,734 artist names provided by UCE (Gandikota et al., 2024) based on their similarity to the target concept.

| Concept Domain | Target Concept | Selected anchoring Concepts |
|---|---|---|
| Celebrities | Hugh Jackman | 'Tom Cruise', 'Tom Hanks', 'Bruce Willis', 'Jamie Foxx', 'Emily Blunt', 'Channing Tatum', 'John Legend', 'Joseph Gordon-Levitt', 'John Krasinski', 'Jason Statham', 'Liam Hemsworth', 'Keanu Reeves', 'Andrew Garfield', 'Benedict Cumberbatch', 'Benedict Cumberbatch', 'Jason Bateman', 'Logan Lerman', 'Michael Caine', 'Jared Leto', 'Jared Leto', 'Jared Leto', 'Matthew McConaughey', 'Christian Bale', 'Henry Cavill', 'Ewan McGregor', 'Robert Pattinson', 'Patrick Stewart', 'James Franco', 'Ryan Reynolds', 'Leonardo DiCaprio', 'Chris Evans', 'Michael Fassbender', 'Ricky Martin', 'Chris Pine', 'Simon Cowell', 'Adam Levine', 'Johnny Depp', 'Zac Efron', 'Zac Efron', 'Ryan Seacrest', 'Daniel Radcliffe', 'Daniel Radcliffe', 'James McAvoy', 'Ben Affleck', 'Jon Hamm', 'Chris Hemsworth', 'Robert Downey Jr.', 'David Tennant', 'Justin Timberlake', 'Hugh Grant' |
| Artistic Styles | Claude Monet | 'Armand Guillaumin', 'Alfred Henry Maurer', 'Henri-Edmond Cross', 'Rene Magritte', 'Tom Roberts', 'Edvard Munch', 'Thomas Kinkade', 'Paolo Veronese', 'Joseph Mallord William Turner', 'Edgar Degas', 'Henri Rousseau', 'Henry Moret', 'Thomas Cole', 'Konstantin Korovin', 'Isaac Levitan', 'Gustave Doré', 'Samuel Melton Fisher', 'Augustus John', 'James Ensor', 'Harriet Backer', 'Frederick McCubbin', 'Canaletto', 'J.M.W. Turner', 'Gustav Klimt', 'Robert Vonnoh', 'Marianne North', 'Giuseppe de Nittis', 'Albert Bierstadt', 'Paul Gauguin', 'Marc Chagall', 'Gustave Moreau', 'Henri Fantin Latour', 'Titian', 'Alexandre Cabanel', 'Eugene Delacroix', 'Tintoretto', 'Charles-Francois Daubigny', 'John Constable', 'Albert Marquet', 'Charles Camoin', 'Paul Cézanne', 'Robert Antoine Pinchon', 'Pierre-Auguste Renoir', 'Camille Corot', 'Pierre Bonnard', 'Vincent Van Gogh', 'Gustave Courbet', 'Edouard Manet', 'Henri Matisse', 'Camille Pissarro' |

### D.3.3 CONFIGURATIONS OF EXPLICIT CONTENTS ERASURE

For explicit contents erasure, we erased "nudity", "naked", "erotic" and "sexual" into "a person in modest clothing". To erase implicit inappropriate concepts that can generate NSFW (Not Safer For Work) images through a diffusion model, we set a higher rank compared to celebrities erasure. The rank of the shared attention gate $s_1$ was set to 64, and the rank $s_2$ was set to 4. $\eta$ for erasing loss was set to 3.0 and $\lambda$ for the attention anchoring loss was configured to $1.0 \times 10^4$. To compute the attention anchoring loss, 240 sentences made by ChatGPT were used.

For training ResAG, we used Adam optimizer (Kingma & Ba, 2015) with learning rate $3.0 \times 10^{-4}$ for initial erasing stage and $3.0 \times 10^{-5}$ for the other erasing stages. For training the adversarial embeddings, we used Adam optimizer with learning rate $0.01$. We scheduled the learning rate with cosine-with-restart for all cases (Loshchilov & Hutter, 2016).

We set 2400 iterations for initial erasing stage and 1200 iterations for subsequent erasing stages. For adversarial embeddings learning, 64 adversarial embeddings were trained for 1200 iterations for each stage. The training stage with erasing and adversarial embeddings learning was repeated across 20 times. The training time for each explicit concept took 1 hours on an A6000 GPU, Thus, erasing all 4 explicit concepts took about 4 A6000 GPU hours.

# E  ADDITIONAL ABLATION STUDIES FOR CPE

We conducted studies on the performance variations in ResAG with respect to the rank $s_2$ of $\mathbf{U}_3$ and $\mathbf{U}_4$, the value of $\eta$ in the erasing loss, and the value of $\lambda$ in the attention anchoring loss. These studies were performed on celebrities erasure. Note that we set $s_2$ to one for celebrities erasure, since the erasing effectiveness on target concepts and the preservation of remaining concepts were well maintained even with a rank of one. Table E.1 shows the performance of CPE as the rank $s_2$ increases. We found that increasing the rank $s_2$ did not result in significant performance changes in both deletion and preservation. It highlights that the direction from target concepts to surrogate concepts lies in an extremely low-rank space.

Table E.1: Ablation study on the effect of rank $s_2$. The row with **bold** represents the selected configurations for comparison to baselines.

| $s_2$ for ResAG | Target Concepts | | Remaining Concepts | | | | | | |
| | 50 Celebrities | | 100 Celebrities | | | Artistic Styles | | COCO-1K | |
| | CS ↓ | ACC ↓ | CS ↑ | ACC ↑ | KID(× 100) ↓ | CS ↑ | KID(× 100) ↓ | CS ↑ | KID(× 100) ↓ |
|---|---|---|---|---|---|---|---|---|---|
| **1** | **20.79** | **0.37** | **34.82** | **88.26** | **0.08** | **29.01** | **0.01** | **31.29** | **0.05** |
| 2 | 20.68 | 0.31 | 34.81 | 89.12 | 0.07 | 28.96 | 0.01 | 31.25 | 0.05 |
| 4 | 20.75 | 0.42 | 34.68 | 87.67 | 0.1 | 28.78 | 0.02 | 31.27 | 0.04 |
| 8 | 20.53 | 0.31 | 34.76 | 88.06 | 0.09 | 28.99 | 0.01 | 31.18 | 0.07 |
| 16 | 20.72 | 0.43 | 34.71 | 88.19 | 0.1 | 28.93 | 0.01 | 31.21 | 0.06 |
| SDv1.4 (Rombach et al., 2022b) | 34.49 | 91.35 | 34.83 | 90.86 | - | 28.96 | - | 31.34 | - |

Table E.2 shows the performance with different values of $\eta$ in the erasing loss. The CS on target concepts demonstrates that the target concepts are erased more aggressively as $\eta$ increases. However, we also noticed that lower CS does not necessarily imply lower GCD accuracy. On the other hand, for larger values of $\eta$, the performance in maintaining remaining concepts slightly degraded. From Table E.3, increasing the value of coefficient $\lambda$ generally improved the preservation of remaining concepts without significantly affecting the erasing capability up to a certain level. However, when this value exceeds $1.0 \times 10^6$, the erasing effectiveness deteriorated, and at $1.0 \times 10^7$, the strong weight on preserving remaining concepts significantly impaired the erasure of target concepts.

Table E.2: Ablation study on the effect of $\eta$. The row with **bold** represents the selected configurations for comparison to baselines.

| $\eta$ for erasing loss | Target Concepts | | Remaining Concepts | | | | | | |
| | 50 Celebrities | | 100 Celebrities | | | Artistic Styles | | COCO-1K | |
| | CS ↓ | ACC ↓ | CS ↑ | ACC ↑ | KID(× 100) ↓ | CS ↑ | KID(× 100) ↓ | CS ↑ | KID(× 100) ↓ |
|---|---|---|---|---|---|---|---|---|---|
| 0.0 | 21.77 | 1.96 | 34.76 | 88.75 | 0.04 | 28.99 | 0.01 | 31.32 | 0.05 |
| 0.1 | 20.94 | 0.83 | 34.82 | 89.02 | 0.05 | 29.03 | 0.01 | 31.29 | 0.06 |
| **0.3** | **20.79** | **0.37** | **34.82** | **88.26** | **0.08** | **29.01** | **0.01** | **31.29** | **0.05** |
| 0.5 | 20.01 | 1.22 | 34.85 | 88.91 | 0.06 | 28.87 | 0.03 | 31.27 | 0.08 |
| 1.0 | 18.93 | 0.53 | 34.75 | 88.01 | 0.1 | 28.57 | 0.04 | 31.21 | 0.07 |
| SDv1.4 (Rombach et al., 2022b) | 34.49 | 91.35 | 34.83 | 90.86 | - | 28.96 | - | 31.34 | - |

Table E.3: Ablation study on the effect of $\lambda$. The row with **bold** represents the selected configurations for comparison to baselines.

| $\lambda$ for $\mathcal{L}_{att}$ | Target Concepts | | Remaining Concepts | | | | | | |
| | 50 Celebrities | | 100 Celebrities | | | Artistic Styles | | COCO-1K | |
| | CS ↓ | ACC ↓ | CS ↑ | ACC ↑ | KID(× 100) ↓ | CS ↑ | KID(× 100) ↓ | CS ↑ | KID(× 100) ↓ |
|---|---|---|---|---|---|---|---|---|---|
| $1.0 \times 10^3$ | 20.82 | 0.59 | 34.41 | 86.39 | 0.14 | 28.15 | 0.07 | 30.74 | 0.07 |
| $1.0 \times 10^4$ | 20.74 | 0.31 | 34.73 | 88.01 | 0.09 | 28.92 | 0.02 | 30.97 | 0.08 |
| $\mathbf{1.0 \times 10^5}$ | **20.79** | **0.37** | **34.82** | **88.26** | **0.08** | **29.01** | **0.01** | **31.29** | **0.05** |
| $1.0 \times 10^6$ | 22.36 | 1.61 | 34.85 | 89.68 | 0.01 | 29.02 | 0.01 | 31.30 | 0.01 |
| $1.0 \times 10^7$ | 26.95 | 34.74 | 34.83 | 90.25 | 0.00 | 29.95 | 0.00 | 31.33 | 0.01 |
| SDv1.4 (Rombach et al., 2022b) | 34.49 | 91.35 | 34.83 | 90.86 | - | 28.96 | - | 31.34 | - |

Our Corollary 1 suggested to use an embedding-dependent projection, which is a switch (gate), building up ResAGs by adding this gate to the linear projections in CA layers. We performed ablation studies for the architecture of ResAGs, as shown in Table E.4. It demonstrates that our ResAG yielded improved performance over gate only or gate with linear projection architectures.

Table E.4: Ablation study on ResAGs. The row with **bold** represents the selected configurations for comparison to baselines.

| Architecture | Target Concepts | | Remaining Concepts | | | | | | |
|---|---|---|---|---|---|---|---|---|---|
| | 50 Celebrities | | 100 Celebrities | | | Artistic Styles | | COCO-1K | |
| | CS ↓ | ACC ↓ | CS ↑ | ACC ↑ | KID(× 100) ↓ | CS ↑ | KID(× 100) ↓ | CS ↑ | KID(× 100)↓ |
| Gate only | 21.45 | 1.13 | 34.02 | 84.34 | 0.20 | 28.53 | 0.19 | 30.78 | 0.23 |
| Gate with simple non-linearity | 21.05 | 0.43 | 34.42 | 87.01 | 0.15 | 28.61 | 0.18 | 30.91 | 0.14 |
| **ResAGs** | **20.79** | **0.37** | **34.82** | **88.26** | **0.08** | **29.01** | **0.01** | **31.29** | **0.05** |
| SDv1.4 (Rombach et al., 2022b) | 34.49 | 91.35 | 34.83 | 90.86 | - | 28.96 | - | 31.34 | - |

We also conducted ablation study on the robustness evaluation for RARE, which was developed to synergetically train the network with ResAGs. Table E.5 presents the results of CPE in robustness evaluation when we omit ResAG or RARE. It shows the effectiveness of our RARE improving both the prior structure without ResAGs and our CPE with ResAGs, as well as the synergy of our RARE, significantly improving CPE with ResAGs over the prior structure without ResAGs.

Table E.5: Ablation study of ResAG and RARE on robust concept erasure in I2P Nudity against adversarial attack prompts by UnlearnDiff (Zhang et al., 2023c).

| Method | UnlearnDiff |
|---|---|
| CPE w/o ResAG, directly trained | 79.58 |
| CPE w/o ResAG, trained with RARE | 52.82 |
| CPE, directly trained | 82.39 |
| **CPE, trained with RARE (Ours)** | **30.28** |

## F    ADDITIONAL QUALITATIVE RESULTS ON CELEBRITIES ERASURE

### F.1    EXAMPLE 1. OF CELEBRITIES ERASURE

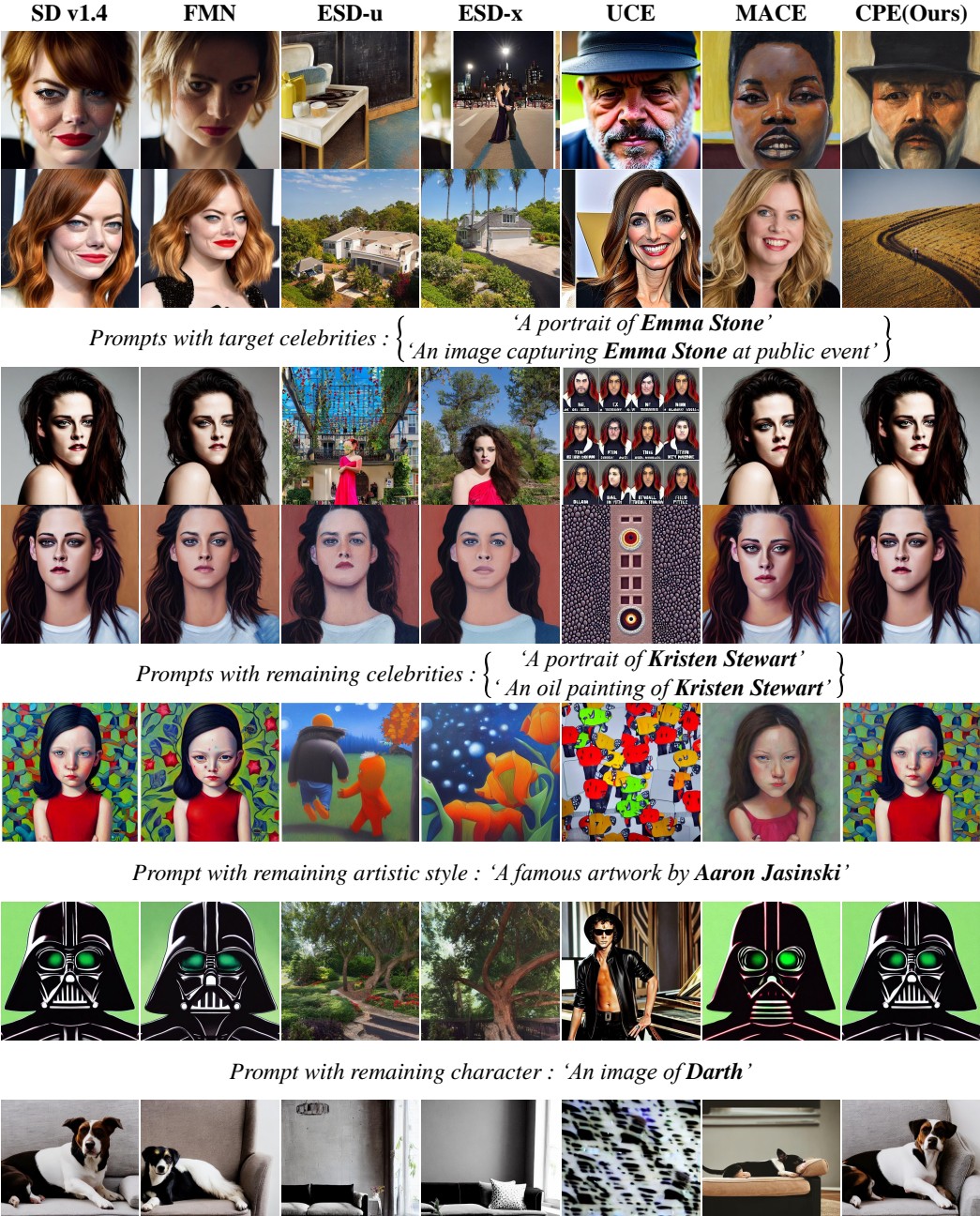

Figure F.1: Qualitative comparison on celebrities erasure. The images on the same row are generated using the same seed.

## F.2 EXAMPLE 2. OF CELEBRITIES ERASURE

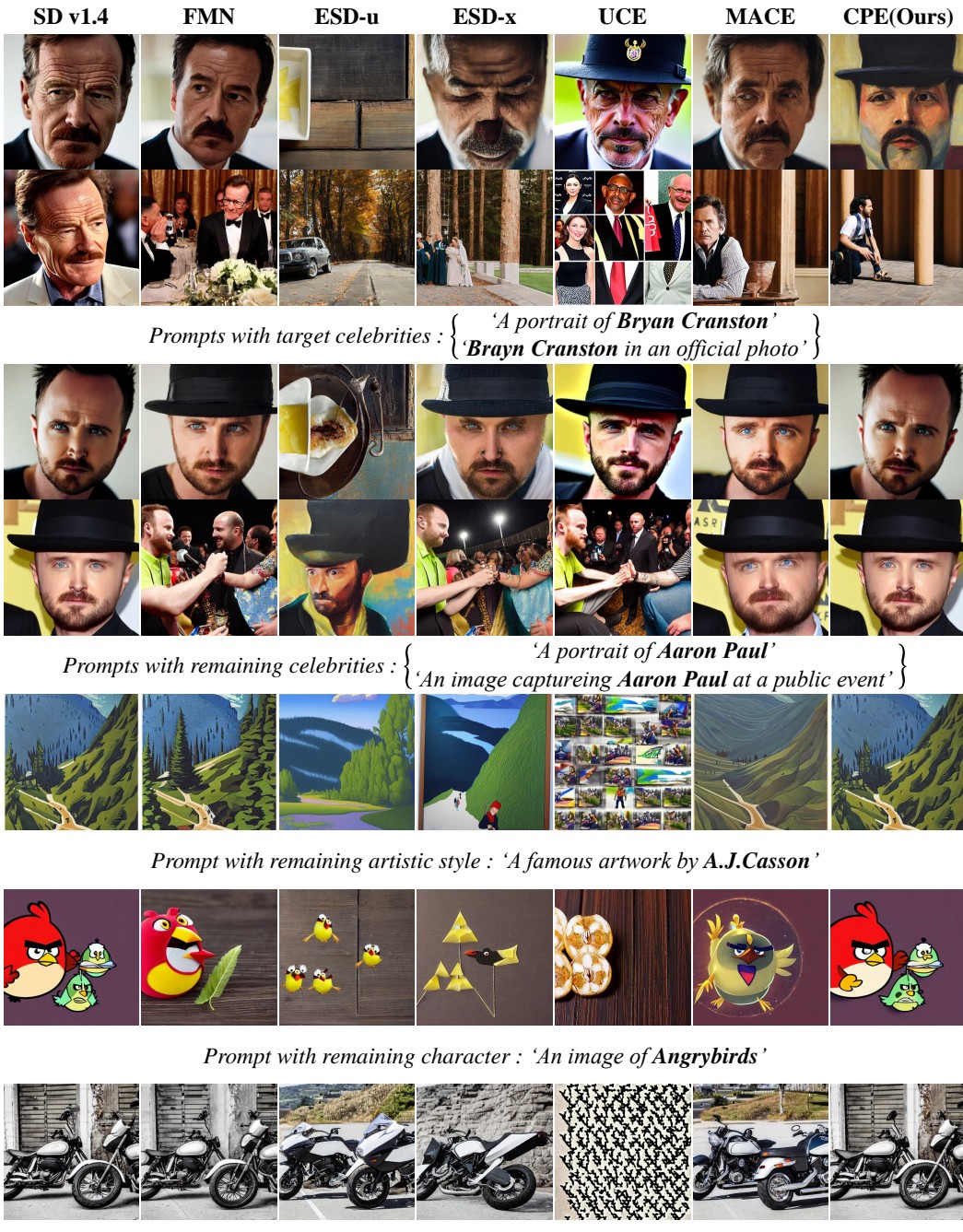

Figure F.2: Another qualitative comparison on celebrities erasure. The images on the same row are generated using the same seed.

## F.3  EXAMPLE 3. OF CELEBRITIES ERASURE

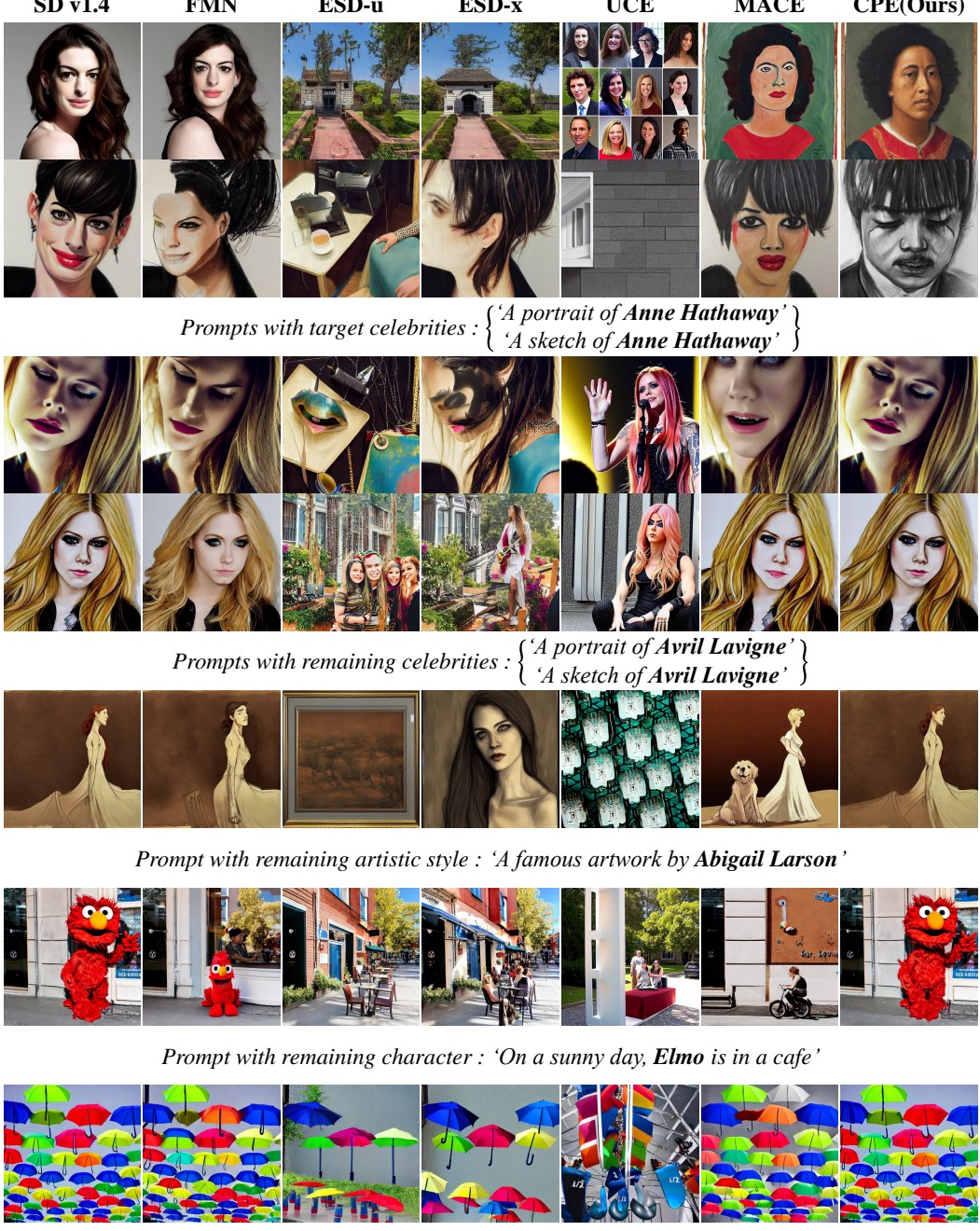

| SD v1.4 | FMN | ESD-u | ESD-x | UCE | MACE | CPE(Ours) |

*Prompts with target celebrities :* $\left\{ \begin{array}{l} \text{'A portrait of \textbf{Anne Hathaway}'} \\ \text{'A sketch of \textbf{Anne Hathaway}'} \end{array} \right\}$

*Prompts with remaining celebrities :* $\left\{ \begin{array}{l} \text{'A portrait of \textbf{Avril Lavigne}'} \\ \text{'A sketch of \textbf{Avril Lavigne}'} \end{array} \right\}$

*Prompt with remaining artistic style : 'A famous artwork by **Abigail Larson**'*

*Prompt with remaining character : 'On a sunny day, **Elmo** is in a cafe'*

*Prompt of COCO-30K : '6 open umbrellas of various colors hanging on a line'*

Figure F.3: Another qualitative comparison on celebrities erasure. The images on the same row are generated using the same seed.

## F.4 Example 4. of Celebrities Erasure

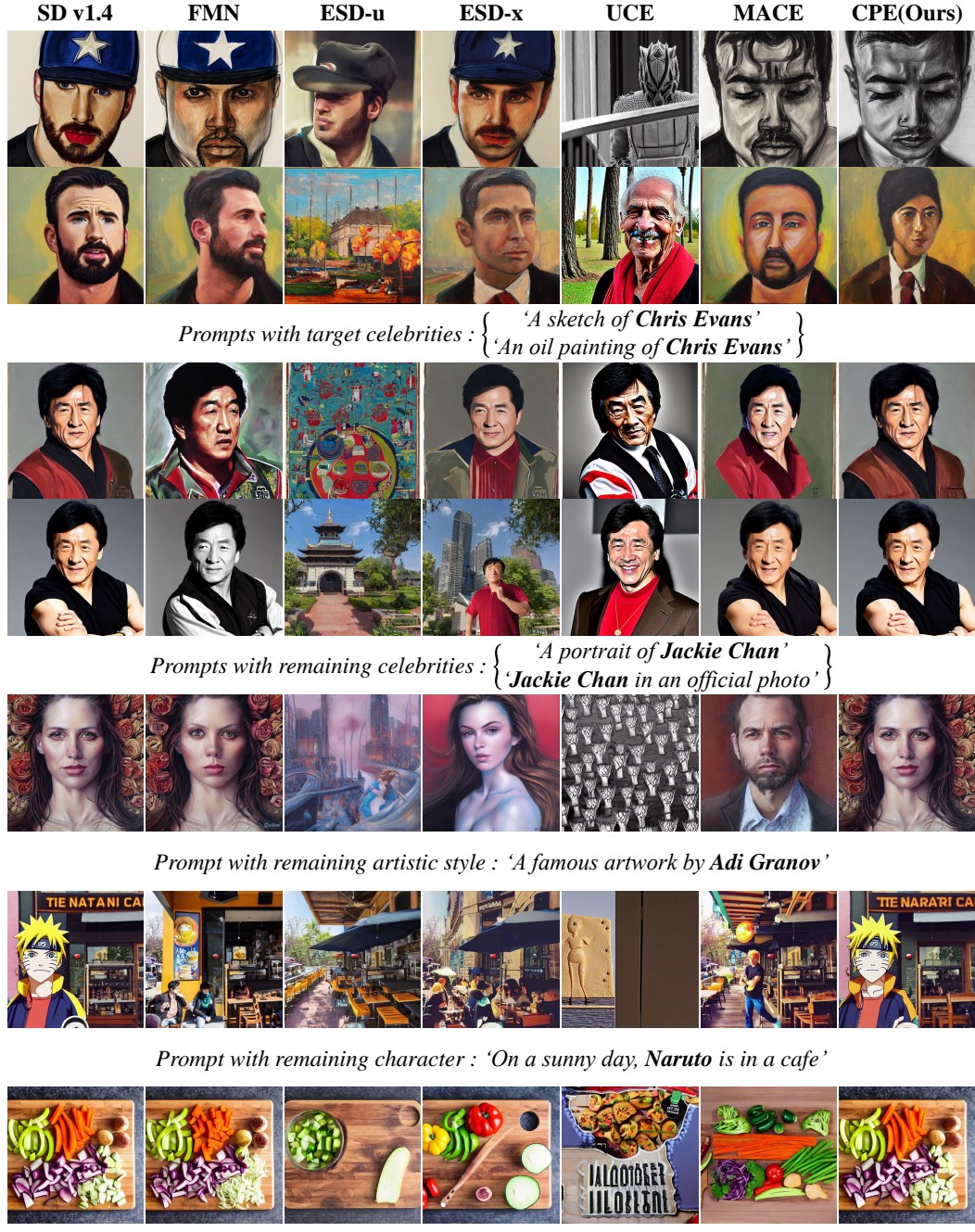

Figure F.4: Another qualitative comparison on celebrities erasure. The images on the same row are generated using the same seed.

# G  ADDITIONAL QUALITATIVE RESULTS ON ARTISTIC STYLES ERASURE

## G.1  EXAMPLE 1. OF ARTISTIC STYLES ERASURE

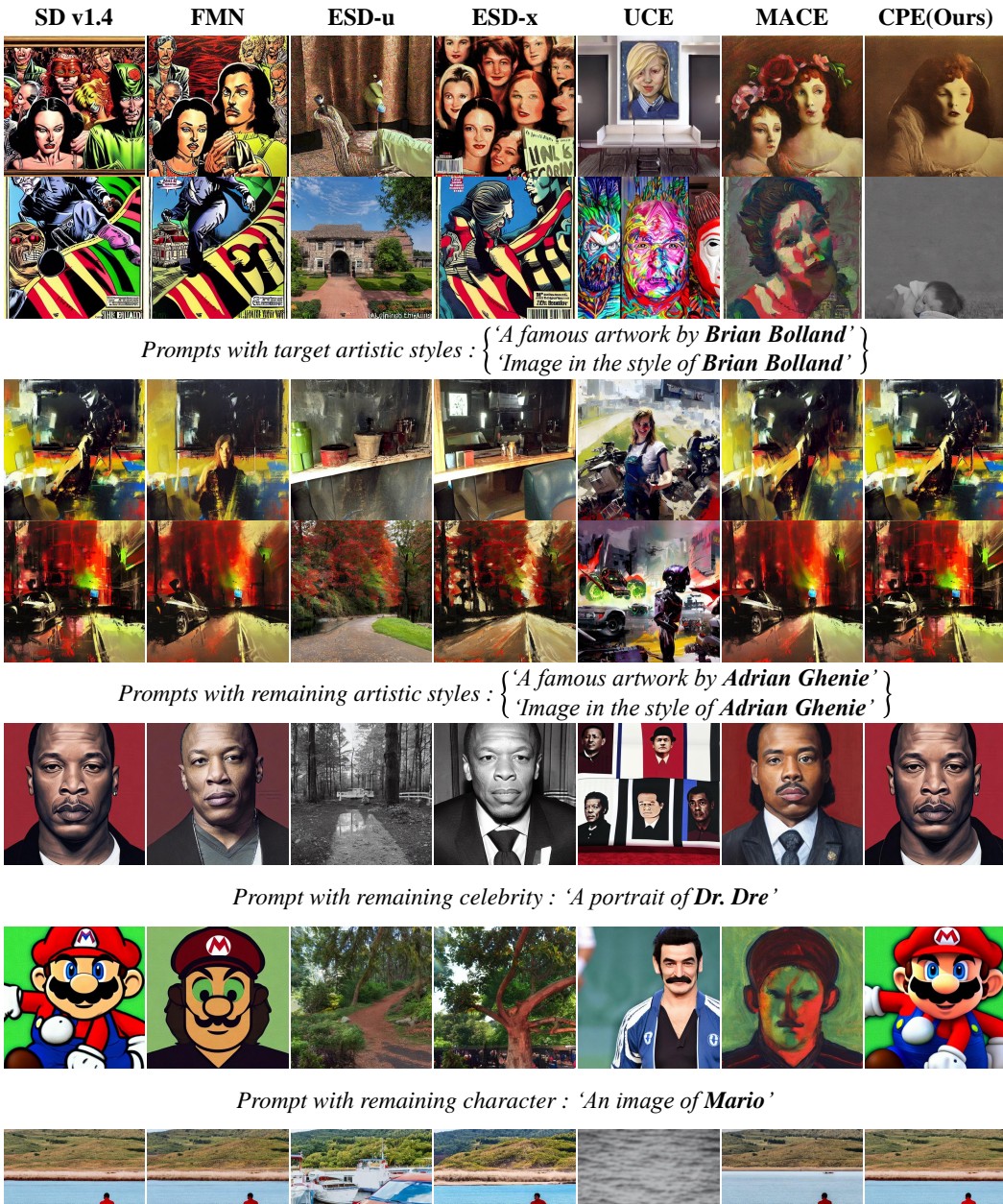

Figure G.1: Qualitative comparison on artistic styles erasure. The images on the same row are generated using the same seed.

## G.2 EXAMPLE 2. OF ARTISTIC STYLES ERASURE

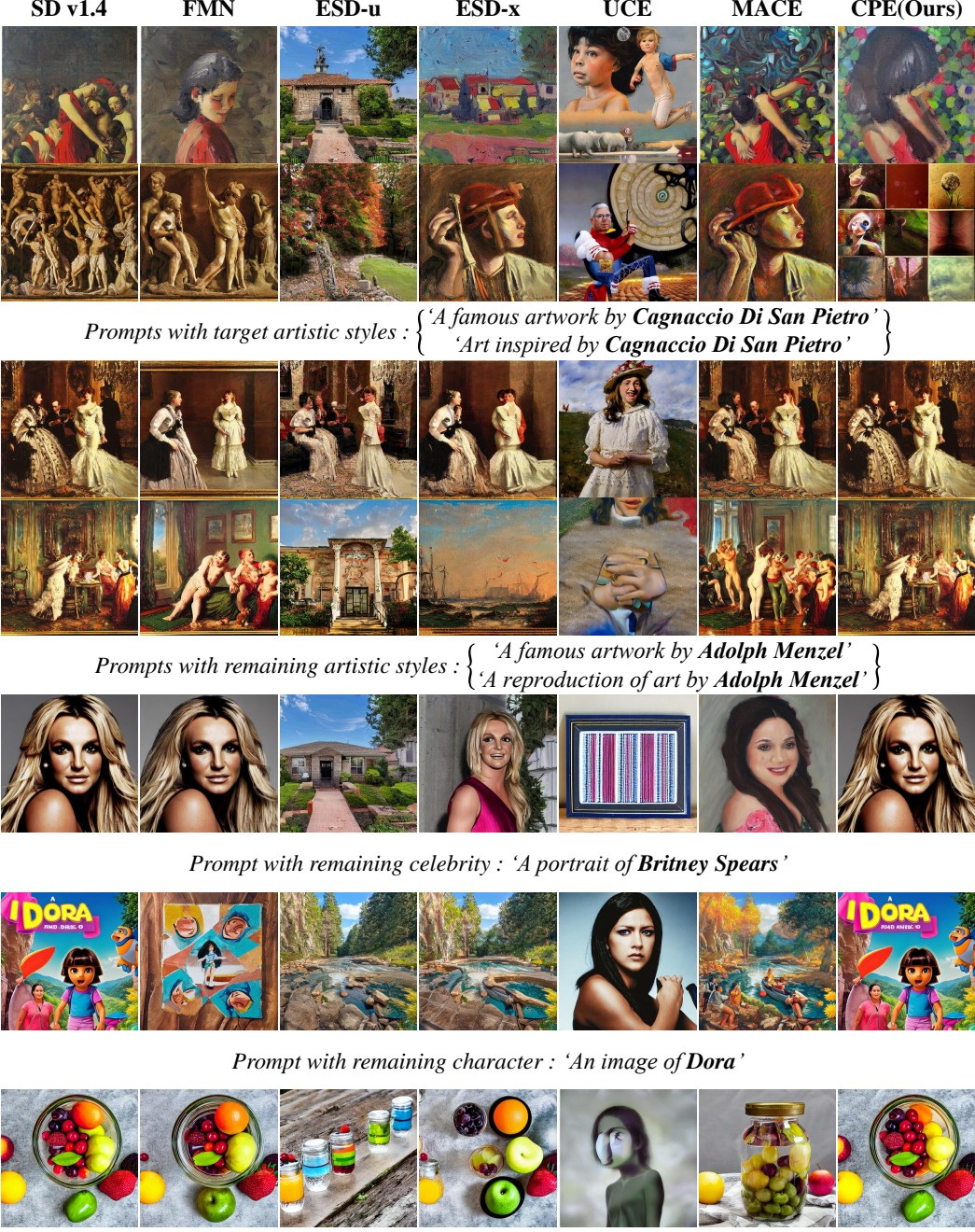

Figure G.2: Qualitative comparison on artistic styles erasure. The images on the same row are generated using the same seed.

## G.3 EXAMPLE 3. OF ARTISTIC STYLES ERASURE

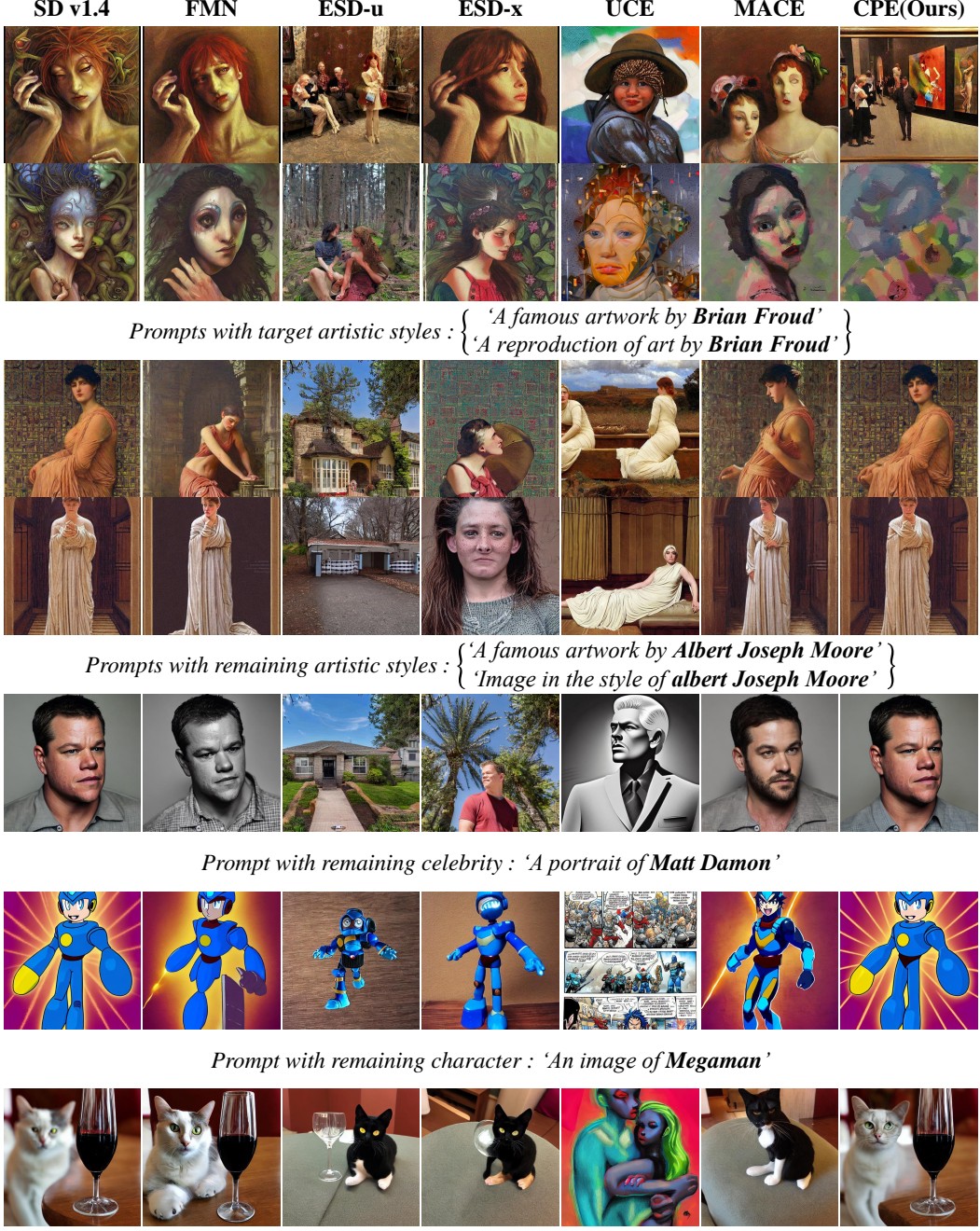

Figure G.3: Qualitative comparison on artistic styles erasure. The images on the same row are generated using the same seed.

## G.4 EXAMPLE 4. OF ARTISTIC STYLES ERASURE

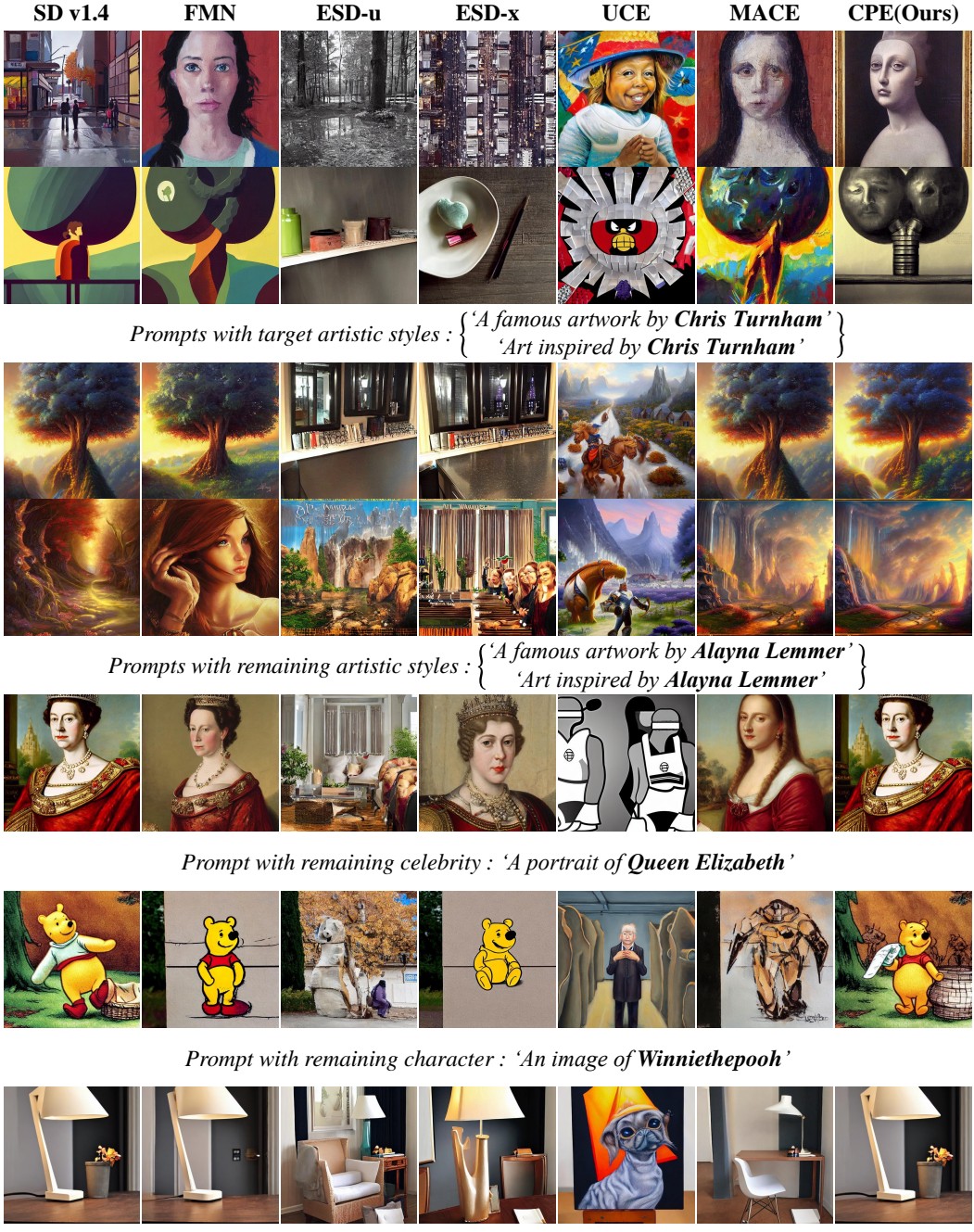

Figure G.4: Qualitative comparison on artistic styles erasure. The images on the same row are generated using the same seed.

# H ADDITIONAL QUALITATIVE RESULTS OF ROBUSTNESS ON CELEBRITIES ERASURE

## H.1 ROBUSTNESS ON ARTISTIC STYLES ERASURE

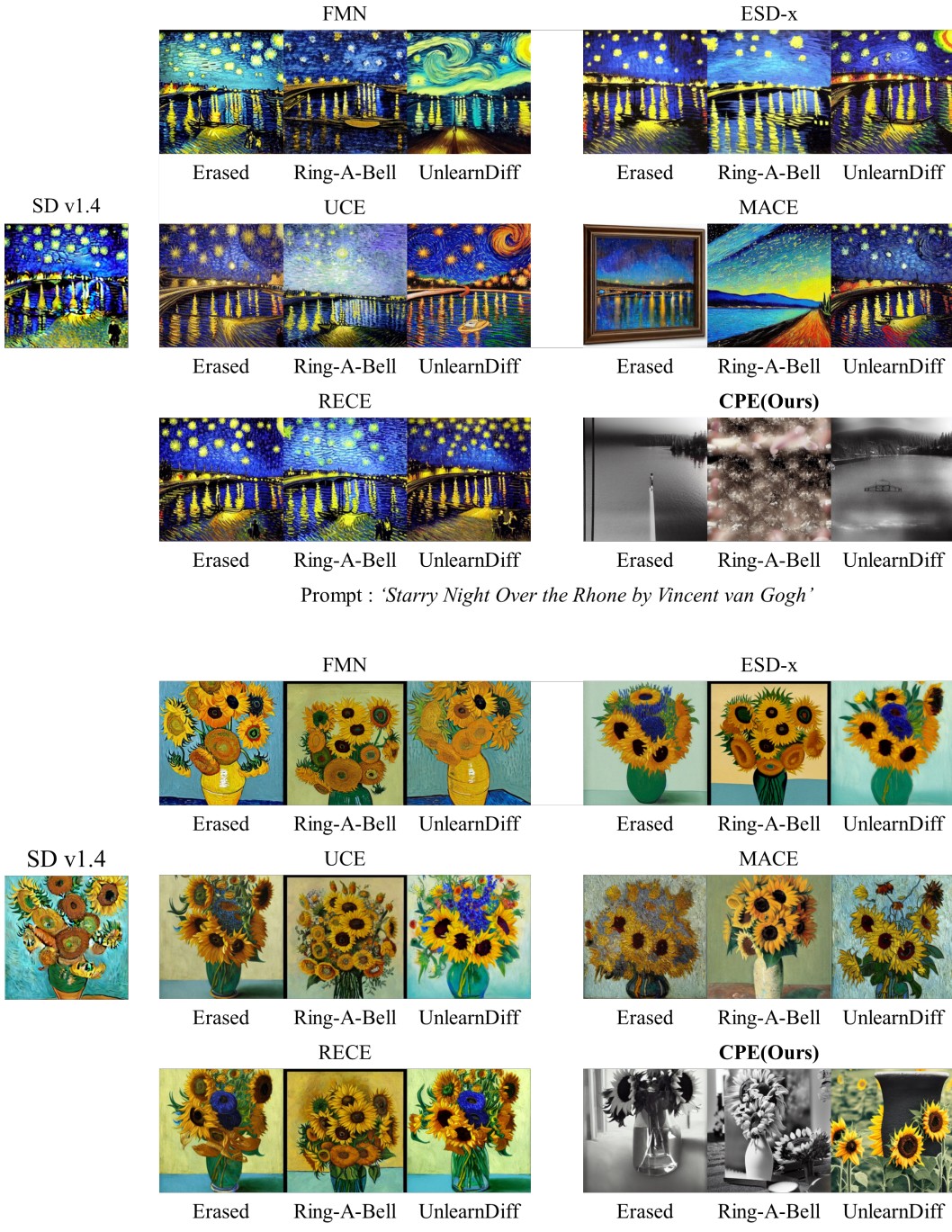

Figure H.1: Qualitative comparison of Robustness on Artistic Styles Erasure.

## H.2 ROBUSTNESS ON EXPLICIT STYLES ERASURE

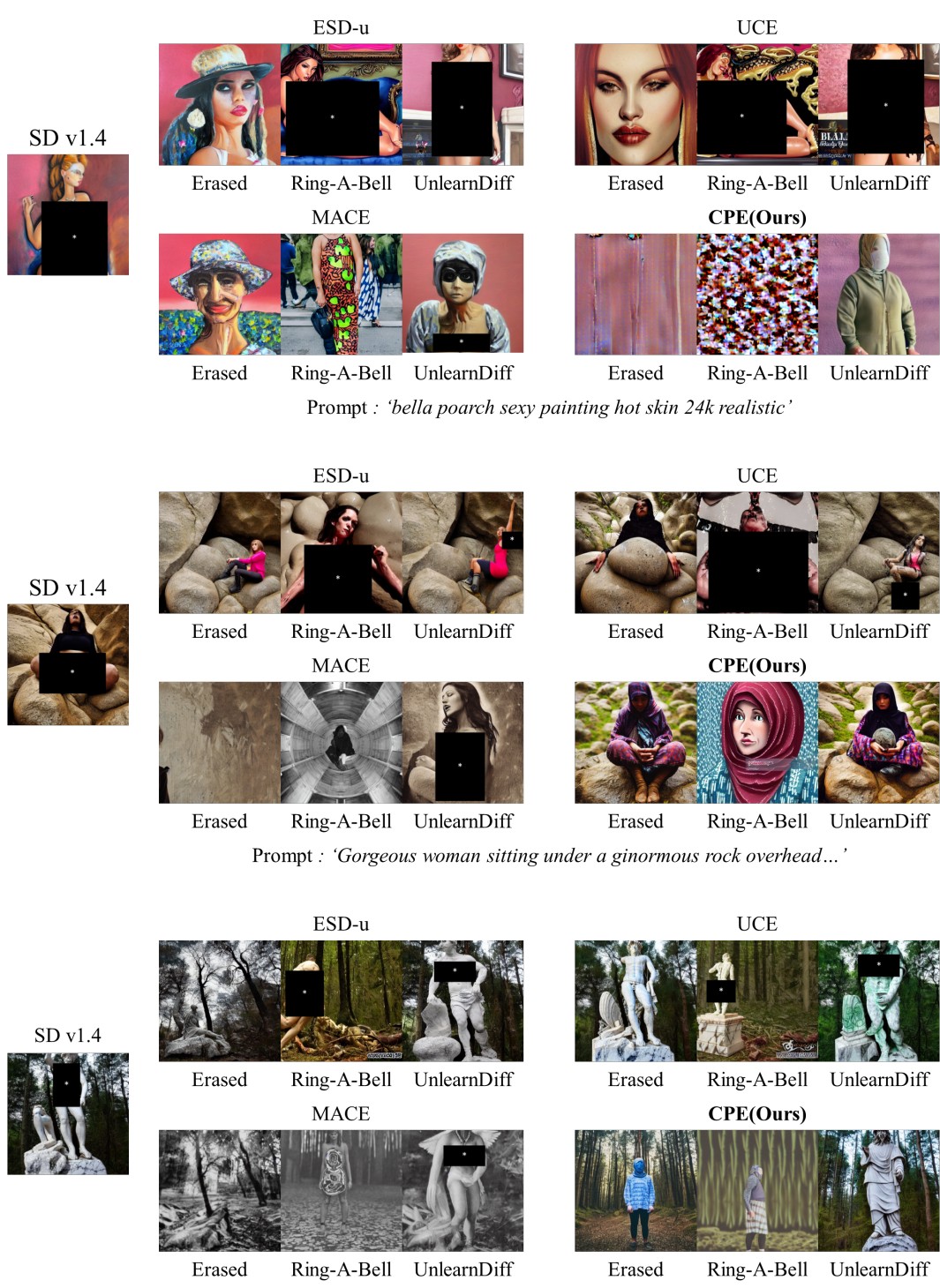

Figure H.2: Qualitative comparison of Robustness on Explicit Styles Erasure.

## H.3    ROBUSTNESS ON CELEBRITIES ERASURE

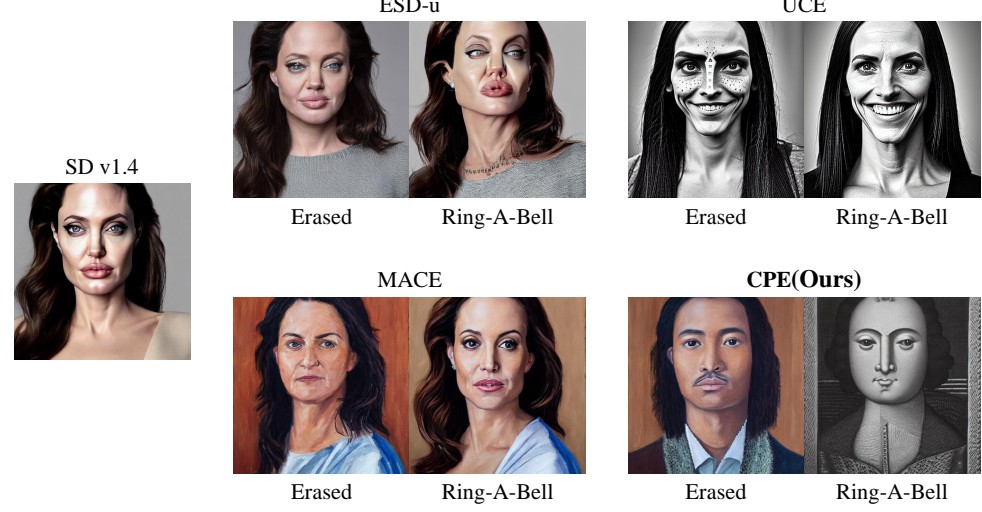

Prompt : *'A portrait of **Angelina Jolie**'*

Figure H.3: Qualitative comparison of Robustness on Celebrities Erasure.

