# OpenReview forum: "Concept Pinpoint Eraser for Text-to-image Diffusion Models via Residual Attention Gate"
_ICLR.cc/2025/Conference — ICLR 2025 Poster_

### Official Review · Reviewer_C9Fa · 2024-10-27

**Soundness:** 3
**Presentation:** 3
**Contribution:** 3
**Rating:** 8
**Confidence:** 4

**Summary:**

This paper focuses on unlearning specific target concepts from a pre-trained text-to-image diffusion model. It first proves that finetuning the query and key matrices in cross-attention layers, a method used in many prior works, may not preserve non-target concepts in the model. Building on this study, this paper introduces a non-linear additive module called Residual Attention Gate (ResAG), designed to selectively adjust the cross-attention layer weights for the target concept only. Combined with the proposed attention anchoring loss and adversarial training, the unlearned diffusion model is able to erase the target concept while preserving the remaining concepts.
Empirical results show that the overall framework achieves an optimal balance between erasing target concepts and preserving non-target ones across various domains. In addition, ablation studies confirm the effectiveness of each proposed component in both erasing and preserving concepts.

**Strengths:**

1. The theoretical analysis of finetuning cross-attention layers is solid and sound.
2. The experimental results show that the proposed CPE successfully preserves the remaining concepts while erasing the target concept better than previous methods across various domains.
3.  Each component in CPE is reasonable and shown to be effective in the ablation study.
4. The paper is well-organized and easy to follow.

**Weaknesses:**

1. The assumption for the theoretical analysis of the proposed ResAG raises some concerns. According to the paper, Equation 4 holds if the modes of samples can be detected and if $||f(E)||^2_F$ is small for remaining concepts. How can one determine the modes of samples? Additionally, what threshold for $||f(E)||^2_F$​ is considered "small enough" to effectively protect the remaining concepts?

**Questions:**

Multiple typos in the proof of Theorem 1 (Appendix B)
1. In the statement of Theorem 1, "Let $\tilde{W}^h_k$ and $\tilde{W}^h_k$ be the updated weights of $W^h_k$ and $W^h_k$." One of $\tilde{W}^h_k$ should be $\tilde{W}^h_v$ and one of $W^h_k$ should be $W^h_v$.
2. Line 795, "$|| \sum^{H}_{h=1} f_1(\tilde{W}^h_k E)f_2(\tilde{W}^h_v E) - f_1(W^h_v E)f_2(W^h_k E) ||_2$". The first term should be $f_1(\tilde{W}^h_v E)f_2(\tilde{W}^h_k E)$.
3. Line 798, same typo as above
4. Line 808, $L_{W^i_o, W^i_v} M$ should be $L_{W^h_o, W^h_v} M_1$

---

> ### Author Response · Authors · 2024-11-20
> **Official Comment by Authors**
>
> We thank the reviewer for their insightful comments and for appreciating the solidity and soundness of the theoretical analysis, confirming experimental results and ablation studies, and a clear structure of the paper.
>
> ${ }$
>
> >**Weakness 1. The assumption for the theoretical analysis of the proposed ResAG raises some concerns. According to the paper, Equation 4 holds if the modes of samples can be detected and if $\|f(\mathbf{E})\|_F^2$ small for remaining concepts. How can one determine the modes of samples? Additionally, what threshold for $\|f(\mathbf{E})\|_F^2$ is considered "small enough" to effectively protect the remaining concepts?**
>
> **Response:**  Detecting the modes of embeddings of concepts by $f(\mathbf{E})$ can be implemented as a classifier for concepts. To further clarify this, we consider the following simple example:
>
> ---
> Let $f(\mathbf{E}) = \alpha(\mathbf{E}) \mathbf{I}$ where $\alpha(\mathbf{E}) \ge 0$, $\mathbf{I}$ is an identity matrix, and $\mathcal{D} _{\textnormal{tar}}$ is the distribution of embeddings for target concepts. Suppose that $\alpha(\mathbf{E})$ can classify the distribution from which embeddings of target concepts are sampled or
> $
> \alpha(\mathbf{E}) = 1 \ \text{if} \  \mathbf{E} \sim \mathcal{D} _{\text{tar}} \ \text{else} \ 0 \.
> $
> Then, Equation (4) becomes zero for $\mathbf{E} _{\textnormal{rem}}$, suppressing $\mathbb{E} _{\mathbf{\mathbf{E} _{\textnormal{rem}}}} \left[  \| \Delta \mathbf{W} \mathbf{E} _{\textnormal{rem}} V(\mathbf{E} _{\textnormal{rem}}) \|_F^2 \right]$.
>
> ---
>
> We included this to the main text as Proposition 1 along with additional explanations (Line 205-224). Thus, we replaced the expression "small" with "suppressed to zero" for
> $\| f(\mathbf{E}) \|_F^2$ (Line 206).
>
>
> ${ }$
>
> >**Question 1. Multiple typos in the proof of Theorem 1 (Appendix B)**
>
> **Response:** Thank you for pointing out the typos. We incorporated all the corrections into the revision (Line 147, 834, 849-853, 862, 885).

---

> ### Comment · Reviewer_C9Fa · 2024-11-23
>
> Thank you for providing further explanations. However, I still have two questions regarding the modes of embeddings of concepts and the modifications (lines 218~220) made in the main text.
> Suppose the target concepts to be erased are "Andrew Garfield", "Bill Clinton", and "Keanu Reeves".
> 1. What are the modes in this example? Could you please clarify how "modes" are defined in the context of these target concepts?
> 2. Why would $f(\mathrm{E})$ respond (either activated or suppressed) to the single word "Bill" for the concept "Bill Clinton" or "Bill Murray"? I thought each concept is encoded as a text embedding (i.e., "Andrew Garfield" is encoded as $e_1$, "Bill Clinton" as $e_2$, "Keanu Reeves" as $e_3$, and $\mathrm{E}_{tar}=[e_1, e_2, e_3]$), and $f(\mathrm{E})$ would respond to the full concept embeddings rather than to individual words within those concepts.

---

> > ### Author Response · Authors · 2024-11-24
> > **Response to Reviewer's Further Questions**
> >
> > Thank you for your constructive questions for our first revision.
> >
> > ${ }$
> >
> > >**1. Suppose the target concepts to be erased are "Andrew Garfield", "Bill Clinton", and "Keanu Reeves". What are the modes in this example? Could you please clarify how "modes" are defined in the context of these target concepts?**
> >
> > **Response**: We appreciate your comment, which allow us to clarify it. Initially, we referred to the mode of each name as the peak value of the conditional distribution of text embeddings given the name, which can also be considered as the representative point of the distribution. Specifically, text prompts containing "Andrew Garfield" are encoded as text embeddings $\mathbf{E}_{\text{Andr}}$ of the prompts in our work, constructing conditional distribution given "Andrew Garfield". For example, text embeddings of "A portrait of Andrew Garfield" or "Andrew Garfield in an official photo" are the instances of $\mathbf{E} _{\text{Andr}}$. In a similar way, we can conceive the conditional ditributions given "Bill Clinton" or "Keanu Reeves". Thus, our initial expression "detect modes" meant "detect concepts through the representative points of their conditional distributions." However, this expression may be slightly unclear to convey our intetions and how we can determine such representative points. Thus, we replaced "detect modes" with "detect concepts" for clarity in a new revision (see lines 198-199, 206, highlighted in red).
> >
> > ${ }$
> >
> > >**2. Why would $f(\textbf{E})$ respond (either activated or suppressed) to the single word "Bill" for the concept "Bill Clinton" or "Bill Murray"? I thought each concept is encoded as a text embedding (i.e., "Andrew Garfield" is encoded as $e_1$, "Bill Clinton" as $e_2$, "Keanu Reeves" as $e_3$, and $\mathbf{E}_{\text{tar}}=[e_1, e_2, e_3]$), and $f(\mathbf{E})$ would respond to the full concept embeddings rather than to individual words within those concepts.**
> >
> > **Response:** Note that text tokens encode each word and the names like "Bill Clinton" or "Bill Murray" are encoded with two text tokens where one token for "Bill" is shared. More formally, we denote $\mathbf{E}$ as a text embedding for all tokens within a text prompt and $e$ as a token of the text embedding corresponding to a word in the text prompt. That is, the text embedding for "Bill Clinton" can be represented as $\mathbf{E} _{1} = [e _{\text{Bill}}, e _{\text{Clinton}}]$, and the text embedding for "Bill Murray" as $\mathbf{E} _{2} = [e _{\text{Bill}}, e _{\text{Murray}}]$. Then, both $f(\mathbf{E} _1)$ and $f(\mathbf{E} _2)$ are influenced by $e _{\text{Bill}}$ within the text embedding. Therefore, if "Bill Clinton" is the target concept and "Bill Murray" is the remaining concept, $f(\mathbf{E}_1)$ should be activated while $f(\mathbf{E}_2)$ be suppressed even if they contain the same token for "Bill". We revised the explanation of this example in Lines 216-220 of the main text (highlighted in red). We also changed the term for $e_i$ from "a text embedding" to "a token in a text embedding" in Theorem 2 (Lines 180-183) of the main text for clarity.

---

> > > ### Comment · Reviewer_C9Fa · 2024-11-25
> > >
> > > All of my questions have been addressed. This paper presents a solid theoretical analysis and an effective method. The authors have also revised the main text to improve clarity. Therefore, I am willing to raise my score to 8.

---

> ### Author Response · Authors · 2024-11-26
> **Thank You for  Your Comment**
>
> We are glad that all of the reviewer’s questions have been addressed. We also appreciate the increase in the score.

---

### Official Review · Reviewer_jZkb · 2024-11-04

**Soundness:** 3
**Presentation:** 4
**Contribution:** 3
**Rating:** 6
**Confidence:** 3

**Summary:**

This work proposes dubbed Concept Pinpoint Eraser by implementing Residual Attention Gates, improving both erasing and perserving performances. The study begins with a numerical analysis of the trade-off between these two aspects, given the current network structure. It then explores the addition of residual attention gates to address this dilemma. The effectiveness and robustness of the approach are evaluated through extensive quantitative and qualitative experiments. An ablation study is conducted to validate the effectiveness of each component.

**Strengths:**

1. To my knowledge, there are no significant flaws in the proofs of theorems; the work appears solid and reliable.
2. The experiments demonstrate the effectiveness of this approach, and the performance gains are compelling enough to validate the work, even though it shows weaker results on certain erasing tasks.
3. Additionally, the ablation study confirms the effectiveness of each component.

**Weaknesses:**

1.The robustness training seems to be a separate endeavor from the residual attention gates, while adversarial training acts more as a supplementary effort to enhance ResAGs. The robustness evaluation is directly performed on the model assessment. Does this indicate that the robustness of the model achieved through direct training is weaker than that of other models?
2. Given the proof, the addition of an extra term could be represented by various architectures. However, why do ResAGs demonstrate superior performance? This comparison is missing and not addressed in the discussion or experiments.
3. A concise algorithm description is recommended to enhance understanding of the training processes.

**Questions:**

1. The first and second weaknesses need to be addressed to ensure a thorough assessment of the proposed model.
2. The third weakness involves minor editing to enhance the overall presentation.

---

> ### Author Response · Authors · 2024-11-20
> **Official Comment by Authors**
>
> We thank the reviewer for their insightful comments and positive feedbacks regarding the solidity and reliability of the theorem proofs, the experiments demonstrating the effectiveness of the proposed approach, and the convincing ablation studies.
>
> ${ }$
>
> >**Weakness 1.The robustness training seems to be a separate endeavor from the residual attention gates, while adversarial training acts more as a supplementary effort to enhance ResAGs. The robustness evaluation is directly performed on the model assessment. Does this indicate that the robustness of the model achieved through direct training is weaker than that of other models?**
>
> **Response:** Direct training our architecture with ResAGs is challenging due to ResAG's nonlinearity. Thus, RARE was developed to synergetically train the network with ResAGs. See the below table, showing the effectiveness of our RARE (can improve both the prior structure without ResAGs and our CPE with ResAGs) as well as the synergy of our RARE (significantly improve our CPE with ResAGs over the prior structure without ResAGs). We added this result in Appendix E. (Line 1522-1526) and Tabel E.5 in the revision.
>
> **Table: Ablation of CPE on robust concept erasure against adversarial attack prompts: UnlearnDiff (Zhang et al., 2023c).**
>
> |**Method**|**UnlearnDiff in I2P Nudity ↓**|
> |---|---|
> |CPE w/o ResAG (Similar to UCE, MACE), directly trained |79.58|
> |CPE w/o ResAG (Similar to UCE, MACE), trained with RARE|52.82|
> |CPE, directly trained |82.39|
> |**CPE, trained with RARE (Ours)**|**30.28**|
>
> ${ }$
>
> >**Weakness 2. Given the proof, the addition of an extra term could be represented by various architectures. However, why do ResAGs demonstrate superior performance? This comparison is missing and not addressed in the discussion or experiments.**
>
> **Response:** Our corollary 1 suggested to use an embedding-dependent projection, which is a switch (gate), so we build up our ResAGs by adding this gate to the usual cross-attention in prior works. However, as suggested, we performed ablation studies as shown in the below table, demonstrating that our ResAG yielded improved performance over gate only or gate with linear projection architectures. We included this ablation study in Appendix E. (Line 1509-1509) and Tabel E.4 in the revision.
>
> **Table: Ablation study on the effect of the components of ResAG. The row with **bold** represents the selected configurations.**
>
> | **Architecture** | **CS (Target Celebs) ↓**| **ACC (Target Celebs) ↓**| **CS (Remaining Celebs) ↑** | **ACC (Remaining Celebs) ↑** | **KID (Remaining Celebs) ↓** | **CS (Remaining Arts) ↑** | **KID (Remaining Arts) ↓** | **CS (COCO-1K) ↑** | **KID (COCO-1K) ↓** |
> |---|---|---|---|---|---|---|---|---|---|
> |Gate only |21.45|1.13|34.02|84.34|0.20|28.53|0.19|30.78|0.23
> |Gate + linear projection |21.05|0.43|34.42|87.01|0.15|28.61|0.18|30.91|0.14|
> |**ResAGs**|**20.79**|**0.37**|**34.82**|**88.26**|**0.08**|**29.01**|**0.01**|**31.29**|**0.05**|
> |SDv1.4 (Rombach et al., 2022b)|34.49|91.35|34.83|90.86|-|28.96|-|31.34|-|
>
> ${ }$
>
> >**Weakness 3. A concise algorithm description is recommended to enhance understanding of the training processes.**
>
> **Response:** As suggested, we included the description of the training processes as Algorithm 1 in the revision. To represent this more comprehensively, we revised the notations in Equations (7)-(10) in the main text (Line 256-260, 267-269, 313-315, 319-323). We also included one sentence referring to it (Line 357-358).

---

> > ### Comment · Reviewer_jZkb · 2024-11-27
> > **Reply to authors' response**
> >
> > The answers addressed my concerns. I would recommend the revised manuscript provides a clearer explanation about "Direct training our architecture with ResAGs is challenging due to ResAG's nonlinearity." I appreciate the responses and would recommend that this paper be accepted.

---

> > > ### Author Response · Authors · 2024-11-27
> > > **Reply to Reviewer's Response**
> > >
> > > We are pleased that reviewer's concerns have been addressed. We also appreciate the recommendation to enhance the clarity of the presentation of RARE. We incorporated it into the new revision at Line 305-306 in the main text.

---

### Official Review · Reviewer_SM5N · 2024-11-04

**Soundness:** 3
**Presentation:** 2
**Contribution:** 2
**Rating:** 6
**Confidence:** 4

**Summary:**

This paper proposes the Concept Pinpoint Eraser (CPE) approach for the purpose of addressing both erasure and preservation in Concept Erasure domain. Specifically, the paper first mathematically proves that modifying the linear mapping matrix of cross-attention cannot achieve both good erasure and preservation results. Then, the paper proposes nonlinear Residual Attention Gates (ResAGs), which are used to reduce the impact on irrelevant concepts. The paper trains ResAGs with novel erasing loss, attention anchoring loss, and adversarial training, and illustrates experimentally the excellent erasure effect, preservation effect, and robustness of the methods.

**Strengths:**

1. This paper elucidates the inadequacy of simply modifying the linear mapping matrix of cross-attention through a solid mathematical proof, which leads to a non-linear ResAgs module and a novel training loss function.
2. This paper conducts a large number of experiments, including character privacy, art style, NSFW content, etc., comparing many baseline methods.

**Weaknesses:**

1. The major weakness of this work is that the propose method may not work in open-source scenarios. Methods like ESD, UCE, RECE, etc. directly modify the UNet, and the LoRA of MACE can merge into the UNet, so all these changes can be directly applied to the UNet, and thus the user can't directly bypass these security mechanisms. CPE uses a non-linear add-on module, which cannot be merged to the UNet, so in the open-source scenario, the user can directly delete the code of the add-on module, thus skipping the security mechanism.

2.  Some parts are not presented clearly, which makes it difficult to follow. I have put the comments in the question section.

**Questions:**

1. From Table D.7, CPE uses hundreds of anchoring concepts to enhance retention. MACE only used 'a person' in the deletion of celebrities[1]. UCE, RECE only used the provided 1724 styles in the deletion of artistic styles. ESD, UCE, and RECE used only the empty text "", and MACE only used "a person wearing clothes" [ 2] in the deletion of NSFW concept. Can the authors explain whether the preservation effect of CPE comes from the methods proposed in the main text or from these numerous anchor concepts in the appendix?
2. For NSFW erasure, CPE erased 4 concepts, "nudity, naked, eratic, sexual" similar with  MACE. The more concepts that are deleted, the more effective the deletion tends to be. However, UCE, ESD, and RECE all remove only one word, "nudity", can the authors provide the results of removing only "nudity"?
3. From APpendix D.2, the authors set negative prompts for celebrities erasure. Was this negative cue word used in all baselines during the experimental comparisons?

---

> ### Author Response · Authors · 2024-11-20
> **Official Comment by Authors-1**
>
> We appreciate the reviewer’s thoughtful comments and encouraging feedbacks on the solid mathematical proof, novelty of the proposed training loss function, extensive experiments.
>
> ${ }$
>
> > **Weakness 1. The major weakness of this work is that the propose method may not work in open-source scenarios. Methods like ESD, UCE, RECE, etc. directly modify the UNet, and the LoRA of MACE can merge into the UNet, so all these changes can be directly applied to the UNet, and thus the user can't directly bypass these security mechanisms. CPE uses a non-linear add-on module, which cannot be merged to the UNet, so in the open-source scenario, the user can directly delete the code of the add-on module, thus skipping the security mechanism.**
>
> **Response:** We would like to point out that a misuse is different from an attack to the security mechanism. In fact the user in your example seems like a superuser with all the privileges to access the code, the model and even the computation to run the model and thus has a lot of options to compromise the model like, not only modifying codes as you mentioned, but also fine-tuning the model to recover erased concepts, which can undermine all the works including the merged approaches. This is the case of misusing the open-source code (e.g., misused deepfake for crime), rather than the case of skipping the security mechanism. For example, if one user ran a modified linux kernel on his or her own computer for potentially bad purpose, it is not considered breaking the security mechanism, but misusing the software, to the best of our knowledge.
>
> Security mechanism for the concept erasing in generative diffusion models should be well-defined. For example, attacker can distribute a modified bad code to other users to run, but this attack can be easily detected in open-source scenarios. Another example is for attackers to modify the input to the generative diffusion model during service to output malicious images and this setting has been used in most concept erasing works for robustness evaluation. This setting assumes the limited access of users to the input only, which is realistic in both open-source and closed-source scenarios (e.g., Veo of DeepMind [1], Firefly of Adobe [2], Sora of OpenAI [3], etc.). Thus, we also stick to this setting like prior works (see Table 4 for our results).
>
> [1] https://deepmind.google/technologies/veo/
>
> [2] https://www.adobe.com/products/firefly.html
>
> [3] https://openai.com/index/sora/
>
> ${ }$
>
> > **Question 1. From Table D.7, CPE uses hundreds of anchoring concepts to enhance retention. MACE only used 'a person' in the deletion of celebrities[1]. UCE, RECE only used the provided 1724 styles in the deletion of artistic styles. ESD, UCE, and RECE used only the empty text "", and MACE only used "a person wearing clothes" [2] in the deletion of NSFW concept. Can the authors explain whether the preservation effect of CPE comes from the methods proposed in the main text or from these numerous anchor concepts in the appendix?**
>
> **Response:** Note that our CPE also used "a person" like MACE and other prior works also used multiple anchor concepts as summarized in the below tables. We included them as Table D.3 and Table D.6 in Appendix. Note that MACE used much more than 200 anchor concepts (100 celebrities + much more than 100 captions from MS-COCO) while our CPE used 150 anchor concepts (50 similar concepts selected from 500 celebrities + 100 synonyms with "celebrities"). We believe that the way of using anchor concepts was important since our CPE with only 50 anchor concepts still outperformed prior arts as illustrated in Table C.2 in the appendix.
>
> ${ }$
>
> **Table. Surrogate and anchor concepts for celebrities erasure**
> | Method | Surrogate Concept | Anchor Concepts |
> |-|-|-|
> | FMN (Zhang et al., 2023a) | " " | - |
> | ESD (Gandikota et al., 2023) | " " | - |
> | UCE (Gandikota et al., 2024) | " " | 100 celebrities |
> | MACE (Lu et al., 2024) | "a person" | 100 celebrities + all captions from MS-COCO |
> | CPE | "a person" | 50 similar celebrities from 500 celebrities + 100 similar words with 'celebrities' |
>
> **Table. Surrogate and anchor concepts for artistic styles erasure**
> | Method | Surrogate Concept | Anchor Concepts |
> |-|-|-|
> | FMN (Zhang et al., 2023a) | " " | - |
> | ESD (Gandikota et al., 2023) | " " | - |
> | UCE (Gandikota et al., 2024) | " " | 1734 styles |
> | MACE (Lu et al., 2024) | "art" | 100 styles + all captions from MS-COCO |
> | CPE | "real photograph" | 50 similar styles from 1734 styles + 100 similar words with 'famous artists' |

---

> > ### Comment · Reviewer_SM5N · 2024-11-21
> > **Reply to authors' response**
> >
> > I appreciate the authors' efforts in replying my questions. Now I have several further questions bellow.
> > 1. Regarding to the reply for weakness 1, the model is indeed robust in closed-source scenarios, however,  in open-source scenarios, deleting the non-linear add-on module is not as difficult as claimed in the response. Firstly, since the authors used the white-box attack UnlearnDiff in experiments, the assumption of modifying the code is reasonable and doesn't require being a "superuser". Assuming that a model provider like Stable Diffusion uses the method to guard the generated content and uploads the model to huggingface, it's possible for a user to delete the non-linear add-on module related code and there are many similar tutorials available on the web, e.g. https://www.reddit.com/r/StableDiffusion/comments/wv2nw0/tutorial_how_to_remove_the_safety_filter_in_5/.
> > Secondly, for models that can be downloaded from huggingface, modifying the code is the most convenient attack method, more  easier than both UnlearnDiff and Ring-A-Bell.
> > 2. Another question I want to ask is that whether CPE robust to this red-teaming method? "Petsiuk V, Saenko K. Concept Arithmetics for Circumventing Concept Inhibition in Diffusion Models[C]//European Conference on Computer Vision. Springer, Cham, 2025: 309-325."

---

> > > ### Author Response · Authors · 2024-11-23
> > > **Response to Reviewer's Further Questions-1**
> > >
> > > >**1. Regarding to the reply for weakness 1, the model is indeed robust in closed-source scenarios, however, in open-source scenarios, deleting the non-linear add-on module is not as difficult as claimed in the response. ...**
> > >
> > > **Response:** Thanks for your acknowledgement about our CPE as "the model is indeed robust in closed-source scenarios".
> > >
> > > We are afraid that your comment "Firstly, since the authors used the white-box attack UnlearnDiff in experiments, the assumption of modifying the code is reasonable and doesn't require being a superuser" may not be entirely true. In fact, the term "white-box attack setting," as mentioned in UnlearnDiff (Zhang et al., 2023c) for red-teaming tools, does not indicate that code modification is possible. It refers to having access to both the original and unlearned models unlike "a black-box attack setting" assuming access only to the unlearned model. Note that these terms have already been used by other red-teaming approaches for diffusion models such as P4D [1] and Ring-A-Bell (Tsai et al., 2024). These approaches, whether white-box or black-box, do not directly edit the diffusion models (i.e., no code modification possible, no binary modification possible), thus they only leverage adversarial text prompts for attacks. This is why SPM (Lyu et al., 2024) and Receler (Huang et al., 2023) could propose add-on approaches for security mechanisms on concept erasing in diffusion models just like ours.
> > >
> > > If part of the code can be modified in open-source scenarios, then it is also possible to modify the unlearned model weight to be the original weight (by simply copying) since we know what part to switch and the original weight is available in open-source scenarios. In fact, numerous attack approaches such as deactivating the safety checker (as in the provided reddit link), deleting add-on modules, overwriting the unlearned model weights with the original weights or fine-tuning to generate intended malicious concepts, can be employed to regenerate target concepts, challenging the effectiveness of all concept erasure works. Meanwhile, representative red-teaming tools for T2I diffusion models like P4D, Ring-A-Bell, and UnlearnDiff, all consider attacking solely through the text prompts as a reasonable setting, where our CPE demonstrated remarkably enhanced results.
> > >
> > > We do NOT claim that our add-on approach is secure in open-source scenarios. We simply argue that your comment in weakness 1 is not only for add-on approaches like ours (remove code), but also for merged approaches (copy weights). In the meanwhile, it seems that we have different ideas on the security mechanism in open-source scenarios. However, we believe that various attacks such as code removal, weight copy and others in open-source scenarios should be carefully investigated for more general settings (not just for concept erasing), thus will be an interesting future work for those who are working on security systems for open-source scenarios.
> > >
> > > To sum up, the robustness of add-on approaches and merged approaches can be evaluated in white-box attack setting, following all prior works on concept erasing. However, in open-source scenarios, all approaches should be carefully used since the code and the original model weights are publicly available.
> > >
> > > [1] Chin, Zhi-Yi, et al. "Prompting4Debugging: Red-Teaming Text-to-Image Diffusion Models by Finding Problematic Prompts." *ICML*, 2023.

---

> > > > ### Comment · Reviewer_SM5N · 2024-11-27
> > > > **Reply to authors' response**
> > > >
> > > > Thanks for the efforts in addressing my concerns. Most of my concerns have been addressed now.

---

> > > > > ### Author Response · Authors · 2024-11-27
> > > > > **Thank You for Your Response**
> > > > >
> > > > > We are pleased that most of reviewer's questions have been addressed and sincerely appreciate the overall evaluation.

---

> > > ### Author Response · Authors · 2024-11-23
> > > **Response to Reviewer's Further Questions-2**
> > >
> > > >**2. Another question I want to ask is that whether CPE robust to this red-teaming method? "Petsiuk V, Saenko K. Concept Arithmetics for Circumventing Concept Inhibition in Diffusion Models. *ECCV*, 2024."**
> > >
> > > **Response:** Since the code of ARC attack is not publicly available, we implemented it and evaluated the robustness to the attack in explicit contents erasure. Following the setup of ARC attack, we first selected 95 original prompts $(S)$ from the I2P dataset, each with a nudity percentage value greater than 50% provided by the dataset. Then, we measured the number of explicit contents detected by the NudeNet detector with its threshold of detection score as 0.6 (used in our work, MACE, and Ring-A-Bell) because the threshold was not shared by the work of ARC attack. Since ARC attack introduces three types of attack prompts, $N_1$, $N_2$ and $N_3$, we evaluated on all these types of attack prompts. From the tables below, our CPE not only achieved the lowest detection counts for explicit content on the original prompts $(S)$, but also significantly outperformed the baselines in terms of robustness to all types of attack prompts $(N_1,N_2,N_3)$ by large margins.
> > >
> > > ${ }$
> > >
> > > **Table. Number of detected explicit contents using 95 original prompts extracted from I2P dataset.**
> > >
> > > |Method|Prompt type|Buttocks|Breast(F)|Genitalia(F)|Breast(M)|Genitalia(M)|Anus|Feet|Armpits|Belly|Total|
> > > |-|-|-|-|-|-|-|-|-|-|-|-|
> > > |ESD|$S$|0|9|0|6|0|0|2|6|16|39|
> > > |UCE|$S$|1|6|0|7|0|0|3|5|15|37|
> > > |MACE|$S$|0|2|0|0|0|0|1|0|2|5|
> > > |CPE|$S$|0|2|0|0|0|0|0|0|0|2|
> > >
> > > ${ }$
> > >
> > > **Table. Robustness to attack prompts of type $N_1$ proposed by ARC attack.**
> > >
> > > |Method|Prompt type|Buttocks|Breast(F)|Genitalia(F)|Breast(M)|Genitalia(M)|Anus|Feet|Armpits|Belly|Total|
> > > |-|-|-|-|-|-|-|-|-|-|-|-|
> > > |ESD|$N_1$|0|23|0|4|0|5|14|20|0|66|
> > > |UCE|$N_1$|1|17|0|4|0|0|2|3|17|44|
> > > |MACE|$N_1$|0|0|0|1|0|0|0|5|5|11|
> > > |CPE|$N_1$|0|0|0|0|0|0|0|0|0|0|
> > >
> > > ${ }$
> > >
> > > **Table. Robustness to attack prompts of type $N_2$ proposed by ARC attack.**
> > >
> > > |Method|Prompt type|Buttocks|Breast(F)|Genitalia(F)|Breast(M)|Genitalia(M)|Anus|Feet|Armpits|Belly|Total|
> > > |-|-|-|-|-|-|-|-|-|-|-|-|
> > > |ESD|$N_2$|2|44|4|4|0|0|15|20|26|115|
> > > |UCE|$N_2$|2|20|1|2|0|0|1|4|16|46|
> > > |MACE|$N_2$|0|0|0|0|0|0|2|3|2|7|
> > > |CPE|$N_2$|0|0|0|0|0|0|0|0|0|0|
> > >
> > > ${ }$
> > >
> > > **Table. Robustness to attack prompts of type $N_3$ proposed by ARC attack.**
> > >
> > > |Method|Prompt type|Buttocks|Breast(F)|Genitalia(F)|Breast(M)|Genitalia(M)|Anus|Feet|Armpits|Belly|Total|
> > > |-|-|-|-|-|-|-|-|-|-|-|-|
> > > |ESD|$N_3$|3|41|1|17|3|0|9|30|36|140|
> > > |UCE|$N_3$|0|23|1|9|0|0|2|3|12|50|
> > > |MACE|$N_3$|0|1|0|0|0|0|0|0|1|2|
> > > |CPE|$N_3$|0|0|0|0|0|0|0|0|0|0|

---

> ### Author Response · Authors · 2024-11-20
> **Official Comment by Authors-2**
>
> > **Question 2. For NSFW erasure, CPE erased 4 concepts, "nudity, naked, eratic, sexual" similar with MACE. The more concepts that are deleted, the more effective the deletion tends to be. However, UCE, ESD, and RECE all remove only one word, "nudity", can the authors provide the results of removing only "nudity"?**
>
> **Response:** Yes, see the below table for the results of removing only "nudity", demonstrating that our proposed CPE still outperformed prior arts by achieving the fewest number of detected NSFW contents, while maintaining high preservation performance on COCO-30K. We have included this table as Table C.5 in Appendix C.4 (Line 1114-1122).
>
>
> **Table. Explicit Contents Removal (I2P prompts)**
> |Method|Armpits|Belly|Buttocks|Feet|Breasts(F)|Genitalia(F)|Breasts(M)|Genitalia(M)|Total$\downarrow$|FID-30K$\downarrow$|CS-30K$\uparrow$|
> |-|-|-|-|-|-|-|-|-|-|-|-|
> |FMN (Zhang et al., 2023a))|43|117|12|59|155|17|19|**2**|424|30.39|13.52|
> |ESD-x (Gandikota et al., 2023)|59|73|12|39|100|6|18|8|315|30.69|14.41|
> |ESD-u (Gandikota et al., 2023)|32|30|**2**|19|27|3|8|**2**|123|30.21|15.10|
> |UCE (Gandikota et al., 2024)|29|62|7|29|35|5|11|4|182|30.85|14.07|
> |MACE (Lu et al., 2024)|17|19|**2**|39|16|2|9|7|111|29.41|13.42|
> |RECE (Gong et al., 2024)|31|25|3|**8**|**10**|**0**|9|3|89| 30.95 | - |
> |**CPE(One word)**| **15**| **13**|5|15|11|2|**3**|**2**|**65**|**31.30**|**12.88**|
> |SDv1.4[47]|148|170|29|63|266|18|42|7|743|31.34|14.04|
> |SDv2.1[45]|105|159|17|60|177|9|57|2|586|31.53|14.87|
>
> ${ }$
>
> > **Question 3. From Appendix D.2, the authors set negative prompts for celebrities erasure. Was this negative cue word used in all baselines during the experimental comparisons?**
>
> **Response:** Yes, we applied the same negative prompts to all the baselines for fair comparisons.

---

### Meta-Review · Area_Chair_Pcfh · 2024-12-17

**Metareview:**

The paper is above the acceptance threshold due to its solid theoretical foundation, extensive experiments, and effective novel contributions. Reliable proofs and strong ablation studies validate the ResAGs module and training loss, addressing key limitations of cross-attention layers. Experiments across diverse domains demonstrate consistent performance gains and robust concept preservation. Despite minor clarity and applicability concerns, the work’s sound analysis, compelling results, and well-organized presentation make it a valuable contribution.

**Additional Comments On Reviewer Discussion:**

N/A

---

### Decision · Program_Chairs · 2025-01-22

Accept (Poster)